# In-situ local phase-transitioned MoSe$_2$ in La$_{0.5}$Sr$_{0.5}$CoO$_{3-\delta}$ heterostructure and stable overall water electrolysis over 1000 hours

Nam Khen Oh[1], Changmin Kim[2], Junghyun Lee[1], Ohhun Kwon[2], Yunseong Choi[1], Gwan Yeong Jung[2], Hyeong Yong Lim[2], Sang Kyu Kwak [ID][2], Guntae Kim[2] & Hyesung Park[1]

Developing efficient bifunctional catalysts for overall water splitting that are earth-abundant, cost-effective, and durable is of considerable importance from the practical perspective to mitigate the issues associated with precious metal-based catalysts. Herein, we introduce a heterostructure comprising perovskite oxides (La$_{0.5}$Sr$_{0.5}$CoO$_{3-\delta}$) and molybdenum diselenide (MoSe$_2$) as an electrochemical catalyst for overall water electrolysis. Interestingly, formation of the heterostructure of La$_{0.5}$Sr$_{0.5}$CoO$_{3-\delta}$ and MoSe$_2$ induces a local phase transition in MoSe$_2$, 2 H to 1 T phase, and more electrophilic La$_{0.5}$Sr$_{0.5}$CoO$_{3-\delta}$ with partial oxidation of the Co cation owing to electron transfer from Co to Mo. Together with these synergistic effects, the electrochemical activities are significantly improved for both hydrogen and oxygen evolution reactions. In the overall water splitting operation, the heterostructure showed excellent stability at the high current density of 100 mA cm$^{-2}$ over 1,000 h, which is exceptionally better than the stability of the state-of-the-art platinum and iridium oxide couple.

[1] Department of Energy Engineering, School of Energy and Chemical Engineering, Low Dimensional Carbon Materials Center, Perovtronics Research Center, Ulsan National Institute of Science and Technology (UNIST), Ulsan 44919, Republic of Korea. [2] Department of Energy Engineering, School of Energy and Chemical Engineering, Ulsan National Institute of Science and Technology (UNIST), Ulsan 44919, Republic of Korea. These authors contributed equally: Nam Khen Oh, Changmin Kim. Correspondence and requests for materials should be addressed to S.K.K. (email: skkwak@unist.ac.kr) or to G.K. (email: gtkim@unist.ac.kr) or to H.P. (email: hspark@unist.ac.kr)

Hydrogen has high energy density (120–140 MJ kg$^{-1}$) and is an environmentally friendly clean energy source that can be produced through water splitting[1,2]. The state-of-the-art water splitting catalyst for hydrogen production is composed of precious noble metals, and it has several drawbacks such as high-cost, limited reserves, and durability[3]. Currently, the Ir- and Ru-based compounds and Pt-group metals are considered the most efficient catalysts for the anodic oxygen evolution reaction (OER) and the cathodic hydrogen evolution reaction (HER), respectively[3,4]. Because each half-cell reaction requires a different pH environment for yielding the best performance, the adoption of such electrodes in integrated electrochemical water splitting has been limited in practical settings. To achieve efficient overall water splitting for hydrogen production, OER performance is also important because the OER process can become a kinetic bottleneck in HER owing to its complex and slow overall reaction involving concerted electron–proton shift steps[3–5]. Therefore, bifunctional catalysts that are earth-abundant and can exhibit concurrently excellent HER and OER performance are required urgently[6,7].

ABO$_3$ perovskite oxides (A: rare-earth or alkaline earth element, B: transition metal ion) have received significant attention as potential alternatives to precious metal-based catalysts (e.g., RuO$_2$ and IrO$_2$) owing to their strong catalytic activity, robust stability, and compositional flexibility[8,9]. Thus, considerable efforts have been devoted to understanding the mechanisms of OER and HER on perovskite oxides, and molecular orbital studies have suggested that cobalt-based oxides can be used as active catalysts in OER and HER[10]. Among the various ABO$_3$ perovskite oxide catalysts, La$_{1-x}$Sr$_x$CoO$_{3-\delta}$ has been extensively studied for its strong catalytic activity. Liu et al. synthesized hierarchical mesoporous La$_{1-x}$Sr$_x$CoO$_{3-\delta}$ by electrospinning and showed improved OER activity and redox stability in a lithium oxygen battery[11]. Mefford et al. showed the manner in which OER performance can be improved by exploiting the oxygen vacancy defect in La$_{1-x}$Sr$_x$CoO$_{3-\delta}$[12]. Grimaud et al. demonstrated that the O$_2$ generated from the lattice oxygen of La$_{1-x}$Sr$_x$CoO$_{3-\delta}$ significantly influenced OER[13].

Both theoretical and experimental investigations on transition metal dichalcogenides (TMDs) have revealed the great potential of TMDs as hydrogen generation catalysts owing to their high catalytic activity; robustness to CO, CO$_2$, and O$_2$; affordability; and scalability[14,15]. Of the various TMDs, molybdenum diselenide (MoSe$_2$) is considered a promising HER catalyst because of its relatively superior electrochemical catalytic activity and chemical stability compared to other TMDs[16,17]. The Gibbs free energy of MoSe$_2$ for hydrogen adsorption is close to zero, and its hydrogen coverage is greater than those of other TMDs[17]. However, owing to low conductivity of the intrinsic 2H-phase MoSe$_2$, MoSe$_2$-based composite structures, such as MoSe$_2$/carbon cloth[16], MoSe$_2$/n$^+$p-Si[18], and MoSe$_2$/graphene[19], have been typically used to improve the electrochemical activity of intrinsic MoSe$_2$. In addition, inducement of the semiconducting (2H) to metallic (1T) phase transition in TMDs has been considered to improve the performance of TMDs-based composite electrochemical catalysts because the metallic phase can improve their intrinsic electrocatalytic nature[20]. However, the phase transition process of TMDs is rather complex and time-consuming, and it requires an inert environment owing to the highly reactive materials involved, such as alkali metals[20,21].

Apart from performance, the durability of water splitting catalysts is an important criterion from the commercial perspective. Chemical instability results in catalyst decomposition in the electrode during continued operation, which hinders long-term catalyst stability. Therefore, developing earth-abundant bifunctional catalysts that are based on non-precious metal elements

and concurrently offer excellent HER and OER performance along with robust chemical stability is indispensable to ensure the industrial viability of electrochemical water splitting.

In present work, we devise a composite perovskite oxide–TMD heterostructure composed of MoSe$_2$ (denoted MoSe$_2$), La$_{0.5}$Sr$_{0.5}$CoO$_{3-\delta}$ (denoted LSC only), and Ketjenblack carbon (denoted KB) as a bifunctional electrocatalyst for overall water electrolysis. The LSC, MoSe$_2$, and KB heterostructure (denoted LSC&MoSe$_2$) offers considerably better HER and OER performances (onset potential, Tafel slope) than LSC and KB heterostructure (denoted LSC) or MoSe$_2$ alone. Interestingly, an in-situ local phase transition in MoSe$_2$ (from 2H- to 1T-MoSe$_2$) is observed during the formation of LSC&MoSe$_2$, possibly because of spontaneous electron transfer from Co to Mo. This charge transfer is expected to enhance the intrinsic conductivity of MoSe$_2$ and increase the amount of Co–O and Co–OH in LSC, which can enhance the water splitting catalytic activity. When LSC&MoSe$_2$ ‖ LSC&MoSe$_2$ electrode was applied to overall water splitting, the initiation potential was observed at 1.52 V, and the proposed electrode exhibited excellent overall water electrolysis stability over 1000 h at a high current density of 100 mA cm$^{-2}$, which is far superior performance compared to that of the Pt/C ‖ IrO$_2$ electrode.

## Results

**Morphological and structural properties of LSC&MoSe$_2$.** LSC&MoSe$_2$ was prepared using the high-energy ball milling process with the optimum weight ratio of LSC:MoSe$_2$:KB = 6:3:1 determined by the electrochemical analyses (Supplementary Figure 1–3; see Experimental Section for details). Morphological and structural analyses of the composite electrocatalyst were first performed by transmission electron microscopy (TEM) and scanning electron microscopy (SEM). Figure 1a, b show bright-field TEM image and high-angle annular dark-field (HAADF) image of LSC&MoSe$_2$, along with energy-dispersive spectroscopy (EDS) elemental mapping, which clearly illustrates the presence of the associated elements (La, Sr, Co, O, Mo, Se, and C) in LSC&MoSe$_2$. Further analysis on the morphology of LSC&MoSe$_2$ heterostructure was carried out by SEM (Supplementary Figure 4). It can be seen that MoSe$_2$ nanoflakes are randomly distributed and adsorbed onto the LSC surface without causing any noticeable aggregation, which can contribute to the increase of overall surface area of the composite structure. The SEM-EDS and elemental quantitative analysis for LSC&MoSe$_2$ further revealed that each constituent atomic component in the composite structure was clearly observed with expected elemental ratio (Supplementary Figure 5). Figure 1c shows a high-resolution TEM (HR-TEM) image of LSC&MoSe$_2$, which highlights the presence of MoSe$_2$ (red). The inset in Fig. 1c shows the fast Fourier transform (FFT) pattern of MoSe$_2$ viewed along the <001> zone axis, indicating highly crystalline structures corresponding to the (100) plane of 2H-MoSe$_2$ with a lattice spacing of 0.28 nm. A HR-TEM image and the corresponding FFT pattern (inset, white section) of LSC along the <110> zone axis are shown in Fig. 1d, which highlight the crystalline structure of LSC corresponding to the (001) plane with a lattice spacing of 0.40 nm. The crystal structures of LSC, MoSe$_2$, and LSC&MoSe$_2$ were analyzed further by using X-ray diffraction (XRD), as shown in Fig. 1e. The peaks related to LSC and MoSe$_2$ can be observed clearly in the XRD pattern of LSC&MoSe$_2$, indicating the well-mixed state of LSC and MoSe$_2$ without the presence of any additional phase, along with the amorphous state of KB.

We then performed Brunauer–Emmett–Teller (BET) analysis on LSC&MoSe$_2$ and LSC to investigate the effect of MoSe$_2$ on the surface area of the heterostructure. We also measured the BET

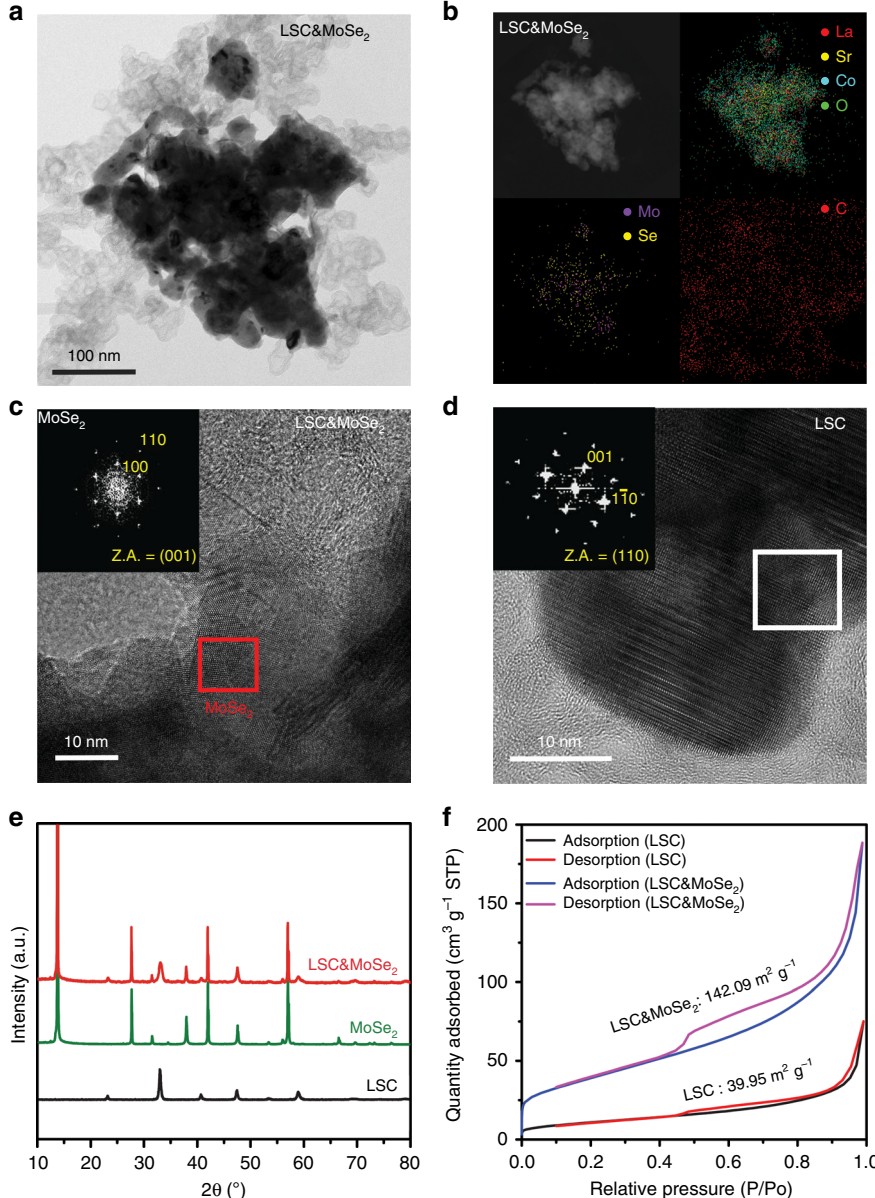

**Fig. 1** Morphological and structural characterizations of LSC&MoSe₂. **a** Bright-field TEM image of as-prepared LSC&MoSe₂. **b** STEM-HAADF image and corresponding STEM-EDS elemental mapping of LSC&MoSe₂, demonstrating uniform atomic distributions of La (red), Sr (yellow), Co (cyan), O (green), Mo (purple), Se (yellow), and C (red). **c** HR-TEM image of LSC&MoSe₂, indicating the presence of MoSe₂. Inset shows a FFT image of MoSe₂. **d** HR-TEM image of LSC showing different lattice structure from that of MoSe₂. The inset is a FFT image of LSC. **e** XRD spectra of LSC, MoSe₂, and LSC&MoSe₂, indicating the well-mixed state of LSC and MoSe₂. **f** BET surface area calculated from N₂ adsorption/desorption isotherms of LSC and LSC&MoSe₂, demonstrating that the surface area of LSC&MoSe₂ is considerably greater than that of LSC

surface area of MoSe₂ and LSC only as 32.55 and 10.47 m² g⁻¹, respectively (Supplementary Figure 6). However, as shown in Fig. 1f, the addition of MoSe₂ to LSC led to a notable increase in the total surface area of the composite structure, where the surface area of LSC&MoSe₂ (142.09 m² g⁻¹) was more than thrice that of LSC (39.95 m² g⁻¹). Statistical analysis of the aforementioned samples is provided in Supplementary Figure 7. Increase in the BET surface area for LSC&MoSe₂ can be attributed to the additional MoSe₂ nanoflakes present that are adsorbed onto the LSC surface. Such a remarkable increase in surface area improves water splitting by increasing the total number of active electrocatalysis sites for both HER and OER[22]. The pore size of LSC&MoSe₂ was investigated using the Barrett–Joyner–Halenda (BJH) method. The pore size distribution in Supplementary Figure 8 shows that mesoporous pores (2–50 nm) are present on

the surface of the heterostructure. We analyzed the BJH pore size distributions of MoSe₂ and LSC only to investigate the origin of mesoporous characteristics in LSC&MoSe₂ (Supplementary Figure 9). In MoSe₂, a sharp peak primarily centered at around 4 nm is observed, indicating the presence of mesoporous structure, whereas the broad peak centered at 55 nm appears for the LSC only. This result is consistent to the BET surface area analysis (Supplementary Figure 6), where N₂ adsorption–desorption isotherms of MoSe₂ shows type-IV characteristics with a hysteresis loop at relative pressures (P/P₀) from 0.45 to 1.0 and LSC only shows much less conspicuous type-IV character than that of MoSe₂ within similar P/P₀ range, which suggests that the mesoporous characteristics observed from LSC&MoSe₂ heterostructure mainly originated from the MoSe₂[23]. This mesoporous-pore-sized catalyst with large surface area is

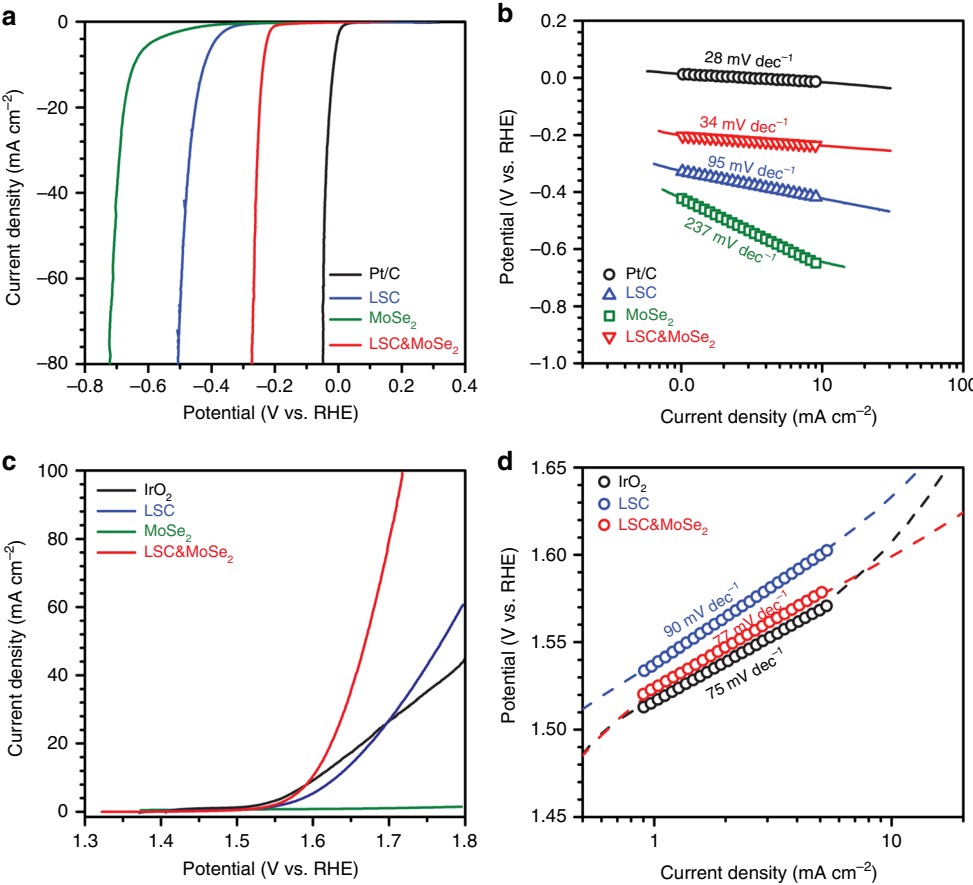

**Fig. 2** Half-cell-configured hydrogen evolution reaction and oxygen evolution reaction profiles. **a** HER polarization curves of various catalysts measured in $N_2$-saturated 1 M KOH. **b** Tafel analysis of the HER profiles. **c** OER polarization curves of various catalysts. **d** Tafel plots derived from the RDE profiles

expected to enhance water splitting efficiency by facilitating effective mass transfer within the catalyst[22].

**Electrochemical performance**. At first, the hydrogen evolution reaction (HER) activities of the proposed catalyst were investigated for various catalyst configurations by examining rotating disk electrode (RDE) polarization curves obtained under $N_2$-saturated 1 M KOH (Fig. 2a). The corresponding Tafel plots were derived from the obtained polarization curves (Fig. 2b). The potential was referenced to the reversible hydrogen electrode (RHE), as obtained from the calibration profile (Supplementary Figure 10). Pt/C exhibits excellent HER activity with an onset potential of 0.0 V vs. RHE and sharply increasing current density with a low Tafel slope value of 28 mV dec.$^{-1}$, which indicates the Volmer–Tafel reaction is its primary HER pathway[24]. In contrast, $MoSe_2$, known as an active HER catalyst under acidic media[25], shows rather inferior HER activity with an onset potential of $-0.42$ V and Tafel slope of 237 mV dec.$^{-1}$ in alkaline media. LSC presents moderate HER activity, as reported previously[12] with an onset potential of $-0.33$ V and Tafel slope of 95 mV dec.$^{-1}$, suggesting the Volmer–Heyrovsky reaction is the dominant HER pathway[26]. Interestingly, the addition of $MoSe_2$ to LSC (LSC&$MoSe_2$) facilitated the HER activities of LSC by positively shifting the onset potential to $-0.20$ V, a 0.13 V increase from that of LSC. This improvement is noteworthy in that the onset potential usually depends on the intrinsic property of the catalyst, and the combination of two low-performance materials, with onset potentials of $-0.33$ V (LSC) and $-0.42$ V ($MoSe_2$), respectively, led to an enhanced overall onset potential of $-0.20$ V

(LSC&$MoSe_2$), demonstrating the synergistic effect of heterostructure formation. The Tafel slope of the composite structure improved to 34 mV dec.$^{-1}$ from 95 mV dec.$^{-1}$ (LSC) and 237 mV dec.$^{-1}$ ($MoSe_2$), respectively, approaching that of Pt/C (28 mV dec.$^{-1}$), which suggests the Volmer–Tafel mechanism is the preferred pathway in the HER process of LSC&$MoSe_2$. This result is comparable to that of the high-performance emerging HER catalysts under identical electrolyte conditions (1 M KOH), and it shows the lowest Tafel slope among the survey groups (Supplementary Table 1), which indicates faster increment in current with small overpotential and lower activation barrier toward rapid reaction rate.

In addition, we investigated the oxygen evolution reaction (OER) activities of the prepared catalysts under $N_2$-saturated 1 M KOH. Figure 2c presents the OER polarization curves, and the corresponding Tafel slopes are derived, as shown in Fig. 2d. For the OER analysis, $IrO_2$, known as the state-of-the-art catalyst for OER, was adopted as a reference catalyst[27]. $IrO_2$ exhibits excellent OER performance with the onset potential of 1.51 V vs. RHE and Tafel slope of 75 mV dec.$^{-1}$. In the case of LSC, active OER performance was observed with the onset potential of 1.54 V and Tafel slope of 90 mV dec.$^{-1}$, but $MoSe_2$ exhibited an inactive OER process along with an increase in overpotential. Notably, LSC&$MoSe_2$ presented considerably improved OER activities compared to that of LSC with an onset potential of 1.52 V and sharply increasing current density with a low Tafel slope of 77 mV dec.$^{-1}$. In the same electrolyte tested for HER (1 M KOH), the OER performance achieved in this work is comparable to that of recently reported high-performance catalysts (Supplementary Table 2). These results demonstrate that the formation of

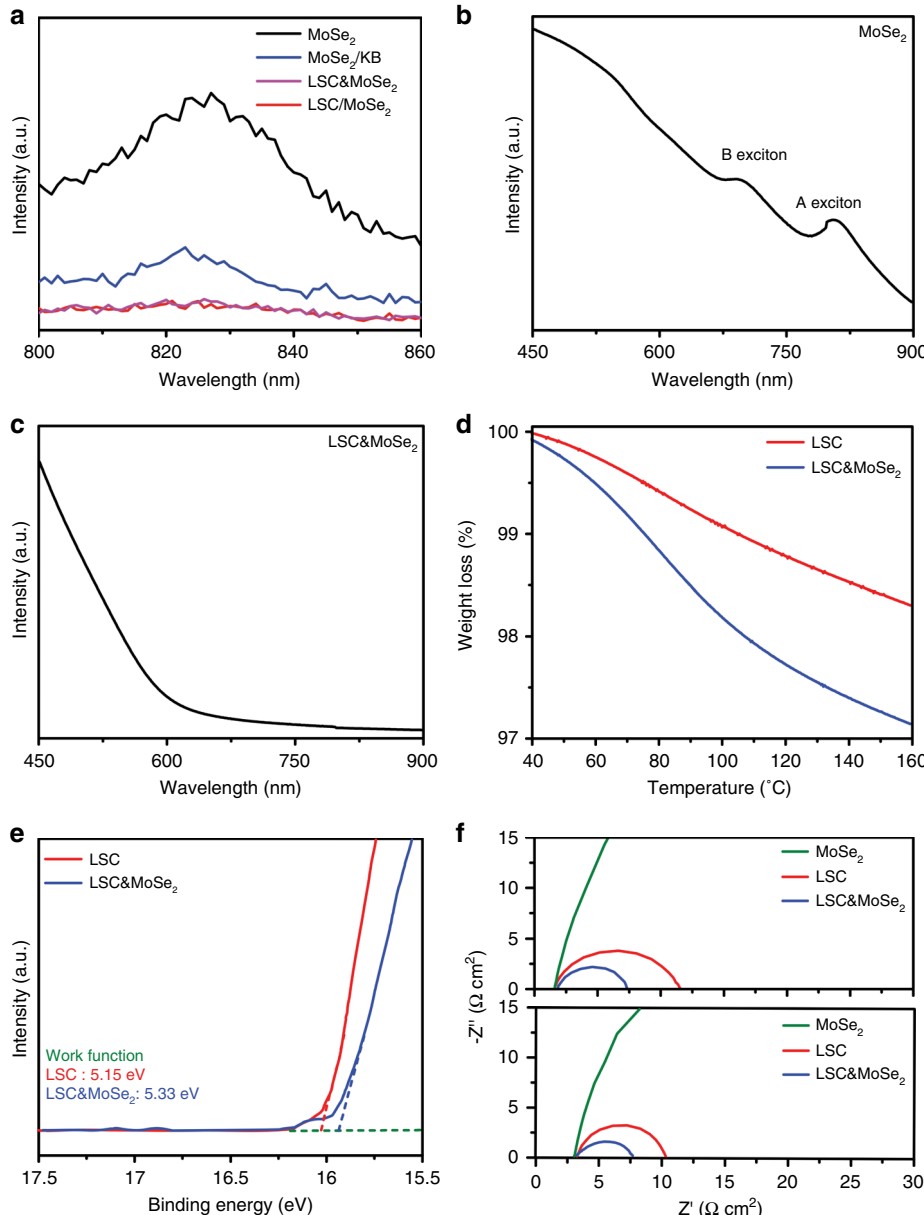

**Fig. 3** Analysis of physical properties of LSC&MoSe₂. Fluorescence emission spectra of **a** MoSe₂, MoSe₂/KB, LSC&MoSe₂, and LSC/MoSe₂ (The concentration of MoSe₂ in all samples is fixed as 0.33 mg/mL), illustrating the 2H-phase of the as-prepared MoSe₂ and electron transfers among MoSe₂, LSC, and KB. UV-vis-NIR spectra of **b** MoSe₂, indicating the presence of A and B excitons, which are attributed to direct excitonic transition of K points in the Brillouin zone, and **c** LSC&MoSe₂, illustrating its metallic feature. **d** TGA analysis of LSC and LSC&MoSe₂ pre-exposed to wet-air for 24 h. **e** UPS valence band spectra of LSC and LSC&MoSe₂. **f** Nyquist plots of HER (upper) and OER (lower) of MoSe₂, LSC, and LSC&MoSe₂

LSC&MoSe₂ leads to simultaneous enhancements in both the HER and OER activities, suggesting that LSC&MoSe₂ has potential as a highly efficient overall water splitting catalyst.

**Analysis of LSC&MoSe₂ properties**. To elucidate the origin of enhanced catalytic activity in LSC in the presence of additional MoSe₂, various analyses, including spectroscopic, chemical, and electrical characterizations, were performed to investigate the potential interaction between the two substances. Figure 3a shows the fluorescence emission spectra (FL) of MoSe₂, LSC only, and KB for various configurations. For the FL measurement, each sample (MoSe₂, LSC&MoSe₂, LSC only and MoSe₂ heterostructure (denoted LSC/MoSe₂), and MoSe₂ and KB heterostructure (denoted MoSe₂/KB) was prepared using the same

concentration of MoSe₂ (0.33 mg/mL). Detailed summary of abbreviations for each material system studied in this work is provided in Supplementary Table 3. Pristine 2H-phase MoSe₂ is semiconducting in nature[28], and LSC only and KB are highly conductive materials with metallic features[29]. A clear FL peak was observed for MoSe₂ at 825 nm, whereas substantial FL quenching occurred in the MoSe₂ composite containing LSC only and KB, that is, LSC&MoSe₂, LSC/MoSe₂, and MoSe₂/KB. In addition, the FL peak was not present in the case of KB, and only a weak FL spectrum was observed in the case of LSC only, as shown in Supplementary Figure 11. However, the FL peak was observed in the case of LSC. These results suggest that charge transfer occurs readily among MoSe₂, LSC only, and KB while that between LSC only and KB is limited. Moreover, optical absorption measurements were performed to further examine the effect of local phase

transition in TMDs to metallic-phase in LSC&MoSe$_2$. Figure 3b shows the UV-vis-NIR spectrum of MoSe$_2$, which highlights the semiconducting feature of the 2H-phase with two exciton peaks at ~700 and 800 nm[30]. In LSC&MoSe$_2$, such excitonic peaks disappeared, indicating that the phase transformed TMDs induced metal-like characteristics in LSC&MoSe$_2$[31,32] (Fig. 3c).

Because intrinsic catalyst activity is closely related to catalyst surface properties (e.g., electronic structure, vacancy, defect, etc.), the surface characteristics of LSC&MoSe$_2$ and LSC were investigated by X-ray photoelectron spectroscopy (XPS). We focused on the chemical state of Co 2p, as shown in Supplementary Figure 12, which has the dominant effect on the catalytic activity of LSC-based composite materials[22]. The Co 2p peak consists of two spin–orbit doublets, Co 2p$_{3/2}$ and Co 2p$_{1/2}$, with two satellites. The Co$^{3+}$ and Co$^{2+}$ of LSC are positioned at 779.6/794.2 eV and 781.5/797.0 eV, whereas those of LSC&MoSe$_2$ are located at 780.6/795.2 eV and 782.5/798.0 eV, respectively, indicating a binding energy upshift of 1 eV in LSC&MoSe$_2$ relative to LSC. This shift in the XPS peak was attributed to changes in the oxidation state of the LSC owing to the presence of additional MoSe$_2$. The Co 2p of LSC&MoSe$_2$ is to have higher oxidation states than the LSC owing to the electron transfer at the interface of LSC and MoSe$_2$ while maintaining the overall electroneutrality of LSC&MoSe$_2$[33,34]. The high electronegativity of Mo may have altered the electronic structure (σ*-orbital occupancy) of Co by decreasing the electron density in Co[35]. The σ*-orbital occupancy ($e_g$) of close to 1 of the perovskite oxides catalyst enhanced the binding of oxygen species at the B site, which contributed to improved OER performance[36]. The large ratio of high oxidation numbers in the Co species is related to the $e_g$ orbital filling of close to 1, which is a desirable feature for OER catalysts[22,37]. Therefore, the enhanced OER activity of LSC&MoSe$_2$ can be attributed to its higher Co$^{3+}$/Co$^{2+}$ ratio than that of LSC (Supplementary Table 4). The chemical state of O 1s in the catalyst, too, can influence its OER performance[38]. The O 1s spectrum consists of lattice oxygen (denoted A$_O$: lattice O$^{2-}$), highly oxidative oxygen (denoted B$_O$: O$_2^{2-}$/O$^-$), surface-active oxygen (denoted C$_O$: hydroxyl group (-OH)), and adsorbed water (denoted D$_O$: H$_2$O or CO$_3^{2-}$), which are located at 529.9, 531.3, 532.6, and 533.5 eV, respectively, in the case of LSC&MoSe$_2$ (Supplementary Figure 13). A high density of surface-active oxygen species on the catalyst surface, which is associated with oxygen vacancies and surface hydroxyl groups, is known to improve OER performance[6,38], and the relative ratio between lattice and surface-active oxygen can be used as an indicator for evaluating OER performance[39]. As shown in Supplementary Table 5, the higher C$_O$/A$_O$ ratio of LSC&MoSe$_2$ than that of LSC, i.e., metals with high oxidation states which enhances the adsorption ability for oxides in the LSC&MoSe$_2$ surface[38], indicates larger surface coverage of hydroxide species in LSC&MoSe$_2$, which can improve its intrinsic OER performance.

In addition, the surface adsorption capability of the catalyst was analyzed by conducting thermal gravimetric analysis (TGA) to elucidate the different electrocatalytic performances of LSC&MoSe$_2$ and LSC. LSC&MoSe$_2$ and LSC were exposed to wet-air conditions before the TGA measurement to facilitate the adsorption of water, H, and OH groups from the atmosphere. As shown in Fig. 3d, the relative weight loss (%) was larger in the case of LSC&MoSe$_2$ than that in the case of LSC, implying that LSC&MoSe$_2$ possesses a higher surface adsorption capability, which can improve its electrocatalytic performance[6].

Ultraviolet photoelectron spectroscopy (UPS) analysis was conducted to investigate the charge transfer between LSC and LSC&MoSe$_2$ to elucidate the improved catalytic performance of LSC&MoSe$_2$. The increase in the work function of the catalyst increases proton concentration in the electronic double layer of

the catalyst, which facilitates easy progress of the Volmer step (H$^+$+e$^-$=H$_{ad}$), which initiates the HER process, with the least overpotential[40,41]. Moreover, it increases the preexponential factor and the rate constant. This reduces the metal-hydrogen (M–H) bond strength, which helps improve the exchange current density[42,43]. Figure 3e shows the secondary electron cutoff energies of LSC and LSC&MoSe$_2$, from which the work function values are derived. The addition of MoSe$_2$ to LSC increased the work function value from 5.15 of LSC to 5.33 eV of LSC&MoSe$_2$. The UPS result indicated that the incorporation of Mo with higher electronegativity into Co, the active site in perovskite oxide catalysts, led to an increase in the work function value of LSC&MoSe$_2$, which improved its HER performance by increasing the exchange current density of LSC&MoSe$_2$. This increase in the work function can induce a decrease in the adsorption bond strength between the catalyst and hydrogen, which brings the Gibbs free energy closer to the thermoneutral point of catalyst-H*, leading to improved catalytic activity of HER.

The improved catalytic performance of LSC&MoSe$_2$ was examined by means of electrochemical impedance spectroscopy (EIS) analysis. Figure 3f shows the Nyquist plots of both the HER and OER of MoSe$_2$, LSC, and LSC&MoSe$_2$, from which the charge transfer resistance ($R_{ct}$) of the catalyst between the electrode and electrolyte was obtained. The $R_{ct}$ of MoSe$_2$ is 1500 Ω cm$^2$ in both HER and OER, which is highest charge transfer resistance compared to the others. In contrast, the $R_{ct}$ of LSC&MoSe$_2$ is 5.71 and 4.36 Ω cm$^2$ for HER and OER, respectively, while that of LSC is 10.16 and 6.93 Ω cm$^2$, respectively. This result suggests that rapid electron transport is feasible in LSC&MoSe$_2$ owing to the additional MoSe$_2$, which can improve conductivity of the catalyst. In brief, the improved electrocatalytic performance of LSC&MoSe$_2$ in HER and OER can be attributed to various combinatorial effects of enhanced electrical and chemical properties in the catalyst that benefit the HER and OER aspects simultaneously.

**Synergetic effect for improved electrochemical performance.** The morphology of the crystal structure of LSC&MoSe$_2$ was investigated to further elucidate the dramatic improvement in the electrochemical performance of LSC&MoSe$_2$ compared to other catalyst configurations such as MoSe$_2$ and LSC. Figure 4a shows a HR-TEM image of LSC&MoSe$_2$. Interestingly, after formation of the heterostructure of LSC with 2H phase MoSe$_2$, the lattice structures of 1T-MoSe$_2$ (metallic, octahedral) and 2H-MoSe$_2$ (semiconducting, trigonal prismatic) were observed simultaneously. Each phase of the MoSe$_2$ lattice structure can be observed clearly in the zoomed-in image presented in Fig. 4b, c along with the corresponding schematic diagram of the lattice structures. The 2H-MoSe$_2$ region showed the typical hexagonal Mo–Se atomic arrangement with a 0.283 nm interlayer distance, corresponding to the (100) plane of MoSe$_2$. In contrast, 1T-MoSe$_2$ exhibited a distinctively different crystal structure that was attributed to electronic structure rearrangement[44]. The lattice spacings of Mo–Mo and Se–Se were 0.563 and 0.324 nm, respectively, confirming the presence of 1T-MoSe$_2$[19,44]. The coexistence of 1T- and 2H-phase of MoSe$_2$ in LSC&MoSe$_2$ was verified by XPS analysis shown in Supplementary Figure 14 (1T-phase: Mo 3d$_{5/2}$ for 228.3 eV, Mo 3d$_{3/2}$ for 231.4 eV, Se 3d$_{5/2}$ for 53.7 eV, and Se 3d$_{3/2}$ for 54.7 eV, 2H-phase: Mo 3d$_{5/2}$ for 229.0 eV, Mo 3d$_{3/2}$ for 232.6 eV, Se 3d$_{5/2}$ for 54.3 eV, and Se 3d$_{3/2}$ for 55.6 eV)[19,45,46]. The relative contents of 1T- and 2H-phase MoSe$_2$ from the as-prepared LSC&MoSe$_2$ are summarized in Supplementary Table 6. To verify whether the ball milling process has any effect on the phase transition of MoSe$_2$, MoSe$_2$/KB was synthesized under the same ball-mill process as LSC&MoSe$_2$ and

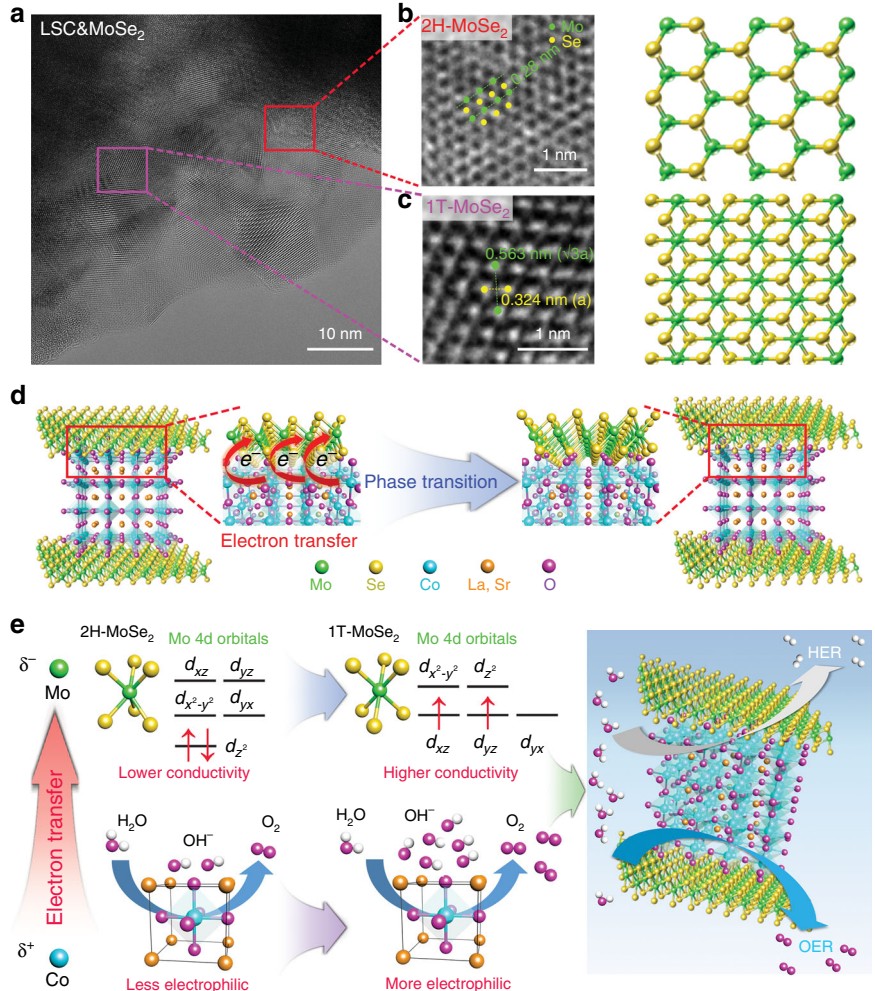

**Fig. 4** Proposed mechanism describing potential interaction between MoSe₂ and LSC. **a** HR-TEM image of LSC&MoSe₂, indicating the presence of both 2H- and 1T-Phase MoSe₂. **b** Enlarged region of 2H-MoSe₂ shown with schematic lattice structure, illustrating the hexagonal crystal structure with Mo–Mo inter-atomic distance of 0.28 nm. **c** Enlarged region of 1T-MoSe₂ shown with schematic lattice structure, indicating the Mo–Mo (0.563 nm) and Se–Se (0.324 nm) inter-atomic distances. **d** Schematic description of local phase transition in MoSe₂ via electron transfer from Co to Mo. **e** Schematic diagram of proposed charge transfer processes between MoSe₂ and LSC

investigated through XPS and HR-TEM analysis. As shown in Supplementary Figs. 15–16, no signature of phase transition is observed and only 2H-phase MoSe₂ is present in ball-milled MoSe₂/KB. These results suggest that the local phase transition in MoSe₂ from the 2H-phase to the 1T-phase occurred in-situ during formation of the heterostructure between LSC and 2H-phase MoSe₂. 1T-MoSe₂ is typically synthesized via alkali metal treatment of 2H-MoSe₂ in inert environment[20,21]. Surprisingly, in the present work, the presence of 1T-MoSe₂ was observed locally in LSC&MoSe₂. We hypothesize that this in-situ phase transition in MoSe₂ is induced by electron transfer from Co to Mo[47], as illustrated schematically in Fig. 4d. The electron transfer between Mo and Co is expected to improve the electrochemical catalytic activity of MoSe₂ and LSC, as described in Fig. 4e. In the case of MoSe₂, additional electrons alter the Mo 4$d$-orbital configuration from the occupied $4d_z^2$ level to incompletely filled $d_{xz}$, $d_{yz}$, $d_{yx}$ orbital, which induces the phase transition[44,48]. As a result, the intrinsic conductivity of MoSe₂ improves, which improves the electrochemical performance of the catalyst. This electron transfer is beneficial for the catalytic activity of LSC to achieve efficient water splitting. The Co in LSC becomes more electrophilic after the electron transfer, which alters the electronic structure of Co

by upshifting the d-band center[35]. Such an up-shift of d-band center strengthens the adsorption capability of OH⁻[49]. Therefore, the enhanced OH⁻ affinity and electrophilicity in Co, that is, increase in Co–O and Co–OH in LSC, can ultimately improve the overall water electrolysis performance. These charge transport phenomena bring synergistic effects to enhance both the HER and OER performance from the heterogenous composite structure of LSC&MoSe₂ in addition to the performance improvement factors mentioned in Analysis of LSC&MoSe₂ properties section.

**Theoretical elucidation of charge transfer in LSC&MoSe₂.** To theoretically demonstrate the charge transfer phenomenon in LSC&MoSe₂ heterostructure, we performed the density functional theory (DFT) calculations (see calculation details in the Experimental section). For this purpose, we first examined the relative stability of two plausible terminations of LSC (001) surface, corresponding to CoO₂-termination (denoted CoO₂-t.) and (La, Sr)O-termination (denoted (La,Sr)O-t.), by surface energy (γ) calculations (see Supplementary Fig. 17 and Surface energy calculations for details in the Experimental section). As a result, (La, Sr)O-t. was predicted to be predominantly exposed on the LSC nanoparticle surface due to its lower surface energy (i.e., γ = 0.58

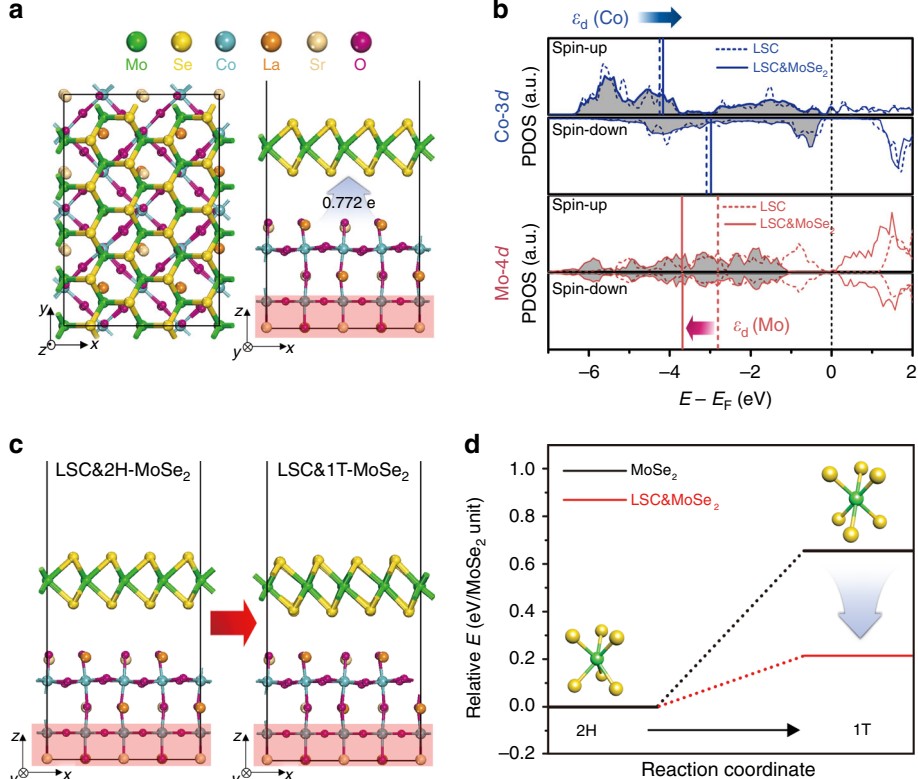

**Fig. 5** DFT calculations for charge transfer phenomenon in LSC&MoSe$_2$. **a** Optimized structure of LSC&2H-MoSe$_2$ heterostructure. The red shaded box represents the fixed atoms in the two bottommost layers. The blue arrow indicates the direction of charge transfer from LSC to MoSe$_2$. **b** Spin-up and spin-down projected density of states (PDOS) on the following species: Co-3$d$ in LSC (blue dotted line), Co-3$d$ in LSC&2H-MoSe$_2$ (blue solid line), Mo-4$d$ in 2H-MoSe$_2$ (red dotted line), and Mo-4$d$ in LSC&2H-MoSe$_2$ (red solid line), respectively. The vertical lines represent the position of d-band center ($\varepsilon_d$) for each species. **c** Schematic illustration of phase transition from LSC&2H-MoSe$_2$ to LSC&1T-MoSe$_2$ heterostructure. **d** Relative energies of 2H- and 1T-phase of MoSe$_2$ monolayer (black line) and LSC&MoSe$_2$ heterostructure (red line). The inset models represent the trigonal prismatic (2H) and octahedral (1T) geometry

J/m$^2$ for (La,Sr)O-t. and $\gamma = 0.71$ J/m$^2$ for CoO$_2$-t., respectively). Accordingly, we built the LSC&2H-MoSe$_2$ heterostructure based on this termination (Fig. 5a). By the Bader charge analysis[50], it was revealed that the charge transfer occurred from LSC into MoSe$_2$ with an amount of 0.772 $e$, whose direction was in accordance with our previous expectation.

To better understand these electron movements, the projected density of states (PDOS) on each d-orbital of Co and Mo in LSC&2H-MoSe$_2$ was further investigated (Fig. 5b). The asymmetric configuration of spin-up and spin-down DOS for LSC indicated the ferromagnetic nature, whereas the symmetric configuration for MoSe$_2$ represented its non-magnetic character[51,52]. Driven by the electron transfer across the interface, the d-band center ($\varepsilon_d$) of Co in LSC shifted upward about 0.1 eV with respect to the Fermi level when coupled with 2H-MoSe$_2$, indicating that LSC became more electrophilic. In a previous study, the enhanced electrophilicity of LSC was reported to strengthen the OH$^-$ affinity, which has a beneficial effect to improve the OER catalytic performance[35]. Meanwhile, the $\varepsilon_d$ of Mo in 2H-MoSe$_2$ was significantly downshifted by ~0.9 eV with respect to the Fermi level when combined with LSC, implying that additional electrons transferred into 2H-MoSe$_2$ side. These extra d-electrons are expected to promote the 2H- to 1T-phase transition by destabilizing the 2H-MoSe$_2$ phase, as similarly observed in the MoS$_2$ system by Gao et al.'s work[53]. To further clarify this, we compared the relative stability for both phases of MoSe$_2$ in the presence or absence of LSC (Fig. 5c, d and Supplementary Figure 18). The endothermicity of 1T-MoSe$_2$

(relative to its 2H-phase) clearly decreased from 0.65 to 0.21 eV per MoSe$_2$ unit when LSC was combined, indicating that the phase transition from 2H- to 1T-phase could occur more easily. Further, the local DOS of MoSe$_2$ in the LSC&MoSe$_2$ heterostructure showed that the intrinsic conductivity can be significantly enhanced by closing the energy gap due to the transition from semiconducting 2H- to metallic 1T-phase, which can improve the HER catalytic performance (Supplementary Figure 19)[54]. These results theoretically elucidate the beneficial effects of charge transfer phenomenon in LSC&MoSe$_2$ for both HER and OER performance.

**Overall water splitting of LSC&MoSe$_2$ || LSC&MoSe$_2$.** An overall water splitting test was conducted to evaluate the performance and stability of the developed LSC&MoSe$_2$ electrocatalyst. By adopting the prepared LSC&MoSe$_2$ catalyst, we evaluated the overall water splitting performance in deaerated alkaline media containing N$_2$-saturated 1 M KOH. The water splitting test was conducted with the three-electrode configuration by using an Ag/AgCl reference electrode to separate the overall cell reaction into the cathodic and anodic reactions of HER and OER, respectively. The electrochemical overall water electrolysis actively generated hydrogen (cathode) and oxygen (anode) gases, as shown in Fig. 6a. Figure 6b shows the HER and OER polarization curves, denoted $E_{cathode}$ and $E_{anode}$, respectively, measured during the water electrolysis reaction, and the obtained cell potentials ($E_{cell} = E_{anode} - E_{cathode}$) of Pt/C || IrO$_2$ (Pt/C for

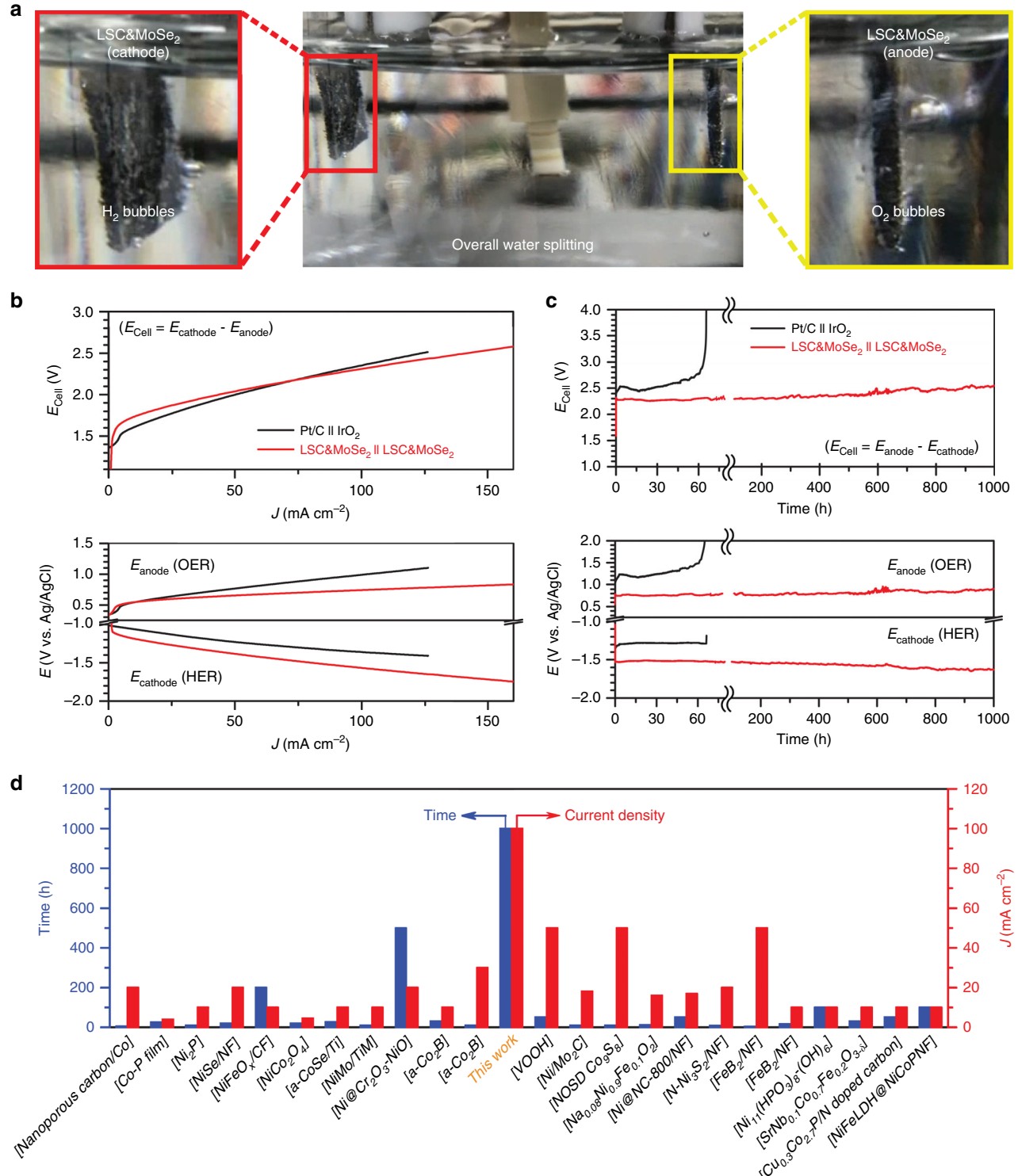

**Fig. 6** Overall water splitting performance. **a** Digital image of the water splitting test conducted with three-electrode configuration by using Ag/AgCl reference electrode. **b** Polarization I–V profiles of various catalysts measured in deaerated 1 M KOH. **c** Chronopotentiometric stability profiles measured at 100 mA cm$^{-2}$ for 1000 h. **d** Comparison of the overall water electrolysis stability of various catalysts reported in the literature

cathode and IrO$_2$ for anode) and LSC&MoSe$_2$ ‖ LSC&MoSe$_2$ (LSC&MoSe$_2$ for both cathode and anode). Also, the overall water electrolysis performance of bare Ni foam is investigated and compared to the published works (Supplementary Figure 20). The reference state-of-the-art catalyst pair, Pt/C ‖ IrO$_2$, performed well in both HER and OER, as observed previously in the half-cell-configured RDE polarization profiles. In the case of Pt/C ‖

IrO$_2$, overall water splitting was observed from near 1.38 V. LSC&MoSe$_2$ ‖ LSC&MoSe$_2$ exhibited slightly lower HER performance than Pt/C, but the OER performance was higher than that of IrO$_2$. Although overall water splitting was observed at a slightly higher value of 1.52 V, current density increased sharply as the potential increases. The determination of the onset potential values for each catalyst can be seen in Supplementary

Figure 21. Notably, LSC&MoSe$_2$ || LSC&MoSe$_2$ outperformed Pt/C || IrO$_2$ after 2.16 V owing to its concurrently effective HER and OER kinetics. To investigate the electrochemical stability and durability of the LSC&MoSe$_2$ catalyst, we measured its chronopotentiometric profiles at a current density of 100 mA cm$^{-2}$, as shown in Fig. 6c. For Pt/C || IrO$_2$, gradual degradation was initiated from 16 h and drastic degradation occurred after 60 h, leading to cell failure. In contrast, LSC&MoSe$_2$ || LSC&MoSe$_2$ exhibited remarkably stable operation over 1000 h with negligible fluctuation in performance, demonstrating its exceptionally high electrochemical durability in HER and OER, even at a high current density of 100 mA cm$^{-2}$. Furthermore, the energy efficiency of overall water electrolysis at the current density of 100 mA cm$^{-2}$ was calculated to be 63.9% as shown in Supplementary Note 1. Considering the energy efficiency for conventional alkaline electrolysis is <70% with the use of noble metal-based electrode[55], the proposed LSC&MoSe$_2$ catalyst can be considered as highly efficient for the water electrolysis. Supplementary Figure 22 shows SEM images of the LSC&MoSe$_2$ electrode after 1000 h of the overall water electrolysis test; clogging or electrochemical/physical damage was not observed on the catalyst after the test. The chemical state of LSC&MoSe$_2$ after the stability test was further examined via XPS analysis. As shown from Supplementary Figure 23 and Tables 7 and 8, the ratio of Co$^{3+}$/Co$^{2+}$ and surface-active oxygen/lattice oxygen showed almost negligible changes even after the 1000 h of overall water splitting measurement, indicating the excellent stability of the proposed LSC&MoSe$_2$ as the electrocatalyst. Progress of the overall water splitting test with H$_2$ and O$_2$ bubble generation from the cathode and anode, respectively, is shown in Supplementary Movie 1. The overall water electrolysis stability achieved in this work was compared with that of the representative overall water splitting catalysts reported to date, as shown in Fig. 6d and summarized in Supplementary Table 9. Despite the harsh test environment (highest current density) in the present study, the developed catalyst exhibited overwhelmingly high overall water electrolysis stability compared to that achieved in other works.

## Discussion

We demonstrated that a composite of perovskite oxides and MoSe$_2$ exhibits excellent electrocatalytic performance and stability in overall water electrolysis. Formation of LSC&MoSe$_2$ led to several synergetic effects, including increased specific surface area for both the HER and OER, enhanced surface adsorption capability, favorable kinetics for the Volmer–Tafel pathway, and decreased charge transfer resistance, which contributed to its improved electrochemical catalytic performance. Interestingly, electron transfer between Co and Mo induced a local phase transition in MoSe$_2$ and increased the amounts of Co–O and Co–OH in La$_{0.5}$Sr$_{0.5}$CoO$_{3-\delta}$, leading to improved intrinsic and extrinsic catalytic activities of LSC&MoSe$_2$. In the overall water splitting electrolysis test, LSC&MoSe$_2$ || LSC&MoSe$_2$ showed comparable operating voltages to that of Pt/C || IrO$_2$ along with remarkable long-term stability over 1000 h. The results obtained in the present study suggest that perovskite oxides and TMDs based heterostructures can be promising bifunctional water splitting catalysts and can serve as potential alternatives to precious metal-based electrochemical catalysts.

## Methods

**Synthesis of MoSe$_2$.** For the preparation of MoSe$_2$ flakes, 500 mg of bulk MoSe$_2$ powder (~325 mesh powder, purity >99.9%, Alfa Aesar) was dispersed in isopropyl alcohol (IPA) (350 mL) and deionized water (150 mL), and subsequently exfoliated via tip-sonication (Sonic & materials, VC 505) during 5 h. The resulting suspension was centrifuged and the supernatant was collected, which was dried in vacuum oven at 100 ℃ for 12 h.

**Synthesis of LSC.** La$_{0.5}$Sr$_{0.5}$CoO$_{3-\delta}$ (LSC) perovskite oxides was synthesized by typical sol-gel process. Stoichiometric amount of metal nitrate precursors and citric acid were dissolved in deionized water to form an aqueous solution. After the nitrate precursors were completely dissolved, an appropriate amount of polyethylene glycol ($M_w$ ~400) was added. All chemical reagents were purchased from Sigma-Aldrich and used as-received without further purification. After a viscous resin was formed, the solution was heated at 300 ℃. Then, the resulting powder was pre-calcined at 600 ℃ for 4 h and calcined at 950 ℃ for 4 h.

**Synthesis of catalysts.** To find the optimum ratio for LSC&MoSe$_2$ catalyst, LSC and MoSe$_2$ were high-energy milled with 10 wt.% of Ketjen black EC-600JD (KB) by a planetary ball mill system (PM-200, Retsch, Germany). For the milling process, the following weight ratios were examined with LSC:MoSe$_2$:KB of 9:0:1, 8:1:1, 7:2:1, 6:3:1, 5:4:1, and 0:9:1. Total weight of catalyst was maintained as 500 mg and each catalyst was dispersed in ethanol and ball-milled using Zr-balls at 400 rpm for 2 h. Thus, for synthesis of LSC:MoSe$_2$:KB = 6:3:1, 300 mg of LSC, 150 mg of MoSe$_2$, and 50 mg of KB was used. In case of LSC/KB, MoSe$_2$/KB, and LSC/MoSe$_2$ synthesis, 450 mg of LSC/50 mg of KB, 450 mg of MoSe$_2$/50 mg of KB, and 300 mg of LSC/150 mg of MoSe$_2$ was used, respectively. Then, the powder was collected by drying the solvent in a 70 ℃ of oven for further analyses.

**Material characterizations.** Structural and morphological characterizations were conducted via HR-TEM (JEM-2100F, JEOL) with a probe forming (STEM) Cs (spherical aberration) corrector at 200 kV and XRD (D8 Advance, Bruker) at a scan rate of 1° min$^{-1}$, respectively. Fluorescence emission and UV-vis NIR spectra were obtained by Fluorometer (Cary Eclipse, Varian) and UV-vis NIR spectrophotometer (Cary 5000, Agilent). BET surface area and pore size of LSC&MoSe$_2$ and LSC were investigated via Physiorption Analyzer (ASAP 2420, Micromeritics Instruments) with N$_2$ desorption/adsorption. Surface adsorption capability was analyzed by TGA (Q500, TA). LSC&MoSe$_2$ and LSC used in TGA analysis were exposed to wet-air for 24 h to absorb the moisture. UPS and XPS measurements were conducted using He I (21.2 eV) discharge lamp and monochromatic Al-Kα radiation source, respectively, (ESCALAB 250Xi, Thermo Fisher Scientific) under ultra-high vacuum condition (<10$^{-10}$ Torr). SEM images of electrodes before and after overall water electrolysis were obtained via Nano 230 FE-SEM (Nova Nano SEM, FEI).

**Half-cell analysis.** Half-cell measurements were conducted in typical three-electrode configuration using a Pt wire and Ag/AgCl electrode (saturated KCl filled) as the counter electrode and reference electrode, respectively. The rotating disk electrode (RDE) tests were carried out by using catalysts of 20 wt% Pt/C (Alfa Aesar), IrO$_2$ (Alfa Aesar), LSC, MoSe$_2$, and LSC&MoSe$_2$ on RRDE-3A (ALS). Each catalyst was prepared in the form of an ink by dispersing 10 mg of the catalyst in 1 mL of a binder solution (45:45:10 = ethanol:isopropyl alcohol:5 wt% Nafion solution (Sigma-Aldrich), volumetric ratio) followed by a bath sonication process. Then, HER and OER profiles were investigated in N$_2$-saturated 1 M KOH aqueous solution at a scan rate of 10 mV s$^{-1}$ by drop-coating 5 μL of each catalyst ink onto glassy carbon disk electrode, where the area is 0.13 cm$^2$. A calibration in reversible hydrogen electrode (RHE) was experimentally determined at a scan rate of 1 mV s$^{-1}$ in H$_2$-saturated 1 M KOH, where platinum wire was used as the working and counter electrode and Ag/AgCl as the reference electrode. All half-cell profiles were iR compensated by measuring the resistance of solution (1 M KOH). All electrochemical tests were carried out using Biologic VMP3.

**Overall water splitting test.** The overall water electrolysis tests were proceeded in three-electrode configuration using Ag/AgCl reference electrode. The cathode and anode were prepared by electro-spraying the prepared catalyst ink onto a Ni mesh current collector with a catalyst loading density of 1 mg cm$^{-2}$. The measurements were proceeded in deaerated 1 M KOH aqueous solution. The current density was normalized with the geometric area of the catalyst. All electrochemical tests were conducted using Biologic VMP3.

**Calculation details.** Spin-polarized DFT calculations were carried out using the Vienna ab initio simulation package (VASP)[56] within the projector-augmented wave (PAW) method[57]. The electron exchange-correlation energy was treated by the generalized gradient approximation (GGA) with Perdew-Burke-Ernzerhof (PBE) functional[58]. The DFT + U method within Dudarev's approach[59] was also adopted with $U = 4.3$ eV and $J = 1.0$ eV for Co-3d and $U = 4.0$ eV for Mo-4d, respectively. The energy cutoff for the plane-wave basis set was set as 400 eV and PAW data sets were used with following valence electronic states: 5 $s^2$, 5$p^6$, 5$d^1$, 6 $s^2$ for La; 4 $s^2$, 4$p^6$, 5 $s^2$ for Sr; 3$d^8$, 4$d^1$ for Co; 2 $s^2$, 2$p^4$ for O;4 $s^2$, 4$p^6$, 4$d^6$ for Mo; and 4 $s^2$, 4$p^4$ for Se, respectively. Geometry optimizations were performed using conjugated gradient (CG) method until the net force on each atom reached <0.02 eV Å$^{-1}$ and the total energy was changed within 10$^{-6}$ eV per atom. Dipole slab corrections were also applied to all slab model calculations. The Monkhorst-Pack scheme of $k$-point grid[60] was set to $\Gamma$ points for geometry optimization, and $3 \times 2 \times 1$ $k$-points in the Brillouin zone for DOS analysis, respectively. Bader analysis[50] was used to calculate the atomic charges.

**Model systems for calculation**. To construct the LSC&MoSe$_2$ heterostructure, each slab model for LSC surface and MoSe$_2$ layer was separately modeled in advance. First, the unit cell structure of LaCoO$_3$ (LCO) was fully relaxed by optimizing both atomic positions and lattice parameters, which were well matched with experimentally reported values[61]. Subsequently, the LCO bulk structure was cleaved along the (001) plane with two plausible terminations (i.e., CoO$_2$ termination and LaO termination). Note that we considered symmetric slab models of LCO (001), which consisted of 5 atomic layers, to remove the fictious dipole moment in the slab. The bottommost two layers were fixed to their bulk positions. Next, a $2\sqrt{2} \times 3\sqrt{2} \times 1$ supercell structure was created, and subsequently half of La atoms were replaced into Sr atoms to attain the stoichiometry of La$_{0.5}$Sr$_{0.5}$CoO$_3$ system (i.e., $a = 10.91$ Å, $b = 16.37$ Å, $c = 25.00$ Å, 156 atoms). For MoSe$_2$ slab model, a $2 \times 5 \times 1$ supercell structure of orthorhombic unit cell for 2H-MoSe$_2$ was created (i.e., $a = 11.49$ Å, $b = 16.78$ Å, $c = 25.00$ Å, 60 atoms). Finally, the LSC&MoSe$_2$ heterostructure was built by combining the LSC and MoSe$_2$ slabs with minimized lattice mismatch <3% (i.e., $a = 11.20$ Å, $b = 16.57$ Å, $c = 35.00$ Å, 216 atoms). The vacuum was sufficiently applied to avoid the self-interaction in $z$-direction.

**Surface energy calculations**. To evaluate the relative stability of complementary terminations in the LSC (001) surface, we calculated the surface energy ($\gamma$) by the sum of cleavage energy ($\gamma_u$) and relaxation energy ($\gamma_r$), which was previously reported by Heifets et al[62].,

$$\gamma = \gamma_u + \gamma_r \tag{1}$$

The cleavage energy can be obtained as follows,

$$\gamma_u = (E^u_{slab}(CoO_2 - t.) + E^u_{slab}((La,Sr)O - t.) - NE_{bulk})/4A \tag{2}$$

where $E^u_{slab}(CoO_2 - t.)$ and $E^u_{slab}((La,Sr)O - t.)$ are unrelaxed CoO$_2$- and (La,Sr) O-terminated slab energies, $E_{bulk}$ is the total energy of bulk unit cell, $N$ is the formula unit of slab models, and $A$ is the surface area. The factor of four in the denominator arises from the four cleaved surfaces of two terminations. Next, we can calculate the relaxation energies for each CoO$_2$-t. and (La,Sr)O-t. as follows,

$$\gamma_r = (E^r_{slab}(X) - E^u_{slab}(X))/2A(X = CoO_2 - t. or (La,Sr)O - t.) \tag{3}$$

where $E^r_{slab}(X)$ is a slab energy after relaxation.

## Data availability

The data measured, simulated, and analyzed in this study are available from the corresponding author on reasonable request.

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

## Acknowledgements

This work was supported by Basic Science Research Program through the National Research Foundation of Korea (NRF) funded by the Ministry of Education (2015R1D1A1A0105791) and the Research Project Funded by U-K Brand (1.190004.01) of UNIST (Ulsan National Institute of Science & Technology). This research was also supported by the Mid-Career Researcher Program (NRF-2018R1A2A1A05077532) through the National Research Foundation of Korea, funded by the Ministry of Science, ICT and Future Planning. S.K.K. acknowledges the financial support by NRF (NRF-2014R1A5A1009799) and computational resources from UNIST-HPC.

## Author contributions

N.K.O. and C.K. carried out most of the experimental work and wrote the manuscript. J.L. performed XPS measurements. O.K. performed TEM analysis. Y.C. contributed to designing of schematics. G.Y.J. and H.Y.L. performed the DFT calculation. S.K.K. directed the theoretical work. G.K. provided constructive comments and conceived the project. H.P. conceived the project and directed the overall experimental work and manuscript writing. All the authors contributed to the discussions and analysis of the results regarding the manuscript.

## Additional information

**Competing interests:** The authors declare no competing interests.

**Journal Peer Review Information:** *Nature Communications* thanks Xihong Lu and the other anonymous reviewer(s) for their contribution to the peer review of this work. Peer reviewer reports are available.

