## [Peer Review File · Nature Communications]

Reviewers' comments:

Reviewer #1 (Remarks to the Author):

The manuscript entitled "In-situ Local Phase Transition of MoSe₂ in La_{0.5}Sr_{0.5}Co_{3-δ} Heterostructure and Excellent Overall Water Electrolysis over 1000 hours" addresses the design of a hetero-structure consisting of two materials, LSC and MoSe₂ and improved electrochemical oxygen (OER) and hydrogen evolution (HER) activity in the composite than the individual materials (LSC and MoSe₂). Here they have explained that on combining LSC and MoSe₂, Co donates electron to more electronegative Mo and causes the phase transition of MoSe₂ from semiconducting 2H to metallic 1T phase which improves HER activity and due to the donation of electron from Co, generation of high ratio of Co³⁺ to Co²⁺ facilitates the OER activity. Although authors have performed long term stability (1000 h), but the activity is still poor in comparison to recently reported many other active catalyst systems in terms of cell voltage. Both of the catalysts are already reported and the approach of making composite is also trivial. This work is not suitable for Nature Communications. However there are issues the authors should address before submitting elsewhere.

Comments:

1. The essence of this work is the phase transition in MoSe₂ that helps the electrocatalytic activity. However this concept is pretty well known and reported earlier. See: Chem. Commun., 2015, 51, 8450-8453 among many.
2. The relative abundance of 1T and 2H phases needs to be quantified. As to co-relate with the proposed structural schematics, the composite should be studied better through TEM.
3. The authors did not optimize the weight ratio of MoSe₂ and LSC, they have taken only one composition of MoSe₂ and LSC.
4. The authors have done OER measurements by purging N₂ in KOH, but according to Nernst equation, the OER measurements should be done followed by purging O₂ to maintain proper equilibrium between O₂ (product) and OH⁻ (reactant) [Ref: ACS Catal. 2018, 8, 1913–1920]
5. When the measurements are performed in GC, the activities for both OER and HER for the mixed LSC and MoSe₂ are poor. They have obtained a cell voltage of 1.65 V only after using Ni foam which itself is an efficient current collector, whereas there are several reports obtaining lesser cell voltage using Ni foam only. [Ref: Adv. Energy Mater. 2017, 1700513; Adv. Funct. Mater. 2017, 27, 1701008]
6. This study demands suitable computational support to prove the charge transfer phenomenon, which the authors have claimed playing a major role here.
7. The authors claimed that they have synthesized MoSe₂ flakes by exfoliating bulk MoSe₂, but the morphology is not clear in Fig. 1(a) and even after exfoliation there is an aggregated morphology.
8. In ADF-EDS mapping for Mo and Se (Fig. 1b), authors claimed uniform distribution of different elements which is not evident from the distribution presented here.
9. PXRD analysis: For LSC presence of small peaks can be observed around 2θ 25 and 36. Peak at around 36 may attribute to the cobalt oxide impurity. Again, the amount of LSC used is more as compared to its counterpart, MoSe₂. But from XRD plot this is not reflected.
10. Surface area analysis: Authors reported that surface in case of composite material is greatly

enhanced in comparison to LSC only. Which one is contributing for this enhancement, LSC or MoSe₂? Generally the flake like MoSe₂ separately has a high BET surface area due to its two dimensional morphology, but the authors did not mention the individual surface area of MoSe₂. They explained that increase in surface area in the composite enhances the exposure of active sites, but according to their explanation, the active site for OER and HER is different. For OER, active site is mainly Co present in LSC, and MoSe₂ is mainly responsible for HER activity while the charge transfer between these two is responsible for the enhanced electrochemical activity in the composite. If the active sites for OER and HER are coming from two different materials in the composite, then just by making the composite cannot increase the total number of active sites for both OER and HER.

- Moreover for determining active sites, electrochemically active surface area determination is more important rather than BET surface area.

- In supplementary fig. 1, the authors showed that mesopores are present in the surface of the heterostructure, but they did not explain the origin of mesopores, whether it is for LSC or MoSe₂.

11. Electronic measurements: Fluorescence intensity dependent proof of the presence of MoSe₂ in the material is misleading when the concentration of MoSe₂ is not the same in all cases. UV-vis spectrum in Fig. 3c corresponds to LCO & MoSe₂. From there how the authors could conclude the charge transfer phenomenon?

12. The fitting of the XPS spectra are improper, e.g. Fig S5 for LSC & MoSe₂. There are no details of fitting.

13. The authors have explained that higher ratio of Co³⁺ and Co²⁺ is responsible for improved OER performance but this contradicts many previous reports which infer that higher ratio of Co²⁺ and Co³⁺ are better for OER because more the number of low coordinated unsaturated metal sites, more it can avail OH⁻ from electrolyte. [Ref: Adv. Mater. 2017, 29, 1606793, ACS Appl. Mater. Interfaces, 2016, 8 (50), 34474–34481]

Moreover in the XPS spectrum of O 1s in mixed LSC and MoSe₂, surface-active oxygen (C0) actually denotes oxygen vacancy according to reference no. 46. So if there is higher ratio of C0 and A0 in the mixed catalyst, then there should be higher ratio of Co²⁺ to Co³⁺. Because for the sake of electro neutrality of the material and to satisfy O vacancy, lower oxidation states of metal should be more. So these two trends contradict to each other.

14. The surface adsorption capability test through TGA may be erroneous as MoSe₂ can get oxidized showing weight loss.

15. Figure 3f caption and the numbers in the main text are not matching.

16. There is no experimental synthetic information corresponding to LSC/MoSe₂.

Reviewer #2 (Remarks to the Author):

In this work entitled "In-situ Local Phase Transition of MoSe₂ in La_{0.5}Sr_{0.5}Co_{3-δ} Heterostructure and Excellent Overall Water Electrolysis over 1000 hours", the authors demonstrated a perovskite oxides-transition metal dichalcogenides material with high surface adsorption capability, favorable kinetics and low charge transfer resistance which has great potential in overall water electrolysis application. The explanation of mechanism is clear and the analysis of date is detailed and basically correct. However, some issues should be addressed before considered for publication.

1. While the high energy ball-milling process might also cause the phase change of MoSe₂, the authors should compare the LSC&MoSe₂ / KB sample and MoSe₂ / KB for clarification.
2. The authors suggest that during the ball-milling process the electrons transfer from Co to Mo spontaneously, so that the difference of products employing different ratios of LSC and MoSe₂ should be measured.
3. While 1000 hours lifespan at high current density of 100 mA cm⁻² is enough to prove the durability of LSC&MoSe₂ / KB, the authors should also give the energy efficiency of overall water electrolysis at such high current density to improve the practicability of this material.
4. Please thoroughly check the whole manuscript and revise the typos and grammar errors.

Reviewer #3 (Remarks to the Author):

In general, the manuscript is very well written and the conclusions are well supported by the presented experimental data. I therefore only have a few questions/comments/suggestions:

1. In the introduction, it is mentioned that the HER should preferentially be performed in acidic media, and the OER in alkaline media. So why did the authors only investigate the performance of their materials in alkaline electrolytes? It would be interesting to know if the excellent stability could also be obtained in acidic electrolytes.
2. On page 9, several different sample abbreviations are used, but it remains unclear to me what is the difference between "LSC&MoSe₂" and "LSC/MoSe₂" and between "LSC" and "pure LSC"...
3. The references used for benchmarking the new catalysts with data available in the literature seems a bit arbitrary, as a lot of different materials are presented, but this list is of course far from complete. Could the authors elaborate on the selection criteria they used for comprising Tables S1, S2 and S5?
4. The SEM images presented in Figure S6 should be of higher magnification in order to really observe any differences between before and after water splitting. Furthermore, it would also be interesting to show XPS data after overall water splitting in order to observe whether the claims made previously (ratio of surface-active oxygen vs lattice oxygen and Co³⁺ vs Co²⁺) still hold during/after the water splitting experiments.

Overall, I would give the following recommendation: Publish with minor revisions.

Reviewer(s)' Comments to Author:

Reviewer: 1

Comments:

The manuscript entitled “In-situ Local Phase Transition of MoSe₂ in La_{0.5}Sr_{0.5}Co_{3-δ} Heterostructure and Excellent Overall Water Electrolysis over 1000 hours” addresses the design of a hetero-structure consisting of two materials, LSC and MoSe₂ and improved electrochemical oxygen (OER) and hydrogen evolution (HER) activity in the composite than the individual materials (LSC and MoSe₂). Here they have explained that on combining LSC and MoSe₂, Co donates electron to more electronegative Mo and causes the phase transition of MoSe₂ from semiconducting 2H to metallic 1T phase which improves HER activity and due to the donation of electron from Co, generation of high ratio of Co³⁺ to Co²⁺ facilitates the OER activity. Although authors have performed long term stability (1000 h), but the activity is still poor in comparison to recently reported many other active catalyst systems in terms of cell voltage. Both of the catalysts are already reported and the approach of making composite is also trivial. This work is not suitable for Nature Communications. However there are issues the authors should address before submitting elsewhere.

[Response]

→ We thank the reviewer for providing the critical comments on our work. However, we believe our findings have their own uniqueness and importance for the following several reasons: First, in overall water splitting catalysts, the stability is as important as cell voltage. As far as we know, our stability results show the highest stability (1,000 h) ever reported compared to the previous works, even under very harsh test conditions (at 100 mA cm⁻²). Second, although each of the perovskite oxide and transition metal dichalcogenides (TMDs) system as the electrocatalyst has been reported, our study on the application of perovskite oxide/TMDs heterostructure composite as the overall water electrolysis catalyst is the first attempt to be reported. Notably, such composite structured electrocatalysts showed significantly improved electrocatalytic performance compared to each consisting component, perovskite oxide and TMDs, in terms of both hydrogen evolution reaction (HER) and oxygen evolution reaction (OER), as well as the dramatically enhanced long-term stability. Third, we actually think that the simple processability introduced in this work is indeed rather advantageous in that the composite structure can be synthesized by a simple method as described in the manuscript, unlike previously reported works which typically involve rather complex synthesis processes^{R1-5}. Lastly, for the first time, we discovered that *in-situ* local phase transition can be induced between the perovskite oxide and TMDs, attributed to the electron transfer between the two constituent materials, which was confirmed through both experimental and computational analyses.

Supplementary references

- R1. Lin, Y., Tian, Z., Zhang, L., Ma, J., Jiang, Z., Deibert, B. J., Ge, R. & Chen, L. Chromium-ruthenium oxide solid solution electrocatalyst for highly efficient oxygen evolution reaction in acidic media. *Nat. commun.* **10**, 162 (2019).
- R2. Zhou, G., Shan, Y., Wang, L., Hu, Y., Guo, J., Hu, F., Shen, J., Gu, Y., Cui, J., Liu, L. & Wu, X. Photoinduced semiconductor-metal transition in ultrathin troilite FeS nanosheets to trigger efficient hydrogen evolution. *Nat. commun.* **10**, 399 (2019).
- R3. Zhang, J., Zhao, Y., Guo, X., Chen, C., Dong, C. L., Liu, R. S., Han, C. P., Li, Y., Gogotsi, Y. & Wang, G. Single platinum atoms immobilized on an MXene as an efficient catalyst for the hydrogen evolution reaction. *Nat. Catal.* **1**, 985 (2018).
- R4. Liu, X., He, J., Zhao, S., Liu, Y., Zhao, Z., Luo, J., Hu, G., Sun, X. & Ding, Y. Self-powered H₂ production with bifunctional hydrazine as sole consumable. *Nat. commun.* **9**, 4365 (2018).

R5. Li, H., Chen, S., Zhang, Y., Zhang, Q., Jia, X., Zhang, Q., Gu, L., Sun, X., Song, L. & Wang, X. Systematic design of superaerophobic nanotube-array electrode comprised of transition-metal sulfides for overall water splitting. *Nat. commun.* **9**, 2452 (2018).

1. The essence of this work is the phase transition in MoSe₂ that helps the electrocatalytic activity. However this concept is pretty well known and reported earlier. See: Chem. Commun., 2015, 51, 8450-8453 among many.

[Response]

→ We thank for the reviewer's comment. We are also well aware of the studies that reported the effect of phase transition in MoSe₂ to electrochemical activity. However, in previous studies (including Chem. Commun., 2015, 51, 8450-8453 mentioned by the reviewer), the phase transition in TMDs is typically conducted through rather complicated alkali metal assisted chemical exfoliation process in inert environment. In this work, we, for the first time, reported *in-situ* phase transition of MoSe₂ through the electron transfer between La_{0.5}Sr_{0.5}Co₃O_{3-δ} (perovskite oxide) and MoSe₂ (TMDs) by forming the composite structure which can be readily synthesized via simple ball-milling processes in ambient condition without the use of alkali metal. We believe that such straightforward and simple approach shall provide a new perspective for the phase transition of TMDs materials in general.

2. The relative abundance of 1T and 2H phases needs to be quantified. As to co-relate with the proposed structural schematics, the composite should be studied better through TEM.

[Response]

→ We appreciate the reviewer for the constructive comments. The reviewer suggested confirming the relative abundance of 1T- and 2H-phase via transmission electron microscopy (TEM). However, quantifying the overall 1T/2H ratio via TEM is rather limited because TEM only allows analysis over a limited region of the given sample. We think it is more reasonable to verify the 1T/2H ratio through x-ray photoelectron spectroscopy (XPS), which is widely adopted technique in such analysis^{R6-9}. Therefore, we quantified the relative presence of 1T- and 2H-phase in MoSe₂ through XPS. We included the additional discussion in the revised manuscript and the related figure and table is presented in Supporting Information as follows.

<Revised text, figure, and table>

Original text:

→ The lattice spacings of Mo-Mo and Se-Se were 0.563 and 0.324 nm, respectively, confirming the presence of 1T-MoSe₂^{29,55}. These results suggest that the local phase transition in MoSe₂ from the 2H-phase to the 1T-phase occurred in-situ during formation of the heterostructure between LSC and 2H-phase MoSe₂.

Revised text:

→ The lattice spacings of Mo-Mo and Se-Se were 0.563 and 0.324 nm, respectively, confirming the presence of 1T-MoSe₂^{29,61}. The coexistence of 1T- and 2H-phase of MoSe₂ in LSC&MoSe₂ was verified by XPS analysis shown in **Supplementary Fig. 13** (1T-phase: Mo 3d_{5/2} for 228.3 eV, Mo 3d_{3/2} for 231.4 eV, Se 3d_{5/2} for 53.7 eV, and Se 3d_{3/2} for 54.7 eV, 2H-phase: Mo 3d_{5/2} for 229.0 eV, Mo 3d_{3/2} for 232.6 eV, Se 3d_{5/2} for 54.3 eV, and Se 3d_{3/2} for 55.6 eV)^{29,70-72}. The relative contents of 1T- and 2H-phase MoSe₂ from the as-prepared LSC&MoSe₂ are summarized in **Supplementary Table 6**. To verify whether the ball milling process has any effect on the phase transition of MoSe₂, MoSe₂/KB was synthesized under the same ball-mill process as LSC&MoSe₂ and investigated through XPS and HR-TEM analysis. As shown in **Supplementary Fig. 14, 15**, no signature of phase transition is observed and only 2H-phase MoSe₂ is present in ball-milled MoSe₂/KB. These

results suggest that the local phase transition in MoSe₂ from the 2H-phase to the 1T-phase occurred in-situ during formation of the heterostructure between LSC and 2H-phase MoSe₂.

Added figure:

Figure S13. Mo 3d and Se 3d XPS spectra of LSC&MoSe₂, indicating the coexistence of 1T- and 2H-phase MoSe₂.

Added table:

Table S6. Quantitative analysis of 1T- and 2H-phase MoSe₂ in LSC&MoSe₂ obtained from the XPS result in Figure S13.

MoSe ₂ in LSC&MoSe ₂ (atom %)	
1T	ca. 58
2H	ca. 42

Added references in the revised manuscript

- ➔ 70. Zhang, J., Wang, T., Liu, P., Liu, Y., Ma, J. & Gao, D. Enhanced Catalytic Activities of Metal-Phase-Assisted 1T@2H-MoSe₂ Nanosheets for Hydrogen Evolution. *Electrochim. Acta.* **217**, 181-186 (2016).
- 71. Qu, Y., Medina, H., Wang, S. W., Wang, Y. C., Chen, C. W., Su, T. Y., Manikandan, A., Wang, K., Shih, Y. C. & Chang, J. W. Wafer Scale Phase-Engineered 1T and 2H MoSe₂/Mo Core-Shell 3D Hierarchical Nanostructures toward Efficient Electrocatalytic Hydrogen Evolution Reaction. *Adv. Mater.* **28**, 9831 (2016)
- 72. Li, X., & Peng, K. MoSe₂/Montmorillonite composite nanosheets: Hydrothermal synthesis, structural characteristics, and enhanced photocatalytic activity. *Minerals*, **8**, 268 (2018).

Supplementary references

- ➔ R6. Deng, S., Zhong, Y., Zeng, Y., Wang, Y., Yao, Z., Yang, F., Lin, S., Wang, X., Lu, X., Xia, X. & Tu, J. Directional Construction of Vertical Nitrogen Doped 1T 2H MoSe₂/Graphene Shell/Core Nanoflake Arrays for Efficient Hydrogen Evolution Reaction. *Adv. Mater.* **29**, 1700748 (2017).
- R7. Zhang, J., Wang, T., Liu, P., Liu, Y., Ma, J. & Gao, D. Enhanced Catalytic Activities of Metal-

Phase-Assisted 1T@2H-MoSe₂ Nanosheets for Hydrogen Evolution. *Electrochim. Acta.* **217**, 181-186 (2016).

R8. Qu, Y., Medina, H., Wang, S. W., Wang, Y. C., Chen, C. W., Su, T. Y., Manikandan, A., Wang, K., Shih, Y. C. & Chang, J. W. Wafer Scale Phase Engineered 1T and 2H MoSe₂/Mo Core–Shell 3D Hierarchical Nanostructures toward Efficient Electrocatalytic Hydrogen Evolution Reaction. *Adv. Mater.* **28**, 9831 (2016)

R9. Eda, G., Yamaguchi, H., Voiry, D., Fujita, T., Chen, M. & Chhowalla, M. Photoluminescence from chemically exfoliated MoS₂. *Nano Lett.* **11**, 5111-5116 (2011).

3. The authors did not optimize the weight ratio of MoSe₂ and LSC, they have taken only one composition of MoSe₂ and LSC.

[Response]

→ We thank the reviewer for the helpful comments. Actually, the weight ratio of 2:1 for LSC and MoSe₂ mentioned in the original manuscript was the optimized value for the heterogenous electrocatalyst formation. We measured the HER and OER polarization profiles for various weight ratios of LSC and MoSe₂ as shown in Figure S1. Both the HER and OER activities increased with the addition of MoSe₂ to LSC upto the weight ratio 6:3:1 of LSC:MoSe₂:KB (LSC&MoSe₂), reaching to the maximum performance. Additional increase of MoSe₂ to LSC (LSC:MoSe₂:KB = 5:4:1) resulted in substantial reduction in HER and OER performance due to the decreased phase transition in MoSe₂ flakes covering the LSC particles. Thus, the optimum ratio for LSC&MoSe₂ was determined to be LSC:MoSe₂:KB = 6:3:1. We included the additional discussion in the revised manuscript and the related figure is presented in Supporting Information as follows.

<Revised text and figure>

Original text:

→ LSC&MoSe₂ was prepared using the high-energy ball milling process (See Experimental Section for details). Morphological and structural analyses of the composite electrocatalyst were first performed by transmission electron microscopy (TEM).

Revised text:

→ LSC&MoSe₂ was prepared using the high-energy ball milling process with the optimum weight ratio of LSC:MoSe₂:KB = 6:3:1 determined by the electrochemical analyses (Supplementary Fig. S1, S2, and S3) (See Experimental Section for details). Morphological and structural analyses of the composite electrocatalyst were first performed by transmission electron microscopy (TEM) and scanning electron microscopy (SEM).

Added figure:

Figure S1. a) HER and b) OER polarization curves obtained by the various weight ratios of LSC and MoSe₂. For the preparation of catalysts, 10 wt.% of KB was included as a conductive support.

4. The authors have done OER measurements by purging N₂ in KOH, but according to Nernst equation, the OER measurements should be done followed by purging O₂ to maintain proper equilibrium between O₂ (product) and OH⁻ (reactant) [Ref: ACS Catal. 2018, 8, 1913–1920]

[Response]

→ We thank the reviewer for the helpful comments. Recently, some publication reported that the oxygen concentration in KOH solution could contribute to the increase of OER performance because the oxygen molecules could participate in the electrochemical reaction process^{R10}. Also, the oxygen concentration could affect the OER activities based on Nernst equation as the reviewer commented. Since a typical water electrolysis experiment proceeds in a deaerated (N₂ purging) KOH solution to prevent oxygen reduction reaction (ORR) at the cathode side, many articles related to water electrolysis report OER performance measured in a deaerated KOH solution^{R11-20}. For metal-air battery systems, OER activities should be evaluated under O₂ purged conditions because metal-air batteries operate at the environment with high oxygen concentration. Therefore, OER performance evaluation under O₂ purging condition may be erroneous when considering the actual water electrolysis operating conditions. Below, we have listed the related references with OER measurement under N₂ purging condition. Thanks for your comments again.

Supplementary references

- R10. Grimaud, A., Diaz-Morales, O., Han, B., Hong, W. T., Lee, Y.-L., Giordano, L., Stoerzinger, K. A., Koper, M. T. M & Shao-Horn, Y. Activating lattice oxygen redox reactions in metal oxides to catalyse oxygen evolution. *Nat. Chem.* **9**, 457-465 (2017).
- R11. Swesi, A. T., Masud, J., Liyanage, W. P. R., Umaphathi, S., Bohannan, E., Medvedeva, J. & Nath, M. Textured NiSe₂ Film: Bifunctional Electrocatalyst for Full Water Splitting at Remarkably Low Overpotential with High Energy Efficiency. *Sci. Rep.* **7**, 2401 (2017).
- R12. Zhu, H. *et al.* When Cubic Cobalt Sulfide Meets Layered Molybdenum Disulfide: A Core-Shell System Toward Synergetic Electrocatalytic Water Splitting. *Adv. Mater.* **27**, 4752-4759 (2015).
- R13. Han, X. *et al.* Engineering Catalytic Active Sites on Cobalt Oxide Surface for Enhanced Oxygen Electrocatalysis. *Adv. Energy Mater.* **8**, 1702222 (2017).
- R14. Chen, C.-F., King, G., Dickerson, R. M., Papin, P. A., Gupta, S., Kellogg, W. R. & Wu, G. Oxygen-deficient BaTiO_{3-x} perovskite as an efficient bifunctional oxygen electrocatalyst. *Nano Energy* **13**, 423-432 (2015).

- R15. Park, H. W., Lee, D. U., Zamani, P., Seo, M. H., Nazar, L. F. & Chen, Z. Electrospun porous nanorod perovskite oxide/nitrogen-doped graphene composite as a bi-functional catalyst for metal air batteries. *Nano Energy* **10**, 192-200 (2014).
- R16. Hua, B., Li, M., Sun, Y.-F., Zhang, Y.-Q., Yan, N., Chen, J., Thundat, T., Li, J. & Luo, J.-L. A coupling for success: Controlled growth of Co/CoO_x nanoshoots on perovskite mesoporous nanofibres as high-performance trifunctional electrocatalysts in alkaline condition. *Nano Energy* **32**, 247-254 (2017).
- R17. Seo, M. H., Park, H. W., Lee, D. U., Park, M. G. & Chen, Z. Design of Highly Active Perovskite Oxides for Oxygen Evolution Reaction by Combining Experimental and ab Initio Studies. *ACS Catal.* **5**, 4337-4344 (2015).
- R18. Oh, A., Kim, H. Y., Baik, H., Kim, B., Chaudhari, N. K., Joo, S. H. & Lee, K. Topotactic Transformations in an Icosahedral Nanocrystal to Form Efficient Water-Splitting Catalysts. *Adv. Mater.* 1805546 (2018).
- R19. Huang, S., Meng, Y., He, S., Goswami, A., Wu, Q., Li, J., Tong, S., Asefa, T. & Wu, M. N., O-, and S-Tridoped Carbon-Encapsulated Co₉S₈ Nanomaterials: Efficient Bifunctional Electrocatalysts for Overall Water Splitting. *Adv. Funct. Mater.* **27**, 1606585 (2017).
- R20. Mefford, J. T., Rong, X., Abakumov, A. M., Hardin, W. G., Dai, S., Kolpak, A. M., Johnston, K. P. & Stevenson, K. J. Water electrolysis on La_{1-x}Sr_xCoO_{3-δ} perovskite electrocatalysts. *Nat. Commun.* **7**, 11053 (2016).

5. When the measurements are performed in GC, the activities for both OER and HER for the mixed LSC and MoSe₂ are poor. They have obtained a cell voltage of 1.65 V only after using Ni foam which itself is an efficient current collector, whereas there are several reports obtaining lesser cell voltage using Ni foam only. [Ref: Adv. Energy Mater. 2017, 1700513; Adv. Funct. Mater. 2017, 27, 1701008]

[Response]

→ We thank the reviewer for the prudent comments. The reviewer considered our OER and HER activities as somewhat low, which is not the best performance ever reported as of now across all material families. Nonetheless, both the OER and HER activities are actually quite excellent compared to the previously reported various types of perovskite oxide and TMDs catalysts as shown in Table S1 and S2. We would like to emphasize this study has great significance in that it first proposed the design of a new overall water splitting catalyst by forming the perovskite oxide/TMDs heterogenous composite structure. Furthermore, this work revealed for the first time that the *in-situ* local phase transition phenomenon in TMDs can take place owing to the charge transfer between perovskite oxide and TMDs. To the best of our knowledge, this work is the first to report such phenomenon, and additional computational studies suggested by the reviewer (please see response to Reviewer #1's question #6) further corroborated our claims. Owing to the local phase transition in MoSe₂, the catalytic performance is notably improved and outstanding operational stability over 1,000 hours at high current density (100 mA cm⁻²) was obtained. In this regard, we strongly believe that the novelty and impact of this work is significant.

In addition, we revisited our cell voltage analysis and replotted the Figure 6b with a logarithmic scale of current density to more precisely evaluate the onset potentials of water electrolysis for each catalyst. As shown in Figure S20, the logarithmic water electrolysis profile clearly indicates that the onset potentials are actually obtained as 1.38 and 1.52 V for Pt/C || IrO₂ and LSC&MoSe₂ || LSC&MoSe₂, respectively, NOT the 1.50 and 1.65 V as initially presented. Regarding the question of “there are several reports obtaining lesser cell voltage using Ni foam only. [Ref: Adv. Energy Mater. 2017, 1700513; Adv. Funct. Mater. 2017, 27, 1701008].”, we double-checked the suggested papers but the papers do not show the water splitting performance using Ni foam only. In this regard, we examined the water electrolysis performance of Ni foam itself, and compared our result to several published works. As shown in Figure S19, the water electrolysis performance of bare Ni foam is in fact measured far less than that of the LSC&MoSe₂ catalyst, which was also comparable

to the result of previously reported literature values^{R21-24}. We included the additional discussion in the revised manuscript and the related figure is presented in Supporting Information as follows.

<Revised text and figures>

Original text:

- When LSC&MoSe₂ || LSC&MoSe₂ electrode was applied to overall water splitting, the initiation potential (1.65 V) was slightly higher than that of the Pt/C || IrO₂ reference electrode (1.50 V), but the former started to outperform the latter after 2.16 V.

Revised text:

- When LSC&MoSe₂ || LSC&MoSe₂ electrode was applied to overall water splitting, the initiation potential (1.52 V) was slightly higher than that of the Pt/C || IrO₂ reference electrode (1.38 V), but the former started to outperform the latter after 2.16 V.

Original text:

- **Figure 5b** shows the HER and OER polarization curves, denoted E_{cathode} and E_{anode} , respectively, measured during the water electrolysis reaction, and the obtained cell potentials ($E_{\text{cell}} = E_{\text{anode}} - E_{\text{cathode}}$) of Pt/C || IrO₂ (Pt/C for cathode and IrO₂ for anode) and LSC&MoSe₂ || LSC&MoSe₂ (LSC&MoSe₂ for both cathode and anode). The reference state-of-the-art catalyst pair, Pt/C || IrO₂, performed well in both HER and OER, as observed previously in the half-cell-configured RDE polarization profiles. In the case of Pt/C || IrO₂, overall water splitting was observed from near 1.5 V. LSC&MoSe₂ || LSC&MoSe₂ exhibited slightly lower HER performance than Pt/C, but the OER performance was higher than that of IrO₂. Although overall water splitting was observed at a slightly higher value of 1.65 V, current density increased sharply as the potential increases. Notably, LSC&MoSe₂ || LSC&MoSe₂ outperformed Pt/C || IrO₂ after 2.16 V owing to its concurrently effective HER and OER kinetics.

Revised text:

- **Figure 6b** shows the HER and OER polarization curves, denoted E_{cathode} and E_{anode} , respectively, measured during the water electrolysis reaction, and the obtained cell potentials ($E_{\text{cell}} = E_{\text{anode}} - E_{\text{cathode}}$) of Pt/C || IrO₂ (Pt/C for cathode and IrO₂ for anode) and LSC&MoSe₂ || LSC&MoSe₂ (LSC&MoSe₂ for both cathode and anode). Also, the overall water electrolysis performance of bare Ni foam is investigated and compared to the published works (**Supplementary Fig. 19**). The reference state-of-the-art catalyst pair, Pt/C || IrO₂, performed well in both HER and OER, as observed previously in the half-cell-configured RDE polarization profiles. In the case of Pt/C || IrO₂, overall water splitting was observed from near 1.38 V. LSC&MoSe₂ || LSC&MoSe₂ exhibited slightly lower HER performance than Pt/C, but the OER performance was higher than that of IrO₂. Although overall water splitting was observed at a slightly higher value of 1.52 V, current density increased sharply as the potential increases. The determination of the onset potential values for each catalyst can be seen in **Supplementary Fig. 20**. Notably, LSC&MoSe₂ || LSC&MoSe₂ outperformed Pt/C || IrO₂ after 2.16 V owing to its concurrently effective HER and OER kinetics.

Added figure:

Figure S19. Linear sweep voltammetry curves of the overall water splitting measured using the LSC&MoSe₂ catalyst loaded Ni foam electrode and the bare Ni foam electrode, compared with the bare Ni foam performance results reported in the literature.

Figure S20. Overall water splitting performance measured by Pt/C || IrO₂ and LSC&MoSe₂ || LSC&MoSe₂. Figure 6b was replotted with a logarithmic scale of current density to evaluate the onset potentials of electrolysis at a current density of 1 mA cm⁻².

Supplementary references

- ➔ R21. Zhu, Y., Zhou, W., Zhong, Y., Bu, Y., Chen, X., Zhong, Q., Liu, M. & Shao, Z. A Perovskite Nanorod as Bifunctional Electrocatalyst for Overall Water Splitting. *Adv. Energy Mater.* **7**, 1602122 (2017).
- R22. Huang, S., Meng, Y., He, S., Goswami, A., Wu, Q., Li, J., Tong, S., Asefa, T. & Wu, M. N-, O-, and S-Tridoped Carbon-Encapsulated Co₉S₈ Nanomaterials: Efficient Bifunctional Electrocatalysts for Overall Water Splitting. *Adv. Funct. Mater.* **27**, 1606585 (2017).

R23. Lendendecker, M., Calderon, S. K., Papp, C., Steinruck, H.-P., Antonietti, M. & Shalom, M. The Synthesis of Nanostructured Ni₃P₄ Films and their Use as a Non-Noble Bifunctional Electrocatalyst for Full Water Splitting. *Angew. Chem. Int. Ed.* **54**, 12361-12365 (2015).

R24. Tang, C., Cheng, N., Pu, Z., Xing, W. & Sun, X. NiSe Nanowire Film Supported on Nickel Foam: An Efficient and Stable 3D Bifunctional Electrode for Full Water Splitting. *Angew. Chem. Int. Ed.* **54**, 9351-9355 (2015).

6. This study demands suitable computational support to prove the charge transfer phenomenon, which the authors have claimed playing a major role here.

[Response]

→ We thank for the reviewer's comments for enhancing the quality of our study. As suggested from the reviewer, we performed density functional theory (DFT) calculations to support the charge transfer phenomenon between LSC and MoSe₂ in the heterostructure. The DFT analysis revealed that the endothermicity required for phase transition from 2H- to 1T-phase of MoSe₂ can be significantly lowered from the charge transfer, implying that the local phase transition could occur more easily. In the revised manuscript, we added new section for theoretical calculation results, and explanations for model systems and calculation details (Experimental section) as follows.

<Added text and figure>

Added text:

→ Theoretical elucidation of charge transfer phenomenon in LSC&MoSe₂

To theoretically demonstrate the charge transfer phenomenon in LSC&MoSe₂ heterostructure, we performed the density functional theory (DFT) calculations (see "Calculation details" in the Experimental section). For this purpose, we first examined the relative stability of two plausible terminations of LSC (001) surface, corresponding to CoO₂-termination (denoted CoO₂-t.) and (La,Sr)O-termination (denoted (La,Sr)O-t.), by surface energy (γ) calculations (see **Supplementary Fig. 16** and "Surface energy calculations" for details in the Experimental section). As a result, (La,Sr)O-t. was predicted to be predominantly exposed on the LSC nanoparticle surface due to its lower surface energy (i.e., $\gamma = 0.58 \text{ J/m}^2$ for (La,Sr)O-t. and $\gamma = 0.71 \text{ J/m}^2$ for CoO₂-t., respectively). Accordingly, we built the LSC&2H-MoSe₂ heterostructure based on this termination (**Fig. 5a**). By the Bader charge analysis⁷⁶, it was revealed that the charge transfer occurred from LSC into MoSe₂ with an amount of 0.772 e , whose direction was in accordance with our previous expectation.

To better understand these electron movements, the projected density of states (PDOS) on each d-orbital of Co and Mo in LSC&2H-MoSe₂ was further investigated (**Fig. 5b**). The asymmetric configuration of spin-up and spin-down DOS for LSC indicated the ferromagnetic nature, whereas the symmetric configuration for MoSe₂ represented its non-magnetic character^{77,78}. Driven by the electron transfer across the interface, the d-band center (ϵ_d) of Co in LSC shifted upward about 0.1 eV with respect to the Fermi level when coupled with 2H-MoSe₂, indicating that LSC became more electrophilic. In a previous study, the enhanced electrophilicity of LSC was reported to strengthen the OH⁻ affinity, which has a beneficial effect to improve the OER catalytic performance⁵⁴. Meanwhile, the ϵ_d of Mo in 2H-MoSe₂ was significantly downshifted by approximately 0.9 eV with respect to the Fermi level when combined with LSC, implying that additional electrons transferred into 2H-MoSe₂ side. These extra d-electrons are expected to promote the 2H- to 1T-phase transition by destabilizing the 2H-MoSe₂ phase, as similarly observed in the MoS₂ system by Gao *et al.*'s work⁷⁹. To further clarify this, we compared the relative stability for both phases of MoSe₂ in the presence or absence of LSC (**Fig. 5c, d** and **Supplementary Fig. 17**). The endothermicity of 1T-MoSe₂ (relative to its 2H-phase) clearly decreased from 0.65 to 0.21 eV per MoSe₂ unit when LSC was combined, indicating that the phase transition from 2H- to 1T-phase could occur more easily. Further, the local DOS of MoSe₂ in the LSC&MoSe₂ heterostructure

showed that the intrinsic conductivity can be significantly enhanced by closing the energy gap due to the transition from semiconducting 2H- to metallic 1T-phase, which can improve the HER catalytic performance (Supplementary Fig. 18)⁸⁰. These results theoretically elucidate the beneficial effects of charge transfer phenomenon in LSC&MoSe₂ for both HER and OER performance.

→ **Calculation details.** Spin-polarized DFT calculations were carried out using the Vienna ab initio simulation package (VASP)⁸² within the projector-augmented wave (PAW) method⁸³. The electron exchange-correlation energy was treated by the generalized gradient approximation (GGA) with Perdew-Burke-Ernzerhof (PBE) functional⁸⁴. The DFT+*U* method within Dudarev's approach⁸⁵ was also adopted with *U* = 4.3 eV and *J* = 1.0 eV for Co-3*d* and *U* = 4.0 eV for Mo-4*d*, respectively. The energy cutoff for the plane-wave basis set was set as 400 eV and PAW data sets were used with following valence electronic states: 5*s*², 5*p*⁶, 5*d*¹, 6*s*² for La; 4*s*², 4*p*⁶, 5*s*² for Sr; 3*d*⁸, 4*d*¹ for Co; 2*s*², 2*p*⁴ for O; 4*s*², 4*p*⁶, 4*d*⁶ for Mo; and 4*s*², 4*p*⁴ for Se, respectively. Geometry optimizations were performed using conjugated gradient (CG) method until the net force on each atom reached less than 0.02 eV Å⁻¹ and the total energy was changed within 10⁻⁶ eV per atom. Dipole slab corrections were also applied to all slab model calculations. The Monkhorst-Pack scheme of *k*-point grid⁸⁶ was set to *Γ* points for geometry optimization, and 3 × 2 × 1 *k*-points in the Brillouin zone for DOS analysis, respectively. Bader analysis⁷⁶ was used to calculate the atomic charges.

Model systems for calculation. To construct the LSC&MoSe₂ heterostructure, each slab model for LSC surface and MoSe₂ layer was separately modeled in advance. First, the unit cell structure of LaCoO₃ (LCO) was fully relaxed by optimizing both atomic positions and lattice parameters, which were well matched with experimentally reported values⁸⁷. Subsequently, the LCO bulk structure was cleaved along the (001) plane with two plausible terminations (i.e., CoO₂ termination and LaO termination). Note that we considered symmetric slab models of LCO (001), which consisted of 5 atomic layers, to remove the fictitious dipole moment in the slab. The bottommost two layers were fixed to their bulk positions. Next, a 2√2 × 3√2 × 1 supercell structure was created, and subsequently half of La atoms were replaced into Sr atoms to attain the stoichiometry of La_{0.5}Sr_{0.5}CoO₃ system (i.e., *a* = 10.91 Å, *b* = 16.37 Å, *c* = 25.00 Å, 156 atoms). For MoSe₂ slab model, a 2 × 5 × 1 supercell structure of orthorhombic unit cell for 2H-MoSe₂ was created (i.e., *a* = 11.49 Å, *b* = 16.78 Å, *c* = 25.00 Å, 60 atoms). Finally, the LSC&MoSe₂ heterostructure was built by combining the LSC and MoSe₂ slabs with minimized lattice mismatch less than 3% (i.e., *a* = 11.20 Å, *b* = 16.57 Å, *c* = 35.00 Å, 216 atoms). The vacuum was sufficiently applied to avoid the self-interaction in *z*-direction.

Surface energy calculations. To evaluate the relative stability of complementary terminations in the LSC (001) surface, we calculated the surface energy (*γ*) by the sum of cleavage energy (*γ*₀) and relaxation energy (*γ*_{*r*}), which was previously reported by Heifets *et al.*⁸⁸.

$$\gamma = \gamma_u + \gamma_r \quad (1)$$

The cleavage energy can be obtained as follows,

$$\gamma_u = (E_{slab}^u(\text{CoO}_2\text{-t.}) + E_{slab}^u(\text{(La,Sr)O-t.}) - NE_{bulk}) / 4A \quad (2)$$

where $E_{slab}^u(\text{CoO}_2\text{-t.})$ and $E_{slab}^u(\text{(La,Sr)O-t.})$ are unrelaxed CoO₂- and (La,Sr)O-terminated slab energies, E_{bulk} is the total energy of bulk unit cell, *N* is the formula unit of slab models, and *A* is the surface area. The factor of four in the denominator arises from the four cleaved surfaces of two

terminations (i.e., CoO_2 -t. and $(\text{La,Sr})\text{O}$ -t.). Next, we can calculate the relaxation energies (γ_r) for each CoO_2 -t. and $(\text{La,Sr})\text{O}$ -t. of LSC (001) surface as follows,

$$\gamma_r = (E_{slab}^r(\text{X}) - E_{slab}^u(\text{X})) / 2A \quad (\text{X} = \text{CoO}_2\text{-t. or } (\text{La,Sr})\text{O-t.}) \quad (3)$$

where $E_{slab}^u(\text{CoO}_2\text{-t.})$ is a slab energy after relaxation

Added figures:

Figure 5. DFT calculations for charge transfer phenomenon in LSC&MoSe₂. a) Optimized structure of LSC&2H-MoSe₂ heterostructure. The red shaded box represents the fixed atoms in the two bottommost layers. The blue arrow indicates the direction of charge transfer from LSC to 2H-MoSe₂. b) Spin-up and spin-down projected density of states (PDOS) on the following species: Co-3d in LSC (blue dotted line), Co-3d in LSC&2H-MoSe₂ (blue solid line), Mo-4d in 2H-MoSe₂ (red dotted line), and Mo-4d in LSC&2H-MoSe₂ (red solid line), respectively. The vertical lines represent the position of d-band center (ϵ_d) for each species. c) Schematic illustration of phase transition from LSC&2H-MoSe₂ to LSC&1T-MoSe₂ heterostructure. d) Relative energies of 2H- and 1T-phase of MoSe₂ monolayer (black line) and LSC&MoSe₂ heterostructure (red line). The inset models represent the trigonal prismatic (2H) and octahedral (1T) geometry.

Figure S16. Optimized structures and corresponding surface energies (γ) of symmetrical slab models for LSC (001) surface with two plausible terminations (i.e., $\text{CoO}_2\text{-t.}$ and $(\text{La,Sr})\text{O-t.}$). The red shaded box represents the fixed atoms in the two bottommost layers.

Figure S17. Model systems used for relative energy calculation for 2H- and 1T-phase of MoSe₂ monolayer and LSC&MoSe₂ heterostructure. The red shaded box represents the fixed atoms in the two bottommost layers.

Figure S18. Spin-up and spin-down LDOS of MoSe₂ in LSC&2H-MoSe₂ (red) and LSC&1T-MoSe₂ (blue) heterostructure. The shaded area represents the valence band region. The green line represents the energy gap (E_g) between valence band maximum and conduction band minimum.

Added references in the revised manuscript

- 76. Tang, W., Sanville, E. & Henkelman, G. A grid-based Bader analysis algorithm without lattice bias. *J. Phys.: Condens. Matter.* **21**, 084204 (2009).
77. Zhang, S., Han, N. & Tan, X. Density functional theory calculations of atomic, electronic and thermodynamic properties of cubic LaCoO₃ and La_{1-x}Sr_xCoO₃ surfaces. *RSC Adv.* **5**, 760-769 (2015).
78. Li, H., Huang, S., Zhang, Q., Zhu, Z., Li, C., Meng, J. & Tian, Y. Nonmetal doping induced electronic and magnetic properties in MoSe₂ monolayer. *Chem. Phys. Lett.* **692**, 69-74 (2018).
79. Gao, G., Jiao, Y., Ma, F., Jiao, Y., Waclawik, E. & Du, A. Charge mediated semiconducting-to-metallic phase transition in molybdenum disulfide monolayer and hydrogen evolution reaction in new 1T' phase. *J. Phys. Chem. C.* **119**, 13124-13128 (2015).
80. Ambrosi, A., Sofer, Z. & Pumera, M. 2H → 1T phase transition and hydrogen evolution activity of MoS₂, MoSe₂, WS₂ and WSe₂ strongly depends on the MX₂ composition. *Chem. Commun.* **51**, 8450 (2015).
82. Kresse, G., & Furthmüller, J. Efficient iterative schemes for *ab initio* total-energy calculations using a plane-wave basis set. *Phys. Rev. B.* **54**, 11169 (1996).
83. Blochl, P. E. Projector augmented-wave method. *Phys. Rev. B.* **50**, 17953 (1994).
84. Perdew, J. P., Burke, K., & Ernzerhof, M. Generalized Gradient Approximation Made Simple. *Phys. Rev. Lett.* **77**, 3865 (1996).
85. Dudarev, S. L., Botton, G. A., Savrasov, S. Y., Humphreys, C. J., & Sutton, A. P. Electron-energy-loss spectra and the structural stability of nickel oxide: An LSDA+U study. *Phys. Rev. B.* **57**, 1505 (1998).
86. Monkhorst, H. J. & James D. Pack, Special points for Brillouin-zone integrations. *Phys. Rev. B.* **13**, 5188 (1976).
87. Jonker, G.H. & Van Santen, J.H. Magnetic compounds with perovskite structure III. ferromagnetic compounds of cobalt. *Physica.* **19**, 120-130 (1953).
88. Heifets, E., Eglitis, R. I., Kotomin, E. A., Maier, J., & Borstel, G. Ab initio modeling of surface structure for SrTiO₃ perovskite crystals. *Phys. Rev. B.* **64**, 235417 (2001).

7. The authors claimed that they have synthesized MoSe₂ flakes by exfoliating bulk MoSe₂, but the morphology is not clear in Fig. 1(a) and even after exfoliation there is an aggregated morphology.

[Response]

→ We thank the reviewer's helpful comments. To provide better observation on the morphology of LSC&MoSe₂, we performed additional analysis using scanning electron microscopy (SEM). As shown in Figure S4, formation of three-dimensional hierarchical architectures of LSC&MoSe₂ composite can be observed, and this kind of morphology from two-dimensional materials in heterostructure composite have been reported previously from various literatures^{R25-28}. This result suggests that, under our experimental conditions including ball milling and drying processes, the exfoliated MoSe₂ nanoflakes adsorbed onto the LSC surface can be obtained without causing much aggregations. Furthermore, the randomly distributed MoSe₂ flake based composite structure, which is expected to increase the surface area of LSC&MoSe₂ over LSC, can be beneficial to enhance the overall electrochemical catalytic performance by exposing more active sites. We included the additional discussion in the revised manuscript and the related figure is presented in Supporting Information as follows.

<Revised text and figure>

Original text:

→ Morphological and structural analyses of the composite electrocatalyst were first performed by transmission electron microscopy (TEM). **Figures 1a and b** show bright-field TEM image and high-angle annular dark-field (HAADF) image of LSC&MoSe₂, along with energy-dispersive spectroscopy (EDS) elemental mapping, which illustrates uniform distribution of the associated elements (La, Sr, Co, O, Mo, Se, and C) in LSC&MoSe₂.

Revised text:

→ Morphological and structural analyses of the composite electrocatalyst were first performed by transmission electron microscopy (TEM) and scanning electron microscopy (SEM). **Figures 1a and b** show bright-field TEM image and high-angle annular dark-field (HAADF) image of LSC&MoSe₂, along with energy-dispersive spectroscopy (EDS) elemental mapping, which clearly illustrates the presence of the associated elements (La, Sr, Co, O, Mo, Se, and C) in LSC&MoSe₂. Further analysis on the morphology of LSC&MoSe₂ heterostructure was carried out by SEM (Supplementary Fig. 4). It can be seen that MoSe₂ nanoflakes are randomly distributed and adsorbed onto the LSC surface without causing any noticeable aggregation, which can contribute to the increase of overall surface area of the composite structure.

Added figure:

Figure S4. SEM images at a) low and b) high magnification of as-prepared LSC&MoSe₂.

Supplementary references

- ➔ R25. Geng, X., Zhang, Y., Han, Y., Li, J., Yang, L., Benamara, M. & Zhu, H. Two-dimensional water-coupled metallic MoS₂ with nanochannels for ultrafast supercapacitors. *Nano Lett.* **17**, 1825-1832 (2017).
- R26. Li, Y., Wang, H., Xie, L., Liang, Y., Hong, G. & Dai, H. MoS₂ nanoparticles grown on graphene: an advanced catalyst for the hydrogen evolution reaction. *J. Am. Chem. Soc.* **133**, 7296-7299 (2011).
- R27. Chang, K., Mei, Z., Wang, T., Kang, Q., Ouyang, S. & Ye, J. MoS₂/graphene cocatalyst for efficient photocatalytic H₂ evolution under visible light irradiation. *ACS Nano* **8**, 7078-7087 (2014).
- R28. Yu, H., Ma, C., Ge, B., Chen, Y., Xu, Z., Zhu, C., Li, C., Ouyang, Q., Gao, P., Li, J., Sun, C., Qi, L., Wang, Y. & Li, F. Three-Dimensional Graphene/MoS₂ Nanoflake Arrays and Their Rapid Charging/Discharging Properties as Lithium Ion Battery Anodes. *Chem. Eur. J.* **19**, 5818-5823 (2013).

8. In ADF-EDS mapping for Mo and Se (Fig. 1b), authors claimed uniform distribution of different elements which is not evident from the distribution presented here.

[Response]

- ➔ We thank for the reviewer's constructive comments. To better support the ADF-EDS mapping result, we conducted SEM-EDS and elemental quantitative analysis to show the relative distribution of each constituent component (La, Sr, Co, O, Mo, and Se) in LSC&MoSe₂. The measurement results revealed that each consisting atomic component in LSC&MoSe₂ was clearly observed with expected elemental ratio. We included the additional discussion in the revised manuscript and the related figure is presented in Supporting Information as follows.

<Revised text and figure>

Original text:

- ➔ Morphological and structural analyses of the composite electrocatalyst were first performed by transmission electron microscopy (TEM). **Figures 1a and b** show bright-field TEM image and high-angle annular dark-field (HAADF) image of LSC&MoSe₂, along with energy-dispersive spectroscopy (EDS) elemental mapping, which illustrates uniform distribution of the associated elements (La, Sr, Co, O, Mo, Se, and C) in LSC&MoSe₂. **Figure 1c** shows a high-resolution TEM (HR-TEM) image of LSC&MoSe₂, which highlights the presence of MoSe₂ (red).

Revised text:

- ➔ Morphological and structural analyses of the composite electrocatalyst were first performed by transmission electron microscopy (TEM) and scanning electron microscopy (SEM). **Figures 1a and b** show bright-field TEM image and high-angle annular dark-field (HAADF) image of LSC&MoSe₂, along with energy-dispersive spectroscopy (EDS) elemental mapping, which clearly illustrates the presence of the associated elements (La, Sr, Co, O, Mo, Se, and C) in LSC&MoSe₂. Further analysis on the morphology of LSC&MoSe₂ heterostructure was carried out by SEM (Supplementary Fig. 4). It can be seen that MoSe₂ nanoflakes are randomly distributed and adsorbed onto the LSC surface without causing any noticeable aggregation, which can contribute to the increase of overall surface area of the composite structure. The SEM-EDS and elemental quantitative analysis for LSC&MoSe₂ further revealed that each constituent atomic component in the composite structure was clearly observed with expected elemental ratio (Supplementary Fig. 5). **Figure 1c** shows a high-resolution TEM (HR-TEM) image of LSC&MoSe₂, which highlights the presence of MoSe₂ (red).

Added figure:

Figure S5. SEM-EDS and elemental quantitative analysis of as-prepared LSC&MoSe₂, indicating that each constituent atomic component (La, Sr, Co, O, Mo, and Se) in LSC&MoSe₂ is clearly observed with expected elemental ratio (The red rectangle in inset SEM image represents the selected EDS mapping area).

9. PXRD analysis: For LSC presence of small peaks can be observed around 2theta 25 and 36. Peak at around 36 may attribute to the cobalt oxide impurity. Again, the amount of LSC used is more as compared to its counterpart, MoSe₂. But from XRD plot this is not reflected.

[Response]

→ We are grateful for the reviewer's valuable comments. To confirm the accuracy of XRD result, we remeasured the XRD spectra of LSC&MoSe₂, MoSe₂, and LSC. The XRD results of the remeasured LSC showed a clean peak without cobalt oxide impurities, and both LSC&MoSe₂ and MoSe₂ showed higher crystallinity than previous XRD measurements. For the synthesis of LSC this time, LSC was prepared with a disk-shaped pellet by pelletizing a pre-calcined LSC powder. Due to the high synthesis temperature for LSC, the LSC pellet was placed onto the sacrificial LSC bedding powder to prevent the direct Co diffusion into the ceramic plate from the LSC pellet. Then, the pellet was sintered at 950 °C for 4 hours. After the completion of sintering process, cobalt oxide impurities may be generated on the surface of the pellet, so the surface was slightly polished before the XRD measurement to avoid any contamination, and a clean XRD profile of LSC was now successfully obtained without cobalt oxide impurities as shown in revised Figure 1e. Regarding the comment on, "The amount of LSC is more than that of MoSe₂ but it is not reflected in the XRD plot.", since the intensity of XRD profiles are closely related to the crystallinity of test materials, the relative quantitative contribution of each component material may not be necessarily directly reflected on the XRD measurement. However, our XPS results do reveal that both the LSC and MoSe₂ related peaks are present in the LSC&MoSe₂ composite. We modified the related figure in revised manuscript as follows.

<Modified figure>

Original figure:

Figure 1. Morphological and structural characterizations of LSC&MoSe₂. e) XRD spectra of LSC, MoSe₂, and LSC&MoSe₂, indicating the well-mixed state of LSC and MoSe₂.

Revised figure:

Figure 1. Morphological and structural characterizations of LSC&MoSe₂. e) XRD spectra of LSC, MoSe₂, and LSC&MoSe₂, indicating the well-mixed state of LSC and MoSe₂.

10. Surface area analysis: Authors reported that surface in case of composite material is greatly enhanced in comparison to LSC only. Which one is contributing for this enhancement, LSC or MoSe₂? Generally the flake like MoSe₂ separately has a high BET surface area due to its two dimensional morphology, but the authors did not mention the individual surface area of MoSe₂. They explained that increase in surface area in the composite enhances the exposure of active sites, but according to their explanation, the active site for OER and HER is different. For OER, active site is mainly Co present in LSC, and MoSe₂ is mainly responsible for HER activity while the charge transfer between these two is responsible for the enhanced electrochemical activity in the composite. If the active sites for OER and HER are coming from two different materials in the composite, then just by making the composite cannot increase the total

number of active sites for both OER and HER.
- Moreover for determining active sites, electrochemically active surface area determination is more important rather than BET surface area.
- In supplementary fig. 1, the authors showed that mesopores are present in the surface of the heterostructure, but they did not explain the origin of mesopores, whether it is for LSC or MoSe₂.

[Response]

- We thank for the reviewer's constructive comments, which we believe can be summarized in the following perspectives.
1. Explain whether the origin of increase in surface area is due to LSC or MoSe₂.
 2. Check the surface area of MoSe₂.
 3. If the HER, OER active site is derived from different materials, the origin of total active site increase in LSC&MoSe₂ needs to be discussed.
 4. ECSA measurement needs to be done for the catalyst materials.
 5. Determine whether the mesoporous character is due to LSC or MoSe₂.

First, our surface area analysis data (Fig. 1f) compares LSC/MoSe₂/KB (denoted LSC&MoSe₂) and LSC/KB (denoted LSC). As shown in Figure 1f, increased surface area was observed in LSC&MoSe₂ after adding the MoSe₂ to LSC. Based on these results, it is reasonable to infer that the increase of surface area in LSC&MoSe₂ is attributed to the additional MoSe₂ present. Similar observations to our study have been previously reported, where the formation of heterostructure using two-dimensional materials led to the increased surface area in the composite materials^{R29-31}. In addition, the morphology analysis of LSC&MoSe₂ further supports our claim in that the increased surface area for the composite structure is originated from the MoSe₂ (please see the response to Reviewer #1's question 7).

<Revised text>

Original text:

- As shown in Fig. 1f, the addition of MoSe₂ to LSC led to a notable increase in the total surface area of the composite structure, where the surface area of LSC&MoSe₂ (142.09 m² g⁻¹) was more than thrice that of LSC (39.95 m² g⁻¹). Such a remarkable increase in surface area improves water splitting by increasing the total number of active electrocatalysis sites³².

Revised text:

- However, as shown in Fig. 1f, the addition of MoSe₂ to LSC led to a notable increase in the total surface area of the composite structure, where the surface area of LSC&MoSe₂ (142.09 m² g⁻¹) was more than thrice that of LSC (39.95 m² g⁻¹). Increase in the BET surface area for LSC&MoSe₂ can be attributed to the additional MoSe₂ nanoflakes present that are adsorbed onto the LSC surface. Such a remarkable increase in surface area improves water splitting by increasing the total number of active electrocatalysis sites for both HER and OER^{32,33}.

Supplementary references

- R29. Wu, Y., Xu, M., Chen, X., Yang, S., Wu, H., Pan, J. & Xiong, X. CTAB-assisted synthesis of novel ultrathin MoSe₂ nanosheets perpendicular to graphene for the adsorption and photodegradation of organic dyes under visible light. *Nanoscale* **8**, 440-450 (2016).
R30. Kirubasankar, B., Palanisamy, P., Arunachalam, S., Murugadoss, V. & Angaiah, S. 2D MoSe₂-Ni(OH)₂ nanohybrid as an efficient electrode material with high rate capability for asymmetric supercapacitor applications. *Chem. Eng. J.* **355**, 881-890 (2019).
R31. Li, X., & Peng, K. MoSe₂/Montmorillonite composite nanosheets: Hydrothermal synthesis, structural characteristics, and enhanced photocatalytic activity. *Minerals*, **8**, 268 (2018).

Second, according to the reviewer's comment, we evaluated the specific surface area of MoSe₂ and the LSC only as well, which was measured as 32.55 and 10.47 m² g⁻¹ (Fig. S6), respectively. Clearly, the surface area of LSC&MoSe₂ (142.09 m² g⁻¹) is much larger than that of each LSC and MoSe₂ only, which is consistent to the previous morphological analysis of the composite structured LSC&MoSe₂.

<Revised text>

Original text:

→ We then performed Brunauer–Emmett–Teller (BET) analysis on LSC&MoSe₂ and LSC to investigate the effect of MoSe₂ on the surface area of the heterostructure. As shown in Fig. 1f, the addition of MoSe₂ to LSC led to a notable increase in the total surface area of the composite structure, where the surface area of LSC&MoSe₂ (142.09 m² g⁻¹) was more than thrice that of LSC (39.95 m² g⁻¹).

Revised text:

→ We then performed Brunauer–Emmett–Teller (BET) analysis on LSC&MoSe₂ and LSC to investigate the effect of MoSe₂ on the surface area of the heterostructure. We also measured the BET surface area of MoSe₂ and LSC only as 32.55 and 10.47 m² g⁻¹, respectively (Supplementary Fig. 6). However, as shown in Fig. 1f, the addition of MoSe₂ to LSC led to a notable increase in the total surface area of the composite structure, where the surface area of LSC&MoSe₂ (142.09 m² g⁻¹) was more than thrice that of LSC (39.95 m² g⁻¹).

Added figure:

Figure S6. BET surface area calculated from N₂ adsorption/desorption isotherms of a) MoSe₂ and b) LSC only.

Third, regarding the comment on “If the active sites for OER and HER are coming from two different materials in the composite, then just by making the composite cannot increase the total number of active sites for both OER and HER”, we are afraid that our intention was not properly delivered to the reviewer. As the reviewer mentioned, IF the HER, OER active site comes from different materials, then it is likely that forming the composite structure may not bring the total active site increase for both the OER and HER. In this work, the phase transition of MoSe₂ and the electron affinity increase of Co attributed to electron transfer (from LSC to MoSe₂) brought synergetic effects to improve the overall electrocatalytic performance of LSC&MoSe₂ which is also closely related to the catalytically active sites (the surface area) of LSC&MoSe₂. That is, the observed increase in the surface area from the heterostructured composite contributed to improve

both the HER and OER performance via increasing the adsorption sites of each hydrogen and oxygen related intermediates as well as the electron and mass transport^{R32-34}. We modified the original manuscript to better convey our claims with supporting references as follows.

<Revised text>

Original text:

- Such a remarkable increase in surface area improves water splitting by increasing the total number of active electrocatalysis sites³².

Revised text:

- Such a remarkable increase in surface area improves water splitting by increasing the total number of active electrocatalysis sites **for both HER and OER**^{32,33}.

Original text:

- **Origin of improved electrochemical performance**

The morphology of the crystal structure of LSC&MoSe₂ was investigated to further elucidate the dramatic improvement in the electrochemical performance of LSC&MoSe₂ compared to other catalyst configurations such as MoSe₂ and LSC.

Revised text:

- **Synergetic effect for improved electrochemical performance**

The morphology of the crystal structure of LSC&MoSe₂ was investigated to further elucidate the dramatic improvement in the electrochemical performance of LSC&MoSe₂ compared to other catalyst configurations such as MoSe₂ and LSC.

Original text:

- Therefore, the enhanced OH⁻ affinity and electrophilicity in Co, that is, increase in Co-O and Co-OH in LSC, can ultimately improve the overall water electrolysis performance.

Revised text:

- Therefore, the enhanced OH⁻ affinity and electrophilicity in Co, that is, increase in Co-O and Co-OH in LSC, can ultimately improve the overall water electrolysis performance. **These charge transport phenomena bring synergistic effects to enhance both the HER and OER performance from the heterogenous composite structure of LSC&MoSe₂ in addition to the performance improvement factors mentioned in "Analysis of LSC&MoSe₂ properties" section.**

Original text:

- **Conclusion**

Formation of LSC&MoSe₂ led to several synergetic effects, including increased specific surface area, enhanced surface adsorption capability, favorable kinetics for the Volmer–Tafel pathway, and decreased charge transfer resistance, which contributed to its improved electrochemical catalytic performance.

Revised text:

- **Conclusion**

Formation of LSC&MoSe₂ led to several synergetic effects, including increased specific surface area **for both the HER and OER**, enhanced surface adsorption capability, favorable kinetics for the Volmer–Tafel pathway, and decreased charge transfer resistance, which contributed to its improved electrochemical catalytic performance.

Added references in the revised manuscript

- 33. Kim, N. I., Afzal, R. A., Choi, S. R., Lee, S. W., Ahn, D., Bhattacharjee, S., Lee, S. C., Kim, J. H. & Park, J. Y. Highly active and durable nitrogen doped-reduced graphene oxide/double perovskite bifunctional hybrid catalysts. *J. Mater. Chem. A* **5**, 13019-13031 (2017).

Supplementary references

- ➔ R32. Kong, D., Wang, H., Lu, Z. & Cui, Y. CoSe₂ nanoparticles grown on carbon fiber paper: an efficient and stable electrocatalyst for hydrogen evolution reaction. *J. Am. Chem. Soc.* **136**, 4897-4900 (2014).
- R33. Wang, J., Cui, W., Liu, Q., Xing, Z., Asiri, A. M. & Sun, X. Recent progress in cobalt based heterogeneous catalysts for electrochemical water splitting. *Adv. Mater.* **28**, 215-230 (2016).
- R34. Huang, J., Jin, B., Liu, H., Li, X., Zhang, Q., Chu, S., Peng, R. & Chu, S. Controllable synthesis of flower-like MoSe₂ 3D microspheres for highly efficient visible-light photocatalytic degradation of nitro-aromatic explosives. *J. Mater. Chem. A* **6**, 11424-11434 (2018).

Fourth, as the reviewer recommended, we examined the electrochemically active surface area (ECSA) profiles for all catalysts as follows (Fig. S2 and S3). As shown in Figure S3, the ECSA of LSC (LSC:MoSe₂:KB = 9:0:1) is measured as 6.305 mF cm⁻² and the ECSA value increases with the addition of MoSe₂ upto the weight ratio 6:3:1 of LSC:MoSe₂:KB (LSC&MoSe₂). Thus, LSC&MoSe₂ with the optimum ratio presents the highest ECSA value of 9.595 mF cm⁻² as similarly observed from the HER and OER polarization profiles (Fig. S1). In case of MoSe₂ (LSC:MoSe₂:KB = 0:9:1), the ECSA value is evaluated quite low as 1.100 mF cm⁻² due to the electrically semiconducting properties of MoSe₂. For this reason, adding more MoSe₂ to the optimum ratio (LSC:MoSe₂:KB = 5:4:1) resulted in substantial reduction of the ECSA value. We added the related discussion with additional explanation and Supplementary Figure in the revised manuscript as follows.

<Revised text>

Original text:

- ➔ LSC&MoSe₂ was prepared using the high-energy ball milling process (See Experimental Section for details).

Revised text:

- ➔ LSC&MoSe₂ was prepared using the high-energy ball milling process with the optimum weight ratio of LSC:MoSe₂:KB = 6:3:1 determined by the electrochemical analyses (Supplementary Fig. S1, S2, and S3) (See Experimental Section for details).

Original text:

- ➔ **Synthesis of LSC&MoSe₂.** LSC, MoSe₂, and LSC&MoSe₂ catalysts were prepared by high-energy milling process by a planetary ball mill system (PM-200, Retsch, Germany). For the milling process of LSC&MoSe₂, 300 mg of LSC, 150 mg of MoSe₂, and 50 mg of KB were dispersed in ethanol and ball-milled using Zr-balls at 400 rpm for 2 h. In case of LSC/KB and MoSe₂/KB synthesis, 450 mg of LSC, 50 mg of KB and 450 mg of MoSe₂, 50 mg of KB was used, respectively. Then, the powder was collected by drying the solvent in a 70 °C of oven for further analyses.

Revised text:

- ➔ **Synthesis of Catalysts.** To find the optimum ratio for LSC&MoSe₂ catalyst, LSC and MoSe₂ were high-energy milled with 10 wt.% of Ketjen black EC-600JD (KB) by a planetary ball mill system (PM-200, Retsch, Germany). For the milling process, the following weight ratios were examined with LSC:MoSe₂:KB of 9:0:1, 8:1:1, 7:2:1, 6:3:1, 5:4:1, and 0:9:1. Total weight of catalyst was maintained as 500 mg and each catalyst was dispersed in ethanol and ball-milled using Zr-balls at 400 rpm for 2 h. Thus, for synthesis of LSC:MoSe₂:KB = 6:3:1, 300 mg of LSC, 150 mg of MoSe₂, and 50 mg of KB was used. In case of LSC/KB, MoSe₂/KB, and LSC/MoSe₂ synthesis, 450 mg of LSC, 50 mg of KB, 450 mg of MoSe₂, 50 mg of KB, and 300 mg of LSC/150 mg of MoSe₂ was used, respectively. Then, the powder was collected by drying the solvent in a 70 °C of oven for further analyses.

Added figures:

Figure S2. Cyclic voltammograms (CVs) measured with various ratios of LSC and MoSe₂ composite catalysts in the double layer capacitance region at scan rates of 20, 40, 60, 80, 100, 120, 140, and 160 mV s⁻¹ in 1.0 M KOH solution. The measured catalyst configurations are: a) LSC:MoSe₂:KB = 9:0:1 (LSC), b) LSC:MoSe₂:KB = 8:1:1, c) LSC:MoSe₂:KB = 7:2:1, d) LSC:MoSe₂:KB = 6:3:1 (LSC&MoSe₂), e) LSC:MoSe₂:KB = 5:4:1, f) LSC:MoSe₂:KB = 0:9:1 (MoSe₂).

Figure S3. Linear fitting profiles of the extraction of the double-layer capacitance values (C_{dl}) allowing the estimation of the electrochemically active surface area ($\Delta j = (j_a - j_c)/2$).

We measured the electrochemically active surface area (ECSA) for all catalysts as shown in Fig. S2 and Fig. S3. The ECSA of LSC (LSC:MoSe₂:KB = 9:0:1) is measured as 6,305 mF cm⁻² and the ECSA value increases with the addition of MoSe₂ upto the weight ratio 6:3:1 of LSC, MoSe₂, and KB

(LSC&MoSe₂). Thus, LSC&MoSe₂ with the optimum ratio presents the highest ECSA value of 9.595 mF cm⁻² as similarly observed from the HER and OER polarization profiles (Fig. S1). In case of MoSe₂ only (LSC:MoSe₂:KB = 0:9:1), the ECSA value is evaluated to be quite low as 1.100 mF cm⁻² due to the electrically semiconducting properties of MoSe₂. For this reason, adding more MoSe₂ to the optimum ratio (LSC:MoSe₂:KB = 5:4:1) resulted in substantial reduction of the ECSA value.

Fifth, we analyzed the BJH pore size distributions and N₂ adsorption-desorption isotherms for MoSe₂ and LSC only to investigate the origin of mesoporous characteristics in LSC&MoSe₂, as the reviewer suggested. As shown in Figure S8, MoSe₂ only exhibited a sharp peak primarily centered at around 4 nm, indicating the presence of mesoporous structure, whereas the broad peak centered at 55 nm was observed for the LSC only. In addition, N₂ adsorption-desorption isotherms of MoSe₂ only shows type-IV characteristics with a hysteresis loop at relative pressures (P/P₀) from 0.45 to 1.0, suggesting the presence of mesopores. However, LSC only shows much less conspicuous type-IV character than that of MoSe₂ only within similar P/P₀ range. These results support that the mesoporous characteristics observed from LSC&MoSe₂ heterostructure mainly originated from the MoSe₂. We included the additional discussion in the revised manuscript and the related figure is presented in Supporting Information as follows.

<Revised text>

Original text:

- The pore size distribution in **Supplementary Fig. 1** shows that mesoporous pores (2–50 nm) are present on the surface of the heterostructure. This mesoporous-pore-sized catalyst with large surface area is expected to enhance water splitting efficiency by facilitating effective mass transfer within the catalyst³².

Revised text:

- The pore size distribution in **Supplementary Fig. 7** shows that mesoporous pores (2–50 nm) are present on the surface of the heterostructure. We analyzed the BJH pore size distributions of MoSe₂ and LSC only to investigate the origin of mesoporous characteristics in LSC&MoSe₂ (**Supplementary Fig. 8**). In MoSe₂, a sharp peak primarily centered at around 4 nm is observed, indicating the presence of mesoporous structure, whereas the broad peak centered at 55 nm appears for the LSC only. This result is consistent to the BET surface area analysis (**Supplementary Fig. 6**), where N₂ adsorption-desorption isotherms of MoSe₂ shows type-IV characteristics with a hysteresis loop at relative pressures (P/P₀) from 0.45 to 1.0 and LSC only shows much less conspicuous type-IV character than that of MoSe₂ within similar P/P₀ range, which suggests that the mesoporous characteristics observed from LSC&MoSe₂ heterostructure mainly originated from the MoSe₂^{34,35}. This mesoporous-pore-sized catalyst with large surface area is expected to enhance water splitting efficiency by facilitating effective mass transfer within the catalyst³².

Added figures:

Figure S8. BJH pore size distribution for a) MoSe₂ and b) LSC only.

Figure S6. BET surface area calculated from N₂ adsorption/desorption isotherms of a) MoSe₂ and b) LSC only.

Added references in the revised manuscript

- ➔ 34. Sun, P., Xu, L., Li, J., Zhai, P., Zhang, H., Zhang, Z. & Zhu, W. Hydrothermal synthesis of mesoporous Mg₃Si₂O₅(OH)₄ microspheres as high-performance adsorbents for dye removal. *Chem. Eng. J.* **334**, 377-388 (2018).
- 35. Chen, R., Yu, J. & Xiao, W. Hierarchically porous MnO₂ microspheres with enhanced adsorption performance. *J. Mater. Chem. A* **1**, 11682-11690 (2013).

11. Electronic measurements: Fluorescence intensity dependent proof of the presence of MoSe₂ in the material is misleading when the concentration of MoSe₂ is not the same in all cases. UV-vis spectrum in Fig. 3c corresponds to LCO&MoSe₂. From there how the authors could conclude the charge transfer phenomenon?

[Response]

- ➔ We thank the reviewer for the helpful comments. It is likely that the fluorescence (FL) intensity may be depending on the concentration of MoSe₂. Therefore, we remeasured the FL spectra using the same concentration of MoSe₂ (0.33 mg/mL) for each MoSe₂, LSC&MoSe₂, LSC/MoSe₂, and MoSe₂/KB. As expected, the FL intensity quenching occurred in all MoSe₂/KB, LSC&MoSe₂, and

LSC/MoSe₂, whereas characteristic peak intensity was observed for the MoSe₂. We revised Figure 3a and related contents in revised manuscript as shown below.

Regarding the UV-vis-NIR analysis, unlike 2H-TMDs (semiconducting), 1T-TMDs (metallic) is typically characterized by the disappearance of the exciton peaks in the absorption spectra^{R35-38}, and the LSC only exhibits distinctive absorption peaks at around 284, 460, and 1410 nm (Fig. R1). In general, metallic 1T-TMDs are synthesized from alkali metal-assisted intercalation processes via electron transfer from alkali metals to TMDs^{R9,38-44}. In this work, the charge transfer between LSC only and MoSe₂ induces local phase transition in TMDs, which consequently results in the disappearance of the exciton peaks from the absorption spectrum of LSC&MoSe₂. Such featureless excitonic signature from the UV-vis-NIR spectrum is often related to the metal-like property induced by charge delocalization^{R45-47}. To avoid any potential ambiguity of relating the charge transfer phenomena directly to the absorption analysis, we modified the original manuscript as follows.

<Revised text>

Original text:

- Figure 3a shows the fluorescence emission spectra (FL) of MoSe₂, LSC, and KB for various configurations. Pristine 2H-phase MoSe₂ is semiconducting in nature³⁸, and LSC and KB are highly conductive materials with metallic features^{39,40}.

Revised text:

- Figure 3a shows the fluorescence emission spectra (FL) of MoSe₂, LSC only, and KB for various configurations. For the FL measurement, each sample (MoSe₂, LSC&MoSe₂, LSC only and MoSe₂ heterostructure (denoted LSC/MoSe₂), and MoSe₂ and KB heterostructure (denoted MoSe₂/KB) was prepared using the same concentration of MoSe₂ (0.33 mg/mL). Detailed summary of abbreviations for each material system studied in this work is provided in Supplementary Table 3. Pristine 2H-phase MoSe₂ is semiconducting in nature⁴¹, and LSC only and KB are highly conductive materials with metallic features^{2,43}.

Original text:

- To further verify the charge transfer between LSC and MoSe₂, we performed optical absorption measurements. Figure 3b shows the UV-vis NIR spectrum of MoSe₂, which highlights the semiconducting feature of the 2H-phase with two exciton peaks at approximately 700 and 800 nm⁴¹. In LSC&MoSe₂, such excitonic peaks disappeared, indicating the occurrence of charge transfer between LSC and MoSe₂ (Fig. 3c).

Revised text:

- Moreover, optical absorption measurements were performed to further examine the effect of local phase transition in TMDs to metallic-phase in LSC&MoSe₂. Figure 3b shows the UV-vis-NIR spectrum of MoSe₂, which highlights the semiconducting feature of the 2H-phase with two exciton peaks at approximately 700 and 800 nm⁴¹. In LSC&MoSe₂, such excitonic peaks disappeared, indicating that the phase transformed TMDs induced metal-like characteristics in LSC&MoSe₂⁴⁵⁻⁴⁸ (Fig. 3c).

<Modified figure>

Original figure:

Figure 3. Analysis of physical properties of LSC&MoSe₂. Fluorescence emission spectra of a) MoSe₂, LSC&MoSe₂, LSC/MoSe₂, and MoSe₂/KB, illustrating the 2H-phase of the as-prepared MoSe₂ and electron transfers among MoSe₂, LSC, and KB. UV-vis NIR spectra

Revised figure:

Figure 3. Analysis of physical properties of LSC&MoSe₂. Fluorescence emission spectra of a) MoSe₂, MoSe₂/KB, LSC&MoSe₂, and LSC/MoSe₂ (The concentration of MoSe₂ in all samples is fixed as 0.33 mg/mL), illustrating the 2H-phase of the as-prepared MoSe₂ and electron transfers among MoSe₂, LSC, and KB. UV-vis NIR spectra

Supplementary Figure:

Figure R1. Optical absorption spectra of LSC only indicating the presence of distinctive absorption peaks of LSC.

Added references in the revised manuscript

- ➔ 45. Vishnoi, P., Sampath, A., Waghmare, U. V. & Rao, C. N. R. Covalent functionalization of nanosheets of MoS₂ and MoSe₂ by substituted benzenes and other organic molecules. *Chem. Eur. J.* **23**, 886-895 (2017).
- 46. Xia, Y., Wiesinger, J. M., MacDiarmid, A. G. & Epstein, A. J. Camphorsulfonic acid fully doped polyaniline emeraldine salt: conformations in different solvents studied by an ultraviolet/visible/near-infrared spectroscopic method. *Chem. Mater.* **7**, 443-445 (1995).
- 47. Monkman, A. P. & Adams, P. Observed anisotropies in stretch oriented polyaniline. *Synth. Met.* **41-43**, 627-633 (1991).
- 48. Rannou, P., Gawlicka, A., Berner, D., Pron, A., Nechtschein, M. & Djurado, D. Spectroscopic, structural and transport properties of conductive polyaniline processed from fluorinated alcohols. *Macromolecules* **31**, 3007-3015 (1998).

Supplementary references

- ➔ R35. Zhang, J., Du, P., Xu, D., Li, Y., Peng, W., Zhang, G., Zhang, F. & Fan, X. Near-Infrared Responsive MoS₂/Poly (N-isopropylacrylamide) Hydrogels for Remote Light-Controlled Microvalves. *Ind. Eng. Chem. Res.* **55**, 4526–4531 (2016).
- R36. Geng, X., Sun, W., Wu, W., Chen, B., Al-Hilo, A., Benamara, M., Zhu, H., Watanabe, F., Cui, J. & Chen, T. P. Pure and stable metallic phase molybdenum disulfide nanosheets for hydrogen evolution reaction. *Nat. Commun.* **7**, 10672 (2016).
- R37. Li, X., Lv, X., Li, N., Wu, J., Zheng, Y. Z., & Tao, X. One-step hydrothermal synthesis of high-percentage 1T-phase MoS₂ quantum dots for remarkably enhanced visible-light-driven photocatalytic H₂ evolution. *Appl. Catal. B: Environ.* **243**, 76-85 (2019).
- R38. Voiry, D., Salehi, M., Silva, R., Fujita, T., Chen, M., Asefa, T., Shenoy, V. B., Eda, G. & Chhowalla, M. Conducting MoS₂ nanosheets as catalysts for hydrogen evolution reaction. *Nano Lett.* **13**, 6222-6227 (2013).
- R39. Zheng, J., Zhang, H., Dong, S., Liu, Y., Nai, C. T., Shin, H. S., Jeong, H. Y., Liu, B. & Loh, K. P. High yield exfoliation of two-dimensional chalcogenides using sodium naphthalenide. *Nat. Commun.* **5**, 2995 (2016).

- R40. Py, M. A. & Haering, R. R. Structural destabilization induced by lithium intercalation in MoS₂ and related compounds. *Can. J. Phys.* **61**, 76-84 (1983).
- R41. Dines, M. B. Lithium intercalation via n-butyllithium of the layered transition metal dichalcogenides. *Mater. Res. Bull.* **10**, 287-291 (1975).
- R42. Whittingham, M. S., & Gamble Jr, F. R. The lithium intercalates of the transition metal dichalcogenides. *Mater. Res. Bull.* **10**, 363-371 (1975).
- R43. Miremadi, B. K. & Morrison, S. R. The intercalation and exfoliation of tungsten disulfide. *J. Appl. Phys.* **63**, 4970-4974 (1988).
- R44. Joensen, P., Frindt, R. F. & Morrison, S. R. Single-layer MoS₂. *Mater. Res. Bull.* **21**, 457-461 (1986).
- R45. Xia, Y., Wiesinger, J. M., MacDiarmid, A. G. & Epstein, A. J. Camphorsulfonic acid fully doped polyaniline emeraldine salt: conformations in different solvents studied by an ultraviolet/visible/near-infrared spectroscopic method. *Chem. Mater.* **7**, 443-445 (1995).
- R46. Monkman, A. P. & Adams, P. Observed anisotropies in stretch oriented polyaniline. *Synth. Met.* **41-43**, 627-633 (1991).
- R47. Rannou, P., Gawlicka, A., Berner, D., Pron, A., Nechtschein, M. & Djurado, D. Spectroscopic, structural and transport properties of conductive polyaniline processed from fluorinated alcohols. *Macromolecules* **31**, 3007-3015 (1998).

12. The fitting of the XPS spectra are improper, e.g. Fig S5 for LSC&MoSe₂. There are no details of fitting.

[Response]

→ We thank the reviewer for providing the constructive comments. According to the reviewer's suggestion, we provided more detailed analysis for the XPS results. The O 1s XPS spectra for LSC&MoSe₂ and LSC (Fig. S12) were deconvoluted into four characteristic oxygen peaks and the associated representing numerical values for A₀, B₀, C₀, and C₀/A₀ in Table S5 were modified to atom % for better clarity with additional component D₀. Similarly, we revised Table S4 corresponding to Figure S11 (XPS spectra for Co 2p). The revised XPS result of LSC&MoSe₂ is provided in the revised manuscript as follows.

<Revised text>

Original text:

→ The O 1s spectrum consists of lattice oxygen (denoted A₀: O²⁻), chemisorbed water (denoted B₀: water adsorbed on catalyst surface), and surface-active oxygen (denoted C₀: hydroxyl group, O⁻, O₂, and O₂²⁻), which are located at 529.9, 532.6, and 532.2 eV, respectively, in the case of LSC&MoSe₂ (**Supplementary Fig. 5**).

Revised text:

→ The O 1s spectrum consists of lattice oxygen (denoted A₀: lattice O²⁻), highly oxidative oxygen (denoted B₀: O₂²⁻/O), surface-active oxygen (denoted C₀: hydroxyl group (-OH)), and adsorbed water (denoted D₀: H₂O or CO₃²⁻), which are located at 529.9, 531.3, 532.6, and 533.5 eV, respectively, in the case of LSC&MoSe₂ (**Supplementary Fig. 12**).

<Modified figure and tables>

Original figure:

Figure S5. O 1s XPS spectra of LSC&MoSe₂ and LSC, which consist of lattice oxygen, chemisorbed water, and surface-active oxygen.

Revised figure:

Figure S12. O 1s XPS spectra of LSC&MoSe₂ and LSC, which consist of lattice oxygen, highly oxidative oxygen, surface-active oxygen, and adsorbed water.

Original table:

Table S4. Quantitative analysis of lattice oxygen (A_o), chemisorbed water (B_o), and surface-active oxygen (C_o) of LSC&MoSe₂ and LSC obtained from the XPS result in Figure S5.

	LSC&MoSe ₂ (area CPS. eV)	LSC (area CPS. eV)
A_o	4153.2	10644.9
B_o	1143.8	387.4
C_o	27973.8	58395.9
C_o/A_o	Ca. 6.7	Ca. 5.5

Revised table:

Table S1. Quantitative analysis of lattice oxygen (A_o), highly oxidative oxygen (B_o), surface-active oxygen (C_o), and adsorbed water (D_o) of LSC&MoSe₂ and LSC obtained from the XPS result in Figure S12.

	LSC&MoSe ₂ (atom %)	LSC (atom %)
A_o	10.5	10.6
B_o	53.6	53.5
C_o	33.6	28.7
D_o	2.3	7.2
C_o/A_o	Ca. 3.2	Ca. 2.7

Original table:

Table S3. Quantitative analysis of Co^{3+}/Co^{2+} ratio in LSC&MoSe₂ and LSC obtained from the XPS result in Figure S4.

	LSC&MoSe ₂ (area CPS. eV)	LSC (area CPS. eV)
Co 2p _{3/2} , Co ³⁺	10490.0	7137.4
Co 2p _{3/2} , Co ²⁺	5345.8	5215.9
Co ³⁺ /Co ²⁺	ca. 2.0	ca. 1.4

Revised table:

Table S4. Quantitative analysis of Co^{3+}/Co^{2+} ratio in LSC&MoSe₂ and LSC obtained from the XPS result in Figure S11.

	LSC&MoSe ₂ (atom %)	LSC (atom %)
Co 2p _{3/2} , Co ³⁺	66.2	57.8
Co 2p _{3/2} , Co ²⁺	33.8	42.2
Co ³⁺ /Co ²⁺	ca. 2.0	ca. 1.4

13. The authors have explained that higher ratio of Co^{3+} and Co^{2+} is responsible for improved OER performance but this contradicts many previous reports which infer that higher ratio of Co^{2+} and Co^{3+} are

better for OER because more the number of low coordinated unsaturated metal sites, more it can avail OH^- from electrolyte. [Ref: Adv. Mater. 2017, 29, 1606793, ACS Appl. Mater. Interfaces, 2016, 8 (50), 34474-34481] Moreover in the XPS spectrum of O 1s in mixed LSC and MoSe_2 , surface-active oxygen (C_0) actually denotes oxygen vacancy according to reference no. 46. So if there is higher ratio of C_0 and A_0 in the mixed catalyst, then there should be higher ratio of Co^{2+} to Co^{3+} . Because for the sake of electro neutrality of the material and to satisfy O vacancy, lower oxidation states of metal should be more. So these two trends contradict to each other.

[Response]

→ We thank the reviewer for carefully reviewing our manuscript and providing valuable comments. We double checked the papers mentioned by the reviewer. There are various perspectives for the interpretation of O 1s and Co 2p of perovskite oxide, and the approach for XPS spectra analysis of Co and O is quite different for the papers (Adv. Mater. 2017, 29, 1606793, ACS Appl. Mater. Interfaces, 2016, 8 (50), 34474-34481) mentioned by the reviewer from our work. First, there are differences in the types of materials used. The mentioned references discuss water splitting catalysts using iron-cobalt oxide nanosheets with abundant oxygen vacancies (Adv. Mater. 2017, 29, 1606793) and defect-rich cobalt-iron layered double hydroxide (ACS Appl. Mater. Interfaces, 2016, 8 (50), 34474-34481), whereas we studied the perovskite oxide focusing on Co and O. In addition, the main contributor of the OER performance improvement from the above two papers is the oxygen vacancy. In our work, since the oxygen vacancy of LSC& MoSe_2 and LSC are almost identical, the OER performance improvement is mainly attributed to the adsorbed oxygen species on the perovskite oxide catalyst surface rather than oxygen vacancy. Below, we provide detailed responses for both the O 1s and the Co 2p aspects.

- O 1s analysis aspect:

We deconvoluted the XPS spectra of the O 1s into four distinctive peaks (please refer to the response to question #12). The O 1s XPS spectra consist of lattice oxygen (denoted Ao: lattice O^{2-} , 529.9 eV), highly oxidative oxygen (denoted Bo: $\text{O}_2^{2-}/\text{O}^-$, 531.3 eV), surface-active oxygen (denoted Co: hydroxyl group (-OH), 532.6 eV), and adsorbed water (denoted Do: H_2O or CO_3^{2-} , 533.5 eV). As shown from the quantitative analysis of each peak in Table S5, Bo value, which is known to be associated with oxygen vacancy^{R48,49}, is similar in both LSC& MoSe_2 and LSC, which can be attributed to the synthesis process of LSC& MoSe_2 . High temperature material synthesis process typically has considerable effects for the formation of oxygen vacancies in crystal lattice structures^{R50}. For the synthesis of LSC& MoSe_2 in this work, each of the as-prepared LSC and MoSe_2 is synthesized without exposing them to high temperature processes, and the formation of heterostructure does not require high temperature environment as well. Consequently, we expected that there exists no significant difference in the oxygen vacancy state between LSC& MoSe_2 and LSC. The coverage of hydroxyl groups on the catalyst surface is known to greatly affect the OER performance^{R51-53}. The higher Co/Ao ratio in LSC& MoSe_2 compared to LSC indicates that more hydroxyl group is adsorbed onto the surface of LSC& MoSe_2 , which contributed to the improved OER performance observed in this work.

- Co 2p analysis aspect: Herein, we discuss the effect of $\text{Co}^{3+}/\text{Co}^{2+}$ ratio in terms of OER performance and electroneutrality of LSC& MoSe_2 .

OER performance and $\text{Co}^{3+}/\text{Co}^{2+}$ ratio: Regarding the reviewer's comment on the ratio of Co^{3+} and Co^{2+} , the papers mentioned by the reviewer refer to water splitting catalysts using iron-cobalt oxide nanosheets with abundant oxygen vacancies (Adv. Mater. 2017, 29, 1606793) and defect-rich cobalt-iron layered double hydroxide (ACS Appl. Mater. Interfaces, 2016, 8 (50), 34474-34481). However, the material system in our work is obviously different in that we used the perovskite oxide ($\text{La}_{0.5}\text{Sr}_{0.5}\text{CoO}_{3-\delta}$, Lanthanum Strontium Cobalt Oxide), and the OER performance of perovskite oxide-based electrocatalysts is reported to be influenced by the σ^* -orbital occupancy (e_g)^{R54-56}. The e-filing of Co^{3+} ($t_{2g}^5e_g^1$) and Co^{2+} ($t_{2g}^5e_g^2$) is 1 and 2,

respectively, and the higher $\text{Co}^{3+}/\text{Co}^{2+}$ ratio, the closer to near-unity e_g . That is, higher $\text{Co}^{3+}/\text{Co}^{2+}$ ratio increases the intrinsic OER activity by strengthening the binding of oxygen-related intermediate species at the B-site of perovskite oxide^{R57}.

Electroneutrality and $\text{Co}^{3+}/\text{Co}^{2+}$ ratio in LSC&MoSe₂: In cobalt oxide system, reduction in oxygen vacancy results in increase of $\text{Co}^{3+}/\text{Co}^{2+}$ ratio to maintain the electroneutrality. In this work using the perovskite oxide system, higher $\text{Co}^{3+}/\text{Co}^{2+}$ ratio was observed in LSC&MoSe₂ compared to the LSC with additional MoSe₂ present despite similar oxygen vacancy concentrations in both the LSC&MoSe₂ and LSC. This result may be attributed to the ligand effect where electron transfer takes place along the interface of two adjacent materials owing to the difference in electronic configurations, as previously reported^{R58-62}. Therefore, the electron transfer from LSC to MoSe₂ increases the $\text{Co}^{3+}/\text{Co}^{2+}$ ratio in LSC&MoSe₂ compared to that of LSC to maintain the overall electroneutrality of LSC&MoSe₂.

<Revised text>

Original text:

- This shift in the XPS peak was attributed to changes in the oxidation state of the LSC owing to the presence of additional MoSe₂. The high electronegativity of Mo may have altered the electronic structure (σ^* -orbital occupancy) of Co by decreasing the electron density in Co^{42,43}. The σ^* -orbital occupancy (e_g) of close to 1 of the perovskite oxides catalyst enhanced the binding of oxygen species at the B site, which contributed to improved OER performance⁴⁴.

Revised text:

- This shift in the XPS peak was attributed to changes in the oxidation state of the LSC owing to the presence of additional MoSe₂. The Co 2p of LSC&MoSe₂ is to have higher oxidation states than the LSC owing to the electron transfer at the interface of LSC and MoSe₂ while maintaining the overall electroneutrality of LSC&MoSe₂⁴⁹⁻⁵³. The high electronegativity of Mo may have altered the electronic structure (σ^* -orbital occupancy) of Co by decreasing the electron density in Co^{54,55}. The σ^* -orbital occupancy (e_g) of close to 1 of the perovskite oxides catalyst enhanced the binding of oxygen species at the B site, which contributed to improved OER performance⁴⁴.

Original text:

- A high density of surface-active oxygen species on the catalyst surface is known to improve OER performance^{10,46}, and the relative ratio between lattice and surface-active oxygen can be used as an indicator for evaluating OER performance^{47,48}.

Revised text:

- A high density of surface-active oxygen species on the catalyst surface, which is associated with oxygen vacancies and surface hydroxyl groups, is known to improve OER performance^{10,58}, and the relative ratio between lattice and surface-active oxygen can be used as an indicator for evaluating OER performance^{59,60}.

Original text:

- As shown in **Supplementary Table 4**, the higher C_0/A_0 ratio of LSC&MoSe₂ than that of LSC indicates larger surface coverage of hydroxide species in LSC&MoSe₂, which can improve its intrinsic OER performance.

Revised text:

- As shown in **Supplementary Table 5**, the higher C_0/A_0 ratio of LSC&MoSe₂ than that of LSC, i.e., metals with high oxidation states which enhances the adsorption ability for oxides in the LSC&MoSe₂ surface^{58,61,62} indicates larger surface coverage of hydroxide species in LSC&MoSe₂, which can improve its intrinsic OER performance.

Added references in the revised manuscript

- 49. Kim, S., Kwon, O., Kim, C., Gwon, O., Jeong, H. Y., Kim, K. H., Shin, J. & Kim, G. Strategy

for Enhancing Interfacial Effect of Bifunctional Electrocatalyst: Infiltration of Cobalt Nanooxide on Perovskite. *Adv. Mater. Interfaces* **5**, 1800123 (2018).

50. Park, S. A., Lee, E. K., Song, H. & Kim, Y. T. Bifunctional enhancement of oxygen reduction reaction activity on Ag catalysts due to water activation on LaMnO₃ supports in alkaline media. *Sci. Rep.* **5**, 13552 (2015).

51. Strasser, P., Koh, S., Anniyev, T., Greeley, J., More, K., Yu, C., Liu, Z., Kaya, S., Nordlund, D., Ogasawara, H., Toney, M. F. & Nilsson, A. Lattice-strain control of the activity in dealloyed core-shell fuel cell catalysts. *Nat. Chem.* **2**, 454-460 (2010).

52. Zhu, Y., Zhou, W. & Shao, Z. Perovskite/carbon composites: applications in oxygen electrocatalysis. *Small* **13**, 1603793 (2017).

53. Tang, W. & Henkelman, G. Charge redistribution in core-shell nanoparticles to promote oxygen reduction. *J. Chem. Phys.* **130**, 194504 (2009).

61. Malkhandi, S., Yang, B., Manohar, A. K., Manivannan, A., Prakash, G. S. & Narayanan, S. R. Electrocatalytic properties of nanocrystalline calcium-doped lanthanum cobalt oxide for bifunctional oxygen electrodes. *J. Phys. Chem. Lett.* **3**, 967-972 (2012).

62. Wu, N. L., Liu, W. R. & Su, S. J. Effect of oxygenation on electrocatalysis of La_{0.6}Ca_{0.4}CoO_{3-x} in bifunctional air electrode. *Electrochim. Acta* **48**, 1567-1571 (2003).

Supplementary references

→ R48. Zhuang, L., Ge, L., Yang, Y., Li, M., Jia, Y., Yao, X. & Zhu, Z. Ultrathin Iron-Cobalt Oxide Nanosheets with Abundant Oxygen Vacancies for the Oxygen Evolution Reaction. *Adv. Mater.* **29**, 1606793 (2017).

R49. Wang, Z., You, Y., Yuan, J., Yin, Y. X., Li, Y. T., Xin, S. & Zhang, D. Nickel-Doped La_{0.8}Sr_{0.2}Mn_{1-x}Ni_xO₃ Nanoparticles Containing Abundant Oxygen Vacancies as an Optimized Bifunctional Catalyst for Oxygen Cathode in Rechargeable Lithium-Air Batteries. *ACS Appl. Mater. Interfaces* **8**, 6520-6528 (2016).

R50. Kim, J., Yin, X., Tsao, K. C., Fang, S. & Yang, H. Ca₂Mn₂O₅ as oxygen-deficient perovskite electrocatalyst for oxygen evolution reaction. *J. Am. Chem. Soc.* **136**, 14646-14649 (2014).

R51. Malkhandi, S., Yang, B., Manohar, A. K., Manivannan, A., Prakash, G. S. & Narayanan, S. R. Electrocatalytic properties of nanocrystalline calcium-doped lanthanum cobalt oxide for bifunctional oxygen electrodes. *J. Phys. Chem. Lett.* **3**, 967-972 (2012).

R52. Wu, N. L., Liu, W. R. & Su, S. J. Effect of oxygenation on electrocatalysis of La_{0.6}Ca_{0.4}CoO_{3-x} in bifunctional air electrode. *Electrochim. Acta* **48**, 1567-1571 (2003).

R53. Hu, J., Wang, L., Shi, L. & Huang, H. Oxygen reduction reaction activity of LaMn_{1-x}Co_xO₃-graphene nanocomposite for zinc-air battery. *Electrochim. Acta* **161**, 115-123 (2015).

R54. Hua, B., Li, M., Sun, Y. F., Zhang, Y. Q., Yan, N., Chen, J., Thundat, T., Li, J. & Luo, J. L. A coupling for success: Controlled growth of Co/CoO_x nanoshoots on perovskite mesoporous nanofibres as high-performance trifunctional electrocatalysts in alkaline condition. *Nano Energy*, **32**, 247-254 (2017).

R55. Maitra, U., Naidu, B. S., Govindaraj, A. & Rao, C. N. R. Importance of trivalency and the e_g¹ configuration in the photocatalytic oxidation of water by Mn and Co oxides. *Proc. Natl. Acad. Sci. U.S.A.* **110**, 11704-11707 (2013).

R56. Wang, Q., Xue, Y., Sun, S., Li, S., Miao, H. & Liu, Z. La_{0.8}Sr_{0.2}Co_{1-x}Mn_xO₃ perovskites as efficient bi-functional cathode catalysts for rechargeable zinc-air batteries. *Electrochim. Acta* **254**, 14-24 (2017).

R57. Suntivich, J., May, K. J., Gasteiger, H. A., Goodenough, J. B. & Shao-Horn, Y. A perovskite oxide optimized for oxygen evolution catalysis from molecular orbital principles. *Science* **334**,

1383-1385 (2011).

R58. Kim, S., Kwon, O., Kim, C., Gwon, O., Jeong, H. Y., Kim, K. H., Shin, J. & Kim, G. Strategy for Enhancing Interfacial Effect of Bifunctional Electrocatalyst: Infiltration of Cobalt Nanooxide on Perovskite. *Adv. Mater. Interfaces* **5**, 1800123 (2018).

R59. Park, S. A., Lee, E. K., Song, H. & Kim, Y. T. Bifunctional enhancement of oxygen reduction reaction activity on Ag catalysts due to water activation on LaMnO₃ supports in alkaline media. *Sci. Rep.* **5**, 13552 (2015).

R60. Strasser, P., Koh, S., Anniyev, T., Greeley, J., More, K., Yu, C., Liu, Z., Kaya, S., Nordlund, D., Ogasawara, H., Toney, M. F. & Nilsson, A. Lattice-strain control of the activity in dealloyed core-shell fuel cell catalysts. *Nat. Chem.* **2**, 454-460 (2010).

R61. Zhu, Y., Zhou, W. & Shao, Z. Perovskite/carbon composites: applications in oxygen electrocatalysis. *Small* **13**, 1603793 (2017).

R62. Tang, W. & Henkelman, G. Charge redistribution in core-shell nanoparticles to promote oxygen reduction. *J. Chem. Phys.* **130**, 194504 (2009).

14. The surface adsorption capability test through TGA may be erroneous as MoSe₂ can get oxidized showing weight loss.

[Response]

→ We understand the concern from the reviewer in that MoSe₂ can be oxidized during the wet-air treatment. We note that TGA measurement was performed to comparatively analyze the adsorption capability of OH⁻ and H₂O groups to LSC&MoSe₂ and LSC by intentionally exposing them to wet-air condition^{R54,63,64}. The TGA result shows larger weight loss in LSC&MoSe₂ than LSC, indicating better surface adsorption capability for LSC&MoSe₂. To confirm whether oxidization of MoSe₂ can affect the weight loss in LSC&MoSe₂, we further performed TGA measurements on wet-air treated MoSe₂ in the same temperature range. As shown in Figure R2, MoSe₂ showed almost negligible weight loss upto 160 °C, and it has been previously reported that weight loss of oxidized MoSe₂ initiates at ca. 350 °C (SeO₂) and further occurs above ca. 600 °C (MoO₃)^{R65,66}. Therefore, the contribution of weight loss by oxidation of MoSe₂ can be excluded in this work.

Supplementary Figure:

Figure R2. TGA analysis of MoSe₂ pre-exposed to wet-air for 24 h.

Supplementary references

- ➔ R63. Hua, B., Li, M., Zhang, Y. Q., Sun, Y. F. & Luo, J. L. All In One Perovskite Catalyst: Smart Controls of Architecture and Composition toward Enhanced Oxygen/Hydrogen Evolution Reactions. *Adv. Energy Mater.* **7**, 1700666 (2017).
- R64. Ardizzone, S., Dioguardi, F. S., Mussini, T., Mussini, P. R., Rondinini, S., Vercelli, B. & Vertova, A. Microcrystalline cellulose powders: structure, surface features and water sorption capability. *Cellulose* **6**, 57-69 (1999).
- R65. Tang, Y., Zhao, Z., Wang, Y., Dong, Y., Liu, Y., Wang, X. & Qiu, J. Carbon-stabilized interlayer-expanded few-layer MoSe₂ nanosheets for sodium ion batteries with enhanced rate capability and cycling performance. *ACS Appl. Mater. Interfaces* **8**, 32324-32332 (2016).
- R66. Zheng, C., Chen, C., Chen, L., & Wei, M. A CMK-5-encapsulated MoSe₂ composite for rechargeable lithium-ion batteries with improved electrochemical performance. *J. Mater. Chem. A* **5**, 19632–19638 (2017).

15. Figure 3f caption and the numbers in the main text are not matching.

[Response]

- ➔ We thank for the reviewer's valuable comments. As the reviewer pointed out, the expression for HER and OER in Figure 3f caption does not match with Figure 3f. We switched the upper and lower plot in Figure 3f and the modified figure is shown below.

<Modified figure>

Original figure:

Figure 3. Analysis of physical properties of LSC&MoSe₂. f) Nyquist plots of HER (upper) and OER (lower) of MoSe₂, LSC, and LSC&MoSe₂.

Revised figure:

Figure 3. Analysis of physical properties of LSC&MoSe₂. f) Nyquist plots of HER (upper) and OER (lower) of MoSe₂, LSC, and LSC&MoSe₂.

16. There is no experimental synthetic information corresponding to LSC/MoSe₂.

[Response]

→ We thank the reviewer for carefully reviewing our manuscript and providing valuable comments. We revised the related contents in "Experimental Section" by adding synthetic information on LSC/MoSe₂ used in this work as follows.

<Revised text>

Original text:

→ **Synthesis of LSC&MoSe₂.** LSC, MoSe₂, and LSC&MoSe₂ catalysts were prepared by high-energy milling process by a planetary ball mill system (PM-200, Retsch, Germany). For the milling process of LSC&MoSe₂, 300 mg of LSC, 150 mg of MoSe₂, and 50 mg of KB were dispersed in ethanol and ball-milled using Zr-balls at 400 rpm for 2 h. In case of LSC/KB and MoSe₂/KB synthesis, 450 mg of LSC, 50 mg of KB and 450 mg of MoSe₂, 50 mg of KB was used, respectively. Then, the powder was collected by drying the solvent in a 70 °C of oven for further analyses.

Revised text:

→ **Synthesis of Catalysts.** To find the optimum ratio for LSC&MoSe₂ catalyst, LSC and MoSe₂ were high-energy milled with 10 wt.% of Ketjen black EC-600JD (KB) by a planetary ball mill system (PM-200, Retsch, Germany). For the milling process, the following weight ratios were examined with LSC:MoSe₂:KB of 9:0:1, 8:1:1, 7:2:1, 6:3:1, 5:4:1, and 0:9:1. Total weight of catalyst was maintained as 500 mg and each catalyst was dispersed in ethanol and ball-milled using Zr-balls at 400 rpm for 2 h. Thus, for synthesis of LSC:MoSe₂:KB = 6:3:1, 300 mg of LSC, 150 mg of MoSe₂, and 50 mg of KB was used. In case of LSC/KB, MoSe₂/KB, and LSC/MoSe₂ synthesis, 450 mg of LSC, 50 mg of KB, 450 mg of MoSe₂, 50 mg of KB, and 300 mg of LSC/150 mg of MoSe₂ was used, respectively. Then, the powder was collected by drying the solvent in a 70 °C of oven for further analyses.

Reviewer: 2

Comments:

In this work entitled “In-situ Local Phase Transition of MoSe₂ in La_{0.5}Sr_{0.5}Co_{3-δ} Heterostructure and Excellent Overall Water Electrolysis over 1000 hours”, the authors demonstrated a perovskite oxides-transition metal dichalcogenides material with high surface adsorption capability, favorable kinetics and low charge transfer resistance which has great potential in overall water electrolysis application. The explanation of mechanism is clear and the analysis of data is detailed and basically correct. However, some issues should be addressed before considered for publication.

[Response]

→ We sincerely appreciate the reviewer for thoughtful and positive comments. The comments were greatly helpful to further improve the quality of our work. Below, we provide detailed point-by-point responses and revised work.

1. While the high energy ball-milling process might also cause the phase change of MoSe₂, the authors should compare the LSC&MoSe₂/KB sample and MoSe₂/KB for clarification.

[Response]

→ We thank the reviewer for carefully reviewing our manuscript and providing valuable comments. As the reviewer suggested, we prepared MoSe₂/KB under the same ball-milling condition as LSC&MoSe₂ to verify the occurrence of any phase transition of MoSe₂ in MoSe₂/KB from the ball-milling process. Then, XPS analysis was performed to the as-prepared samples, and we confirmed that local phase transition in MoSe₂ only occurred from the LSC&MoSe₂ sample, while no local phase transition was observed from MoSe₂/KB. As shown from Figure S13 and S14, XPS spectra of Mo 3d and Se 3d in LSC&MoSe₂ exhibit the presence of both 1T- and 2H-phase MoSe₂ (please see also the response to question 2 from Reviewer #1), while only 2H-phase related MoSe₂ peaks are found in MoSe₂/KB (2H-phase: Mo 3d_{5/2} for 229.0 eV, Mo 3d_{3/2} for 232.6 eV, Se 3d_{5/2} for 54.3 eV, and Se 3d_{3/2} for 55.6 eV), consistent to previously reported values for 1T- and 2H-phase MoSe₂^{R6-9, R67-70}. HR-TEM analysis shown in Figure S15 further confirms the 2H-MoSe₂ with hexagonal crystal structure in MoSe₂/KB. These results corroborate that the ball-milling energy in our experimental conditions has a negligible effect on the local phase transition of MoSe₂. During the revision process, we carried out DFT analysis to support this phase transition phenomenon, and the results also validated that the electron transfer from LSC to MoSe₂ induces the phase transition in MoSe₂ (Fig. 5 and Fig. S16, S17, and S18). We included the additional discussion in the revised manuscript and the related figure is presented in Supporting Information as follows.

<Revised text>

Original text:

→ The lattice spacings of Mo-Mo and Se-Se were 0.563 and 0.324 nm, respectively, confirming the presence of 1T-MoSe₂^{29,55}. These results suggest that the local phase transition in MoSe₂ from the 2H-phase to the 1T-phase occurred in-situ during formation of the heterostructure between LSC and 2H-phase MoSe₂.

Revised text:

→ The lattice spacings of Mo-Mo and Se-Se were 0.563 and 0.324 nm, respectively, confirming the presence of 1T-MoSe₂^{29,61}. The coexistence of 1T- and 2H-phase of MoSe₂ in LSC&MoSe₂ was verified by XPS analysis shown in **Supplementary Fig. 13** (1T-phase: Mo 3d_{5/2} for 228.3 eV, Mo 3d_{3/2} for 231.4 eV, Se 3d_{5/2} for 53.7 eV, and Se 3d_{3/2} for 54.7 eV, 2H-phase: Mo 3d_{5/2} for 229.0 eV, Mo 3d_{3/2} for 232.6 eV, Se 3d_{5/2} for 54.3 eV, and Se 3d_{3/2} for 55.6 eV)^{29,70-72}. The relative contents of 1T- and 2H-phase MoSe₂ from the as-prepared LSC&MoSe₂ are summarized in **Supplementary**

Table 6. To verify whether the ball milling process has any effect on the phase transition of MoSe_2 , MoSe_2/KB was synthesized under the same ball-mill process as $\text{LSC}\&\text{MoSe}_2$ and investigated through XPS and HR-TEM analysis. As shown in **Supplementary Fig. 14, 15**, no signature of phase transition is observed and only 2H-phase MoSe_2 is present in ball-milled MoSe_2/KB . These results suggest that the local phase transition in MoSe_2 from the 2H-phase to the 1T-phase occurred in-situ during formation of the heterostructure between LSC and 2H-phase MoSe_2 .

Added figures:

Figure S13. Mo 3d and Se 3d XPS spectra of $\text{LSC}\&\text{MoSe}_2$, indicating the coexistence of 1T- and 2H-phase MoSe_2 .

Figure S14. Mo 3d and Se 3d XPS spectra of MoSe_2/KB , indicating the presence of 2H-phase MoSe_2 only.

Figure S15. HR-TEM image of MoSe₂/KB shown with selected enlarged area illustrating the hexagonal crystal structure of 2H-phase MoSe₂ (scale bar: 0.5 nm).

Supplementary references

- ➔ R67. Barrera, D., Wang, Q., Lee, Y. J., Cheng, L., Kim, M. J., Kim, J. & Hsu, J. W. Solution synthesis of few-layer 2H MX₂ (M= Mo, W; X= S, Se). *J. Mater. Chem. C* **5**, 2859-2864 (2017).
- R68. Malmsten, G., Thorén, I., Högberg, S., Bergmark, J. E., Karlsson, S. E. & Rebane, E. Selenium compounds studied by means of ESCA. *Phys. Scr.* **3**, 96 (1971).
- R69. Wa'el A, A. & Nelson, A. E. Characterization of MoSe₂ (0001) and ion-sputtered MoSe₂ by XPS. *J. mater. Sci.* **40**, 2679-2681 (2005).
- R70. Vishnoi, P., Sampath, A., Waghmare, U. V. & Rao, C. N. R. Covalent functionalization of nanosheets of MoS₂ and MoSe₂ by substituted benzenes and other organic molecules. *Chem. Eur. J.* **23**, 886-895 (2017).

2. The authors suggest that during the ball-milling process the electrons transfer from Co to Mo spontaneously, so that the difference of products employing different ratios of LSC and MoSe₂ should be measured.

[Response]

- ➔ We thank for the reviewer's helpful comments. As the reviewer suggested, we further carried out HER and OER measurements with various ratios of LSC and MoSe₂ (Fig. S1). Both the HER and OER activities increased with the addition of MoSe₂ to LSC upto the weight ratio 6:3:1 of LSC:MoSe₂:KB (LSC&MoSe₂), reaching to the maximum performance. Additional increase of MoSe₂ to LSC (LSC:MoSe₂:KB = 5:4:1) resulted in substantial reduction in HER and OER performance due to the decreased phase transition in MoSe₂ flakes covering the LSC particles. Furthermore, we examined the electrochemically active surface area (ECSA) for all catalysts as shown in Figure S2 and S3. The ECSA of LSC (LSC:MoSe₂:KB = 9:0:1) is measured as 6.305 mF cm⁻² and the ECSA value increases with the addition of MoSe₂ upto the weight ratio 6:3:1 of LSC:MoSe₂:KB (LSC&MoSe₂). Thus, LSC&MoSe₂ with the optimum ratio presents the highest ECSA value of 9.595 mF cm⁻² as similarly observed from the HER and OER polarization profiles (Fig. S1). In case of MoSe₂ (LSC:MoSe₂:KB = 0:9:1), the ECSA value is evaluated quite low as 1.100 mF cm⁻² due to the electrically semiconducting properties of MoSe₂. For this reason, adding more MoSe₂ to the optimum ratio (LSC:MoSe₂:KB = 5:4:1) resulted in substantial reduction of the ECSA value. We included the additional discussion in the revised manuscript and the related figure is presented in Supporting Information as follows.

<Revised text and figures>

Original text:

→ LSC&MoSe₂ was prepared using the high-energy ball milling process (See Experimental Section for details). Morphological and structural analyses of the composite electrocatalyst were first performed by transmission electron microscopy (TEM).

Revised text:

→ LSC&MoSe₂ was prepared using the high-energy ball milling process with the optimum weight ratio of LSC:MoSe₂:KB = 6:3:1 determined by the electrochemical analyses (Supplementary Fig. S1, S2, and S3) (See Experimental Section for details). Morphological and structural analyses of the composite electrocatalyst were first performed by transmission electron microscopy (TEM) and scanning electron microscopy (SEM).

Original text:

→ **Synthesis of LSC&MoSe₂.** LSC, MoSe₂, and LSC&MoSe₂ catalysts were prepared by high-energy milling process by a planetary ball mill system (PM-200, Retsch, Germany). For the milling process of LSC&MoSe₂, 300 mg of LSC, 150 mg of MoSe₂, and 50 mg of KB were dispersed in ethanol and ball-milled using Zr-balls at 400 rpm for 2 h. In case of LSC/KB and MoSe₂/KB synthesis, 450 mg of LSC, 50 mg of KB and 450 mg of MoSe₂, 50 mg of KB was used, respectively. Then, the powder was collected by drying the solvent in a 70 °C of oven for further analyses.

Revised text:

→ **Synthesis of Catalysts.** To find the optimum ratio for LSC&MoSe₂ catalyst, LSC and MoSe₂ were high-energy milled with 10 wt.% of Ketjen black EC-600JD (KB) by a planetary ball mill system (PM-200, Retsch, Germany). For the milling process, the following weight ratios were examined with LSC:MoSe₂:KB of 9:0:1, 8:1:1, 7:2:1, 6:3:1, 5:4:1, and 0:9:1. Total weight of catalyst was maintained as 500 mg and each catalyst was dispersed in ethanol and ball-milled using Zr-balls at 400 rpm for 2 h. Thus, for synthesis of LSC:MoSe₂:KB = 6:3:1, 300 mg of LSC, 150 mg of MoSe₂, and 50 mg of KB was used. In case of LSC/KB, MoSe₂/KB, and LSC/MoSe₂ synthesis, 450 mg of LSC, 50 mg of KB, 450 mg of MoSe₂, 50 mg of KB, and 300 mg of LSC/150 mg of MoSe₂ was used, respectively. Then, the powder was collected by drying the solvent in a 70 °C of oven for further analyses.

Added figures:

Figure S1. a) HER and b) OER polarization curves obtained by the various weight ratios of LSC and MoSe₂. For the preparation of catalysts, 10 wt.% of KB was included as a conductive support.

Figure S2. Cyclic voltammograms (CVs) measured with various ratios of LSC and MoSe₂ composite catalysts in the double layer capacitance region at scan rates of 20, 40, 60, 80, 100, 120, 140, and 160 mV s⁻¹ in 1.0 M KOH solution. The measured catalyst configurations are: a) LSC:MoSe₂:KB = 9:0:1 (LSC), b) LSC:MoSe₂:KB = 8:1:1, c) LSC:MoSe₂:KB = 7:2:1, d) LSC:MoSe₂:KB = 6:3:1 (LSC&MoSe₂), e) LSC:MoSe₂:KB = 5:4:1, f) LSC:MoSe₂:KB = 0:9:1 (MoSe₂).

Figure S3. Linear fitting profiles of the extraction of the double-layer capacitance values (C_{dl}) allowing the estimation of the electrochemically active surface area ($\Delta j = (j_a - j_c)/2$).

We measured the electrochemically active surface area (ECSA) for all catalysts as shown in Fig. S2 and Fig. S3. The ECSA of LSC (LSC:MoSe₂:KB = 9:0:1) is measured as 6.305 mF cm⁻² and the ECSA value increases with the addition of MoSe₂ upto the weight ratio 6:3:1 of LSC, MoSe₂, and KB (LSC&MoSe₂). Thus, LSC&MoSe₂ with the optimum ratio presents the highest ECSA value of 9.595 mF cm⁻² as similarly observed from the HER and OER polarization profiles (Fig. S1). In case of MoSe₂ only (LSC:MoSe₂:KB = 0:9:1), the ECSA value is evaluated to be quite low as 1.100 mF cm⁻² due to the

electrically semiconducting properties of MoSe₂. For this reason, adding more MoSe₂ to the optimum ratio (LSC:MoSe₂:KB = 5:4:1) resulted in substantial reduction of the ECSA value.

3. While 1000 hours lifespan at high current density of 100 mA cm⁻² is enough to prove the durability of LSC&MoSe₂/KB, the authors should also give the energy efficiency of overall water electrolysis at such high current density to improve the practicability of this material.

[Response]

→ We appreciate the reviewer's comments for enhancing the quality of this study. We calculated the energy efficiency of overall water electrolysis at the current density of 100 mA cm⁻² as follows. The specific energy for producing 1 kg of hydrogen is thermodynamically given as 143 MJ kg⁻¹ or 39.4 kWh kg⁻¹. Since the electrolysis cell operates near 2.3 V at 100 mA cm⁻², the energy required for producing 1 kg H₂ can be calculated as follows. The current density of 100 mA cm⁻² can be expressed as 0.1 C s⁻¹ cm⁻² (∵ 1 A = 1 C s⁻¹). Then, the transferred amount of electron can be calculated as 1.036 × 10⁻⁶ mol s⁻¹ cm⁻² (∵ F = 96500 C mol⁻¹ e⁻), and H₂ generation rate is calculated as 5.181 × 10⁻⁷ mol H₂ s⁻¹ cm⁻², which is equivalent to 3.731 × 10⁻³ g H₂ h⁻¹ cm⁻². For producing 1 kg of H₂, the multiplication constant can be calculated as 2.681 × 10⁵ h cm². Since the electrolysis cell operates at 0.23 W cm⁻², the energy required for producing 1 kg H₂ is calculated as 61.65 kWh. The energy efficiency can be calculated by dividing the theoretical specific energy for 1 kg H₂ production, *i.e.*, 39.4 kWh. Then, the energy efficiency turns out to be 63.9 %. Considering the energy efficiency for conventional alkaline electrolysis is less than 70 % with the use of platinum electrode^{R71}, the proposed LSC&MoSe₂ catalyst can be considered as highly efficient for the water electrolysis. We included the additional discussion in the revised manuscript as follows.

<Revised text>

Original text:

→ By contrast, LSC&MoSe₂ || LSC&MoSe₂ exhibited remarkably stable operation over 1,000 h with negligible fluctuation in performance, demonstrating its exceptionally high electrochemical durability in HER and OER, even at a high current density of 100 mA cm⁻². **Supplementary Fig. 6** shows scanning electron microscopy (SEM) images of the LSC&MoSe₂ electrode after 1,000 h of the overall water electrolysis test; clogging or electrochemical/physical damage was not observed on the catalyst after the test.

Revised text:

→ By contrast, LSC&MoSe₂ || LSC&MoSe₂ exhibited remarkably stable operation over 1,000 h with negligible fluctuation in performance, demonstrating its exceptionally high electrochemical durability in HER and OER, even at a high current density of 100 mA cm⁻². Furthermore, the energy efficiency of overall water electrolysis at the current density of 100 mA cm⁻² was calculated to be 63.9 % as shown in **Supplementary Text 1**. Considering the energy efficiency for conventional alkaline electrolysis is less than 70 % with the use of noble metal-based electrode⁸¹, the proposed LSC&MoSe₂ catalyst can be considered as highly efficient for the water electrolysis. **Supplementary Fig. 21** shows SEM images of the LSC&MoSe₂ electrode after 1,000 h of the overall water electrolysis test; clogging or electrochemical/physical damage was not observed on the catalyst after the test.

<Added text in supporting information>

→ **Supplementary Text S1. Energy efficiency calculation.**

We calculated the energy efficiency of overall water electrolysis at the current density of 100 mA cm⁻² as follows. The specific energy for producing 1 kg of hydrogen is thermodynamically given as 143 MJ kg⁻¹ or 39.4 kWh kg⁻¹. Since the electrolysis cell operates near 2.3 V at 100 mA

cm^{-2} , the energy required for producing 1 kg H_2 can be calculated as follows. The current density of 100 mA cm^{-2} can be expressed as $0.1 \text{ C s}^{-1} \text{ cm}^{-2}$ ($\because 1 \text{ A} = 1 \text{ C s}^{-1}$). Then, the transferred amount of electron can be calculated as $1.036 \times 10^{-6} \text{ mol s}^{-1} \text{ cm}^{-2}$ ($\because F = 96500 \text{ C mol}^{-1} \text{ e}^-$), and H_2 generation rate is calculated as $5.181 \times 10^{-7} \text{ mol H}_2 \text{ s}^{-1} \text{ cm}^{-2}$, which is equivalent to $3.731 \times 10^{-3} \text{ g H}_2 \text{ h}^{-1} \text{ cm}^{-2}$. For producing 1 kg of H_2 , the multiplication constant can be calculated as $2.681 \times 10^5 \text{ h cm}^{-2}$. Since the electrolysis cell operates at 0.23 W cm^{-2} , the energy required for producing 1 kg H_2 is calculated as 61.65 kWh. The energy efficiency can be calculated by dividing the theoretical specific energy for 1 kg H_2 production, *i.e.*, 39.4 kWh. Then, the energy efficiency turns out to be 63.9 %. Considering the energy efficiency for conventional alkaline electrolysis is less than 70 % with the use of platinum electrode⁵³, the proposed LSC&MoSe₂ catalyst can be considered as highly efficient for the water electrolysis.

Added reference in the revised manuscript

- ➔ 81. Stolten, D. & Emonts, B. Hydrogen Science and Engineering: Materials, Process, Systems, and Technology (John Wiley & Sons, 2016).

Added reference in the supporting information

- ➔ 53. Stolten, D. & Emonts, B. Hydrogen Science and Engineering: Materials, Process, Systems, and Technology (John Wiley & Sons, 2016).

Supplementary reference

- ➔ R71. Stolten, D. & Emonts, B. Hydrogen Science and Engineering: Materials, Process, Systems, and Technology (John Wiley & Sons, 2016).

4. Please thoroughly check the whole manuscript and revise the typos and grammar errors.

[Response]

- ➔ We are grateful for the reviewer's comments. We double checked the whole manuscript for any typos and grammar errors and fixed them if necessary. In addition, we found the x-axis of Figure S7 is mislabeled (Wavelength (nm) → Pore Diameter (nm)), and the inset in Figure S9 is also fixed (Theoretically (-1.023 V) → Theoretically (-1.026 V)). We included the modified figures in Supporting Information as follows.

<Modified figure>

Original figure:

Figure S1. BJH pore size distribution for LSC&MoSe₂.

Revised figure:

Figure S7. BJH pore size distribution for LSC&MoSe₂.

Original figure:

Figure S2. Potential calibration of the Ag/AgCl reference electrode in 1 M KOH.

Revised figure:

Figure S1. Potential calibration of the Ag/AgCl reference electrode in 1 M KOH.

Reviewer: 3

Comments:

In general, the manuscript is very well written and the conclusions are well supported by the presented experimental data. I therefore only have a few questions/comments/suggestions:

[Response]

→ We sincerely appreciate the reviewer for providing the constructive comments to strengthen the quality of our manuscript. The detailed response to the comments of the reviewer is provided below.

1. In the introduction, it is mentioned that the HER should preferentially be performed in acidic media, and the OER in alkaline media. So why did the authors only investigate the performance of their materials in alkaline electrolytes? It would be interesting to know if the excellent stability could also be obtained in acidic electrolytes.

[Response]

→ We thank for the reviewer's comments. Hydrogen evolution reaction (HER) performs favorably in acidic media according to Nernst equation^{R71-74}. However, the overall water electrolysis test is generally performed in alkaline media because the oxygen evolution reaction (OER) is considerably sluggish than HER^{R75-84}. Since OER is considered as a rate-determining-step for overall water electrolysis, the use of efficient OER catalysts is essential and we herein adopted the perovskite oxide which is known with excellent OER efficiency. In this regard, we investigated the HER and OER catalytic activities under the alkaline media, and the overall water electrolysis test was also performed under the alkaline media for that matter. Unfortunately, the HER activities are difficult to be investigated in acidic media because perovskite oxides are very easily decomposed in acidic solutions. These are why the perovskite oxides are usually adopted in alkaline media as the water electrolysis catalyst^{R85-86}.

Supplementary references

- R71. Ganassin, A., Colic, V., Tymoczko, J., Bandarenka, A. S. & Schuhmann, W. Non-covalent interactions in water electrolysis: influence on the activity of Pt (111) and iridium oxide catalysts in acidic media. *Phys. Chem. Chem. Phys.*, **17**, 8349-8355 (2015).
- R72. McCrory, C. C., Uyeda, C. & Peters, J. C. Electrocatalytic hydrogen evolution in acidic water with molecular cobalt tetraazamacrocycles. *J. Am. Chem. Soc.* **134**, 3164-3170 (2012).
- R73. Deng, Y., Ye, C., Chen, G., Tao, B., Luo, H. & Li, N. EDTA-assisted hydrothermal synthesis of flower-like CoSe₂ nanorods as an efficient electrocatalyst for the hydrogen evolution reaction. *J. Energy Chem.* **28**, 95-100 (2019).
- R74. Wei, Y., He, W., Sun, P., Yin, J., Deng, X. & Xu, X. Synthesis of hollow Cu/Cu₂O/Cu₂S nanotubes for enhanced electrocatalytic hydrogen evolution. *Appl. Surf. Sci.* **476**, 966-971 (2019).
- R75. Jiang, N., You, B., Sheng, M. & Sun, Y. Electrodeposited Cobalt Phosphorous-Derived Films as Competent Bifunctional Catalysts for Overall Water Splitting. *Angew. Chem. Int. Ed.* **127**, 6349-6352 (2015).
- R76. Stern, L. A., Feng, L., Song, F. & Hu, X. Ni₂P as a Janus catalyst for water splitting: the oxygen evolution activity of Ni₂P nanoparticles. *Energy Environ. Sci.* **8**, 2347-2351 (2015).
- R77. Tang, C., Cheng, N., Pu, Z., Xing, W. & Sun, X. NiSe nanowire film supported on nickel foam: an efficient and stable 3D bifunctional electrode for full water splitting. *Angew. Chem. Int. Ed.* **127**, 9483-9487 (2015).
- R78. Wang, H., Lee, H. W., Deng, Y., Lu, Z., Hsu, P. C., Liu, Y., Lin, D. & Cui, Y. Bifunctional

non-noble metal oxide nanoparticle electrocatalysts through lithium-induced conversion for overall water splitting. *Nat. Commun.* **6**, 7261 (2015).

R79.Gao, X., Zhang, H., Li, Q., Yu, X., Hong, Z., Zhang, X., Liang, C. & Lin, Z. Hierarchical NiCo₂O₄ Hollow Microcuboids as Bifunctional Electrocatalysts for Overall Water-Splitting. *Angew. Chem. Int. Ed.* **128**, 6398–6402 (2016).

R80.Liu, T., Liu, Q., Asiri, A. M., Luo, Y., & Sun, X. An amorphous CoSe film behaves as an active and stable full water-splitting electrocatalyst under strongly alkaline conditions. *Chem. Commun.* **51**, 16683–16686 (2015).

R81.Tian, J., Cheng, N., Liu, Q., Sun, X., He, Y., & Asiri, A. M. Self-supported NiMo hollow nanorod array: an efficient 3D bifunctional catalytic electrode for overall water splitting. *J. Mater. Chem. A* **3**, 20056–20059 (2015).

R82.Gong, M., Zhou, W., Kenney, M. J., Kapusta, R., Cowley, S., Wu, Y., Lu, B., Lin, M. C., Wang, D. Y., Yang, J., Hwang, B. J. & Dai, H. Blending Cr₂O₃ into a NiO–Ni electrocatalyst for sustained water splitting. *Angew. Chem. Int. Ed.* **54**, 11989–11993 (2015).

R83.Masa, J., Weide, P., Peeters, D., Sinev, I., Xia, W., Sun, Z., Somsen, C., Muhler, M. & Schuhmann, W. Amorphous cobalt boride (Co₂B) as a highly efficient nonprecious catalyst for electrochemical water splitting: oxygen and hydrogen evolution. *Adv. Energy Mater.* **6**, 1502313 (2016).

R84.Wang, H., Min, S., Ma, C., Liu, Z., Zhang, W., Wang, Q., Li, D., Li, Y., Turner, S., Han, Y., Zhu, H., Abou-hamad, E., Hedhili, N. M., Pan, J., Yu, W., Huang, K. W., Li, L. J. Yuan, J., Antonietti, M. & Wu, T. Synthesis of single-crystal-like nanoporous carbon membranes and their application in overall water splitting. *Nat. Commun.* **8**, 13592 (2017).

R85. Xu, X., Chen, Y., Zhou, W., Zhu, Z., Su, C., Liu, M. & Shao, Z. A perovskite electrocatalyst for efficient hydrogen evolution reaction. *Adv. Mater.* **28**, 6442–6448 (2016).

R86. Hua, B., Li, M., Zhang, Y. Q., Sun, Y. F. & Luo, J. L. All-In-One Perovskite Catalyst: Smart Controls of Architecture and Composition toward Enhanced Oxygen/Hydrogen Evolution Reactions. *Adv. Energy Mater.* **7**, 1700666 (2017).

2. On page 9, several different sample abbreviations are used, but it remains unclear to me what is the difference between "LSC&MoSe₂" and "LSC/MoSe₂" and between "LSC" and "pure LSC"...

[Response]

→ We thank the reviewer for giving us the opportunity to provide more clear explanation regarding this matter. Below, we provide summary of abbreviations for each material system studied in our work and Figure S3 was modified accordingly.

<Revised text>

Original text:

→ In present work, we devise a composite perovskite oxide–TMD heterostructure composed of MoSe₂, La_{0.5}Sr_{0.5}CoO_{3-δ} (denoted LSC), and Ketjenblack carbon (denoted KB) as a bifunctional electrocatalyst for overall water electrolysis. The LSC and MoSe₂ heterostructure (denoted LSC&MoSe₂) offers considerably better HER and OER performances (onset potential, Tafel slope) than LSC or MoSe₂ alone.

Revised text:

→ In present work, we devise a composite perovskite oxide–TMD heterostructure composed of MoSe₂ (denoted MoSe₂), La_{0.5}Sr_{0.5}CoO_{3-δ} (denoted LSC only), and Ketjenblack carbon (denoted KB) as a bifunctional electrocatalyst for overall water electrolysis. The LSC, MoSe₂, and KB

heterostructure (denoted LSC&MoSe₂) offers considerably better HER and OER performances (onset potential, Tafel slope) than LSC and KB heterostructure (denoted LSC/KB) or MoSe₂ alone.

Original text:

- Figure 3a shows the fluorescence emission spectra (FL) of MoSe₂, LSC, and KB for various configurations. Pristine 2H-phase MoSe₂ is semiconducting in nature³⁸, and LSC and KB are highly conductive materials with metallic features^{39,40}.

Revised text:

- Figure 3a shows the fluorescence emission spectra (FL) of MoSe₂, LSC only, and KB for various configurations. For the FL measurement, each sample (MoSe₂, LSC&MoSe₂, LSC only and MoSe₂ heterostructure (denoted LSC/MoSe₂), and MoSe₂ and KB heterostructure (denoted MoSe₂/KB) was prepared using the same concentration of MoSe₂ (0.33 mg/mL). Detailed summary of abbreviations for each material system studied in this work is provided in Supplementary Table 3. Pristine 2H-phase MoSe₂ is semiconducting in nature⁴¹, and LSC only and KB are highly conductive materials with metallic features^{2,43}.

Original text:

- A clear FL peak was observed for MoSe₂ at 825 nm, whereas substantial FL quenching occurred in the MoSe₂ composite containing LSC and KB, that is, LSC&MoSe₂, LSC/MoSe₂, and MoSe₂/KB. In addition, the FL peak was not present in the case of KB, and only a weak FL spectrum was observed in the case of pure LSC, as shown in Supplementary Fig. 3. However, the FL peak was observed in the case of LSC. These results suggest that charge transfer occurs readily among MoSe₂, LSC, and KB while that between LSC and KB is limited.

Revised text:

- A clear FL peak was observed for MoSe₂ at 825 nm, whereas substantial FL quenching occurred in the MoSe₂ composite containing LSC only and KB, that is, LSC&MoSe₂, LSC/MoSe₂, and MoSe₂/KB. In addition, the FL peak was not present in the case of KB, and only a weak FL spectrum was observed in the case of LSC only, as shown in Supplementary Fig. 10. However, the FL peak was observed in the case of LSC. These results suggest that charge transfer occurs readily among MoSe₂, LSC only, and KB while that between LSC only and KB is limited.

<Modified figure and added supplementary table>

Original figure:

Figure S3. Fluorescence emission spectra of pure LSC, LSC, and KB.

Revised figure:

Figure S10. Fluorescence emission spectra of LSC only, LSC, and KB.

Added table:

Table S3. Summary of abbreviations for various materials studied in this work.

Abbreviation	Consisting materials
LSC only	$\text{La}_{0.5}\text{Sr}_{0.5}\text{Co}_{0.5}\text{O}_{3-d}$
KB	Ketjenblack carbon
MoSe_2	Molybdenum diselenide
LSC	$\text{La}_{0.5}\text{Sr}_{0.5}\text{Co}_{0.5}\text{O}_{3-d}$ + Ketjenblack carbon
LSC/ MoSe_2	$\text{La}_{0.5}\text{Sr}_{0.5}\text{Co}_{0.5}\text{O}_{3-d}$ + Molybdenum diselenide
MoSe_2/KB	Molybdenum diselenide + Ketjenblack carbon
LSC& MoSe_2	$\text{La}_{0.5}\text{Sr}_{0.5}\text{Co}_{0.5}\text{O}_{3-d}$ + Molybdenum diselenide + Ketjenblack carbon

3. The references used for benchmarking the new catalysts with data available in the literature seems a bit arbitrary, as a lot of different materials are presented, but this list is of course far from complete. Could the authors elaborate on the selection criteria they used for comprising Tables S1, S2 and S5?

[Response]

→ We thank for the reviewer's constructive comments. Previously, survey of Table S1 (HER), S2 (OER), and S9 (overall water electrolysis) was summarized by three representative criteria including "non-precious metal-based catalysts", "1 M KOH electrolyte", and "most recently

published results”. In addition to these criteria, we added subsidiary criteria of “transition metal dichalcogenides based electrocatalyst”, “perovskite oxide based electrocatalyst”, and “other representative non-precious metal based electrocatalyst”, all in 1 M KOH. Table S1 and S2 was updated with these additional criteria. Only marginal changes were made for Table S9 (stability table), i.e., with originally provided three criteria, owing to the limited availability of publications for the overall water splitting work in alkaline solution (1 M KOH) and chronopotentiometric stability test. We also corrected the onset potential value of HER in “this work” (Table S1), which was incorrectly provided from original manuscript. We included the revised Table S1, S2, and S9 in Supporting Information as follows.

<Revised tables>

Original table:

Table S1. HER performance survey of representative electrocatalysts in 1 M KOH electrolytes.

Catalyst	Onset potential (mV)	Tafel slope (mV dec ⁻¹)	Electrolyte	Ref.
This work	130	34	1 M KOH	
Mo ₂ C nanoparticles	150	60	1 M KOH	1
CoP nanowires/carbon cloth	38	129	1 M KOH	2
WN nanowires/carbon cloth	100	170	1 M KOH	3
NiNC-800	105	160	1 M KOH	4
FeP nanorod array	86	146	1 M KOH	5
EG/Co _{0.85} Se/NiFe-LDH	240	57	1 M KOH	6
CF-NG-Co	104	75	1 M KOH	7
Ni ₃ S ₂ nanoparticle/CNTs	350	102	1 M KOH	8
NiSe ₂ nanosheets	90	184	1 M KOH	9
Ni ₂ P nanoparticles	150	100	1 M KOH	10
NiCo ₂ S ₄ nanowires/carbon cloth	230	141	1 M KOH	11
NiSn@C	100	145	1 M KOH	12
MoS _{2+x} nanoparticles	200	84	1 M KOH	13
MoB particles	150	59	1 M KOH	14
Carbon paper/carbon tubes/Co-S	50	131	1 M KOH	15

Revised table:**Table S1.** HER performance survey of representative electrocatalysts in 1 M KOH electrolytes.

Catalyst	Onset potential (mV)	Tafel slope (mV dec ⁻¹)	Electrolyte	Ref.
This work	200	34	1 M KOH	
Transition metal dichalcogenides based electrocatalyst				
NiSe ₂ nanosheets	90	184	1 M KOH	1
MoS _{2+x} nanoparticles	200	84	1 M KOH	2
MoS ₂ /MoSe ₂ -0.5	180	96	1 M KOH	3
Pristine MoSe ₂	270	135	1 M KOH	3
Pristine MoS ₂	310	157	1 M KOH	3
Ni(OH) ₂ /MoS ₂	210	105	1 M KOH	4
CoSe ₂ /MoSe ₂	211	76	1 M KOH	5
MoSe ₂ @Ni _{0.85} Se	36	66	1 M KOH	6
MoSe ₂ /GCA	120	119	1 M KOH	7
ex-MoSe ₂ /NiCl ₂	230	114	1 M KOH	8
Perovskite oxide based electrocatalyst				
Ba _{0.5} Sr _{0.5} Co _{0.8} Fe _{0.2} O _{3-δ}	261	75	1M KOH	9
Pr _{0.5} (Ba _{0.5} Sr _{0.5}) _{0.5} Co _{0.8} Fe _{0.2} O _{3-δ}	179	45	1M KOH	9
NdBaMn ₂ O _{5.5}	200	87	1M KOH	10
SrNb _{0.1} Co _{0.7} Fe _{0.2} O _{3-δ} -nanorod	210	103	1M KOH	11
SrNb _{0.1} Co _{0.7} Fe _{0.2} O _{3-δ}	265	128	1M KOH	11
Pr(Ba _{0.8} Ca _{0.2}) _{0.95} (Co _{1.5} Fe _{0.5}) _{0.95} Co _{0.05} O _{5-δ}	200	42	1M KOH	12
La _{0.5} (Ba _{0.4} Sr _{0.4} Ca _{0.2}) _{0.5} Co _{0.8} Fe _{0.2} O _{3-δ}	180	59	1M KOH	13

$\text{Pr}_{0.5}(\text{Ba}_{0.5}\text{Sr}_{0.5})_{0.5}\text{Co}_{0.8}\text{Fe}_{0.2}\text{O}_{3-x}$	180	63	1 M KOH	13
Other representative non-precious metal based electrocatalyst				
Mo ₂ C nanoparticles	150	60	1 M KOH	14
CoP nanowires/carbon cloth	38	129	1 M KOH	15
WN nanowires/carbon cloth	100	170	1 M KOH	16
NiNC-800	105	160	1 M KOH	17
FeP nanorod array	86	146	1 M KOH	18
EG/Co _{0.85} Se/NiFe-LDH	240	57	1 M KOH	19
CF-NG-Co	104	75	1 M KOH	20
Ni ₃ S ₂ nanoparticle/CNTs	350	102	1 M KOH	21
Ni ₂ P nanoparticles	150	100	1 M KOH	22
NiCo ₂ S ₄ nanowires/carbon cloth	230	141	1 M KOH	23
NiSn@C	100	145	1 M KOH	24
MoB particles	150	59	1 M KOH	25
Carbon paper/carbon tubes/Co-S	50	131	1 M KOH	26
N, P, Co-doped graphene	350	145	1 M KOH	27

Original table:

Table S2. OER performance survey of representative electrocatalysts in 1 M KOH electrolytes.

Catalyst	Onset potential (V)	Tafel slope (mV dec ⁻¹)	Electrolyte	Ref.
This work	1.52	77	1 M KOH	
FeCo@NG/NCNT	1.54	77	1 M KOH	17
Co ₃ O ₄ -MTA	1.52	84	1 M KOH	18
NiNC-800	1.45	45	1 M KOH	4
FeNi ₃ N/Ni foam	1.43	40	1 M KOH	19

NPCN/CoNi–NCNT	1.57	165	1 M KOH	20
Ni–P	1.48	64	1 M KOH	21
Co ₃ O ₄ NCs	1.52	101	1 M KOH	22
Cu(OH) ₂	1.57	78.9	1 M KOH	23
MW CNT/Cu(OH) ₂	1.65	127.9	1 M KOH	23
MW CNT/CuO–400	1.55	59.9	1 M KOH	23
FeB ₂	1.48	52.4	1 M KOH	24
Ni _x B	1.54	89	1 M KOH	25
Nickel borate@Ni ₃ B	1.48	52	1 M KOH	26
Pristine CNTs	1.58	60	1 M KOH	27
Ni ₃ B	1.51	81.4	1 M KOH	28
Ni ₃ B–rGO	1.43	88.4	1 M KOH	28
O–CNTs	1.52	47.7	1 M KOH	29
N, O, P tri–doped porous carbon	1.52	84	1 M KOH	30

Revised table:

Table S2. OER performance survey of representative electrocatalysts in 1 M KOH electrolytes.

Catalyst	Onset potential (V)	Tafel slope (mV dec ⁻¹)	Electrolyte	Ref.
This work	1.52	77	1 M KOH	
Transition metal dichalcogenides based electrocatalyst				
Few-layer BP	1.45	88	1 M KOH	28
MoS₂/Co₃S₄ hollow polyhedra	1.64	90.1	1 M KOH	29
Co₉S₈@MoS₂/CNFs	1.58	61	1 M KOH	30
Fe–MoS₂	1.35	126	1 M KOH	30
MoS₂ QDs	1.51	39	1 M KOH	31

MoS ₂ /Ni ₃ S ₂	1.41	88	1 M KOH	32
CoTe ₂ nanowire	1.58	67	1 M KOH	33
Perovskite oxide based electrocatalyst				
3D microporous-LaFeO ₃	1.59	62	1 M KOH	34
3D microporous-LaFe _{0.8} Co _{0.2} O ₃	1.57	56	1 M KOH	34
La _{0.7} Sr _{0.3} Co _{0.7} Fe _{0.3} O ₃ -975	1.56	103	1 M KOH	35
La _{0.7} Sr _{0.3} Co _{0.25} Mn _{0.75} O ₃ -NPs-800	1.59	132	1 M KOH	36
SrCo _{0.4} Fe _{0.2} W _{0.4} O _{3-d}	1.63	58	1 M KOH	37
SrCo _{0.4} Fe _{0.2} W _{0.05} O _{3-d}	1.67	102	1 M KOH	37
Ball-milled SrCo _{0.4} Fe _{0.2} W _{0.4} O _{3-d}	1.58	81	1 M KOH	37
La _{0.5} Sr _{0.5} Ni _{0.2} Fe _{0.8} O _{3-d}	1.59	90	1 M KOH	38
La _{0.5} Sr _{0.5} Ni _{0.4} Fe _{0.6} O _{3-d}	1.47	85	1 M KOH	38
La _{0.5} Sr _{0.5} Ni _{0.5} Fe _{0.5} O _{3-d}	1.59	95	1 M KOH	38
La _{0.5} Sr _{0.5} Ni _{0.8} Fe _{0.2} O _{3-d}	1.59	96	1 M KOH	38
Ba _{0.5} Sr _{0.5} Co _{0.8} Fe _{0.2} O _{3-d}	1.55	80	1 M KOH	38
Other representative non-precious metal based electrocatalyst				
FeCo@NG/NCNT	1.54	77	1 M KOH	39
Co ₃ O ₄ -MTA	1.52	84	1 M KOH	40
NiNC-800	1.45	45	1 M KOH	17
FeNi ₃ N/Ni foam	1.43	40	1 M KOH	41
NPCN/CoNi-NCNT	1.57	165	1 M KOH	42
Ni-P	1.48	64	1 M KOH	43
Co ₃ O ₄ NCs	1.52	101	1 M KOH	44
Cu(OH) ₂	1.57	78.9	1 M KOH	45
MW CNT/Cu(OH) ₂	1.65	127.9	1 M KOH	45

MW CNT/CuO-400	1.55	59.9	1 M KOH	45
FeB ₂	1.48	52.4	1 M KOH	46
Ni _x B	1.54	89	1 M KOH	47
Nickel borate@Ni ₃ B	1.48	52	1 M KOH	48
Pristine CNTs	1.58	60	1 M KOH	49
Ni ₃ B	1.51	81.4	1 M KOH	50
Ni ₃ B-rGO	1.43	88.4	1 M KOH	50
O-CNTs	1.52	47.7	1 M KOH	51
N, O, P tri-doped porous carbon	1.52	84	1 M KOH	52

Original table:

Table S5. Survey of overall water splitting stability with current density and cell voltage of representative electrocatalysts in 1 M KOH electrolytes.

Catalyst	Current density (mA cm ⁻²)	Cell Voltage (V)	Stability (h)	Electrolyte	Ref.
This work	100	2.3	1000	1 M KOH	
Co-P film	4	η at 0.4	25	1 M KOH	31
Ni ₂ P	10	1.65	10	1 M KOH	32
NiSe/NF	20	1.75	20	1 M KOH	33
NiFeO _x /CF	10	1.51	200	1 M KOH	34
NiCo ₂ O ₄ Ni _{0.33} Co _{0.67} S ₂	4.5	1.65	20	1 M KOH	35
a-CoSe/Ti	10	1.7	27	1 M KOH	36
NiMo/TiM	10	1.64	10	1 M KOH	37
Ni@Cr ₂ O ₃ -NiO	20	1.5	500	1 M KOH	38
a-Co ₂ B	10	1.81	30	1 M KOH	39
a-Co ₂ B	30	2.04	10	1 M KOH	39

Nanoporous carbon/Co	20	1.57	5	1 M KOH	40
VOOH	50	1.75	50	1 M KOH	41
Ni/Mo ₂ C	18	1.74	10	1 M KOH	42
N-, O-, S- doped (NOSD) Co ₉ S ₈	50	1.85	10	1 M KOH	43
Na _{0.08} Ni _{0.9} Fe _{0.1} O ₂	16	1.6	12	1 M KOH	44
Ni@NC-800/NF	17	1.62	50	1 M KOH	4
N-Ni ₃ S ₂ /NF	20	1.55	8	1 M KOH	45
FeB ₂ /NF	50	1.7	4	1 M KOH	24
FeB ₂ /NF	10	1.55	16	1 M KOH	24
Ni ₁₁ (HPO ₃) ₈ -(OH) ₆	10	1.65	100	1 M KOH	46
SrNb _{0.1} Co _{0.7} Fe _{0.2} O _{3-δ}	10	1.7	30	1 M KOH	47
Cu _{0.3} Co _{2.7} P/N doped carbon	10	η at 10 mA cm ⁻²	50	1 M KOH	48
NiFe LDH@NiCoP/NF	10	1.57	100	1 M KOH	49

Revised table:

Table S9. Survey of overall water splitting stability with current density and cell voltage of representative electrocatalysts in 1 M KOH electrolytes.

Catalyst	Current density (mA cm ⁻²)	Cell Voltage (v)	Stability (h)	Electrolyte	Ref.
This work	100	2.3	1000	1 M KOH	
Co-P film	4	η at 0.4	25	1 M KOH	54
Ni ₂ P	10	1.65	10	1 M KOH	55
NiSe/NF	20	1.75	20	1 M KOH	56
NiFeO _x /CF	10	1.51	200	1 M KOH	57
NiCo ₂ O ₄ Ni _{0.33} Co _{0.67} S ₂	4.5	1.65	20	1 M KOH	58
a-CoSe/Ti	10	1.7	27	1 M KOH	59

NiMo/TiM	10	1.64	10	1 M KOH	60
Ni@Cr ₂ O ₃ -NiO	20	1.5	500	1 M KOH	61
a-Co ₂ B	10	1.81	30	1 M KOH	62
a-Co ₂ B	30	2.04	10	1 M KOH	62
Nanoporous carbon/Co	20	1.57	5	1 M KOH	63
VOOH	50	1.75	50	1 M KOH	64
Ni/Mo ₂ C	18	1.74	10	1 M KOH	65
N-, O-, S- doped (NOSD) Co ₉ S ₈	50	1.85	10	1 M KOH	66
Na _{0.08} Ni _{0.9} Fe _{0.1} O ₂	16	1.6	12	1 M KOH	67
Ni@NC-800/NF	17	1.62	50	1 M KOH	67
N-Ni ₃ S ₂ /NF	20	1.55	8	1 M KOH	68
FeB ₂ /NF	50	1.7	4	1 M KOH	46
FeB ₂ /NF	10	1.55	16	1 M KOH	46
Ni ₁₁ (HPO ₃) ₈ -(OH) ₆	10	1.65	100	1 M KOH	69
SrNb _{0.1} Co _{0.7} Fe _{0.2} O _{3-δ}	10	1.7	30	1 M KOH	70
Cu _{0.3} Co _{2.7} P/N doped carbon	10	η at 10 mA cm ⁻²	50	1 M KOH	71
NiFe LDH@NiCoP/NF	10	1.57	100	1 M KOH	72

Added supplementary information references

- 3. Zhou, Q., Zhao, G., Rui, K., Chen, Y., Xu, X., Dou, S. & Sun, W. Engineering Additional Edge Sites on Molybdenum Dichalcogenides toward Accelerated Alkaline Hydrogen Evolution Kinetics. *Nanoscale* DOI: 10.1039/C8NR08028C (2018).
4. Zhao, G., Lin, Y., Rui, K., Zhou, Q., Chen, Y., Dou, S. X. & Sun, W. Epitaxial growth of Ni(OH)₂ nanoclusters on MoS₂ nanosheets for enhanced alkaline hydrogen evolution reaction. *Nanoscale* **10**, 19074-19081 (2018).
5. Zhao, G., Li, P., Rui, K., Chen, Y., Dou, S. X. & Sun, W. CoSe₂/MoSe₂ Heterostructures with Enriched Water Adsorption/Dissociation Sites towards Enhanced Alkaline Hydrogen Evolution Reaction. *Chem. Eur. J.* **24**, 11158-11165 (2018).
6. Wang, C., Zhang, P., Lei, J., Dong, W. & Wang, J. Integrated 3D MoSe₂@ Ni_{0.85}Se nanowire network with synergistic cooperation as highly efficient electrocatalysts for hydrogen evolution reaction in alkaline medium. *Electrochim. Acta* **246**, 712-719 (2017).

7. Huang, Y., Lai, F., Zhang, L., Lu, H., Miao, Y. E. & Liu, T. Elastic carbon aerogels reconstructed from electrospun nanofibers and graphene as three-dimensional networked matrix for efficient energy storage/conversion. *Sci. Rep.* **6**, 31541 (2016).
8. Najafi, L., Bellani, S., Oropesa - Nuñez, R., An Bonaccorso, F. Doped MoSe₂ Nanoflakes/3d Metal Oxide-Hydr (Oxy) Oxides Hybrid Catalysts for pH Universal Electrochemical Hydrogen Evolution Reaction. *Adv. Energy Mater.* **8**, 1801764 (2018).
9. Xu, X., Chen, Y., Zhou, W., Zhu, Z., Su, C., Liu, M. & Shao, Z. A perovskite electrocatalyst for efficient hydrogen evolution reaction. *Adv. Mater.* **28**, 6442-6448 (2016).
10. Wang, J., Gao, Y., Chen, D., Liu, J., Zhang, Z., Shao, Z. & Ciucci, F. Water Splitting with an Enhanced Bifunctional Double Perovskite. *ACS Catal.* **8**, 364-371 (2017).
11. Zhu, Y., Zhou, W., Zhong, Y., Bu, Y., Chen, X., Zhong, Q., Liu, M. & Shao, Z. A perovskite nanorod as bifunctional electrocatalyst for overall water splitting. *Adv. Energy Mater.* **7**, 1602122 (2017).
12. Hua, B., Li, M., Sun, Y. F., Zhang, Y. Q., Yan, N., Chen, J., Thundat, T., Li, J. & Luo, J. L. A coupling for success: Controlled growth of Co/CoO_x nanoshoots on perovskite mesoporous nanofibres as high-performance trifunctional electrocatalysts in alkaline condition. *Nano Energy* **32**, 247-254 (2017).
13. Hua, B., Li, M., Zhang, Y. Q., Sun, Y. F., & Luo, J. L. All In One Perovskite Catalyst: Smart Controls of Architecture and Composition toward Enhanced Oxygen/Hydrogen Evolution Reactions. *Adv. Energy Mater.* **7**, 1700666 (2017).
28. Ren, X., Zhou, J., Qi, X., Liu, Y., Huang, Z., Li, Z., Ge, Y., Dhanabalan, S. C., Ponraj, J. S., Wang, S., Zhong, J. & Zhang, H. Few Layer Black Phosphorus Nanosheets as Electrocatalysts for Highly Efficient Oxygen Evolution Reaction. *Adv. Energy Mater.* **7**, 1700396 (2017).
29. Lei, X., Yu, K., Li, H. & Zhu, Z. A functional design and synthesis for electrocatalytic hydrogen evolution material on MoS₂/Co₃S₄ hybrid hollow nanostructure. *Electrochim. Acta*, **269**, 262-273 (2018).
30. Tang, B., Yu, Z. G., Seng, H. L., Zhang, N., Liu, X., Zhang, Y. W., Yang, W. & Gong, H. Simultaneous edge and electronic control of MoS₂ nanosheets through Fe doping for an efficient oxygen evolution reaction. *Nanoscale* **10**, 20113-20119 (2018).
31. Mohanty, B., Ghorbani-Asl, M., Kretschmer, S., Ghosh, A., Guha, P., Panda, S. K., Jena, B., Krashennikov, A. V. & Jena, B. K. MoS₂ Quantum Dots as Efficient Catalyst Materials for the Oxygen Evolution Reaction. *ACS Catal.* **8**, 1683-1689 (2018).
32. Zhang, J., Wang, T., Pohl, D., Rellinghaus, B., Dong, R., Liu, S., Zhuang, X. & Feng, X. Interface Engineering of MoS₂/Ni₃S₂ Heterostructures for Highly Enhanced Electrochemical Overall Water Splitting Activity. *Angew. Chem. Int. Ed.* **128**, 6814-6819 (2016).
33. Ji, L., Wang, Z., Wang, H., Shi, X., Asiri, A. M. & Sun, X. Hierarchical CoTe₂ Nanowire Array: An Effective Oxygen Evolution Catalyst in Alkaline Media. *ACS Sustainable Chem. Eng.* **6**, 4481-4485 (2018).
34. Dai, J., Zhu, Y., Zhong, Y., Miao, J., Lin, B., Zhou, W. & Shao, Z. Enabling High and Stable Electrocatalytic Activity of Iron Based Perovskite Oxides for Water Splitting by Combined Bulk Doping and Morphology Designing. *Adva. Mater. Interfaces* 1801317 (2018).
35. Majee, R., Chakraborty, S., Salunke, H. G. & Bhattacharyya, S. Maneuvering the Physical Properties and Spin States To Enhance the Activity of La-Sr-Co-Fe-O Perovskite Oxide Nanoparticles in Electrochemical Water Oxidation. *ACS Appl. Energy Mater.* **1**, 3342-3350 (2018).
36. Wan, M., Zhu, H., Zhang, S., Jin, H., Wen, Y., Wang, L., Zhang, M & Du, M. (2018). Building block nanoparticles engineering induces multi-element perovskite hollow nanofibers structure evolution to trigger enhanced oxygen evolution. *Electrochim. Acta* **279**, 301-310 (2018).
37. Chen, G., Hu, Z., Zhu, Y., Chen, Z. G., Zhong, Y., Lin, H. J., Chen, C. T., Tjeng, L. H., Zhou, W. & Shao, Z. Ultrahigh-performance tungsten-doped perovskites for the oxygen evolution reaction. *J. Mater. Chem. A* **6**, 9854-9859 (2018).

38. Wang, C. C., Cheng, Y., Ianni, E., Jiang, S. P. & Lin, B. A highly active and stable $\text{La}_{0.5}\text{Sr}_{0.5}\text{Ni}_{0.4}\text{Fe}_{0.6}\text{O}_{3-\delta}$ perovskite electrocatalyst for oxygen evolution reaction in alkaline media. *Electrochim. Acta* **246**, 997-1003 (2017).

4. The SEM images presented in Figure S6 should be of higher magnification in order to really observe any differences between before and after water splitting. Furthermore, it would also be interesting to show XPS data after overall water splitting in order to observe whether the claims made previously (ratio of surface-active oxygen vs lattice oxygen and Co^{3+} vs Co^{2+}) still hold during/after the water splitting experiments.

[Response]

→ We are grateful for the reviewer's thoughtful comments for enhancing the quality of this work. We added higher magnification SEM images of the electrode before and after the overall water splitting test, and XPS analysis results (ratio of Co^{3+} vs Co^{2+} and surface-active oxygen vs lattice oxygen) after chronopotentiometric stability test measured at 100 mA cm^{-2} for 1,000 h. As shown below, similar values of " $\text{Co}^{3+}/\text{Co}^{2+}$ " and "surface-active oxygen/lattice oxygen" ratio were observed even after the 1,000 h of overall water splitting measurements. These results further confirm the excellent stability of the proposed LSC&MoSe₂ as the electrocatalyst. Related discussions with revised figures are provided in the revised manuscript and Supporting Information as follows.

<Revised text>

Original text:

→ **Supplementary Fig. 6** shows scanning electron microscopy (SEM) images of the LSC&MoSe₂ electrode after 1,000 h of the overall water electrolysis test; clogging or electrochemical/physical damage was not observed on the catalyst after the test. Progress of the overall water splitting test with H₂ and O₂ bubble generation from the cathode and anode, respectively, is shown in **Supplementary Video 1**.

Revised text:

→ **Supplementary Fig. 21** shows SEM images of the LSC&MoSe₂ electrode after 1,000 h of the overall water electrolysis test; clogging or electrochemical/physical damage was not observed on the catalyst after the test. The chemical state of LSC&MoSe₂ after the stability test was further examined via XPS analysis. As shown from **Supplementary Fig. 22 and Table S7, S8**, the ratio of $\text{Co}^{3+}/\text{Co}^{2+}$ and surface-active oxygen/lattice oxygen showed almost negligible changes even after the 1,000 h of overall water splitting measurement, indicating the excellent stability of the proposed LSC&MoSe₂ as the electrocatalyst. Progress of the overall water splitting test with H₂ and O₂ bubble generation from the cathode and anode, respectively, is shown in **Supplementary Video 1**.

<Modified figure>

Original figure:

Figure S6. SEM images of the electro-sprayed pristine LSC&MoSe₂ electrode and LSC&MoSe₂ electrode after 1,000 h of overall water splitting test illustrating negligible electrode damage after the test.

Revised figure:

Figure S21. SEM images of the electro-sprayed pristine LSC&MoSe₂ electrode and LSC&MoSe₂ electrode after 1,000 h of overall water splitting test illustrating negligible electrode damage after the test.

<Added figure and tables>

Added figure:

Figure S22. Co 2p and O 1s XPS spectra of LSC&MoSe₂ after chronopotentiometric stability test measured at 100 mA cm⁻² for 1,000 h.

Added tables:

Table S7. Quantitative analysis of Co³⁺/Co²⁺ ratio in LSC&MoSe₂ obtained from the XPS result in Figure S22

LSC&MoSe ₂ (atom %)	
Co 2p _{3/2} , Co ³⁺	65.9
Co 2p _{3/2} , Co ²⁺	34.1
Co ³⁺ /Co ²⁺	ca. 2.0

Table S8. Quantitative analysis of lattice oxygen (A_o), highly oxidative oxygen (B_o), surface-active oxygen (C_o), and adsorbed water (D_o) of LSC&MoSe₂ obtained from the XPS result in Figure S22.

LSC&MoSe ₂ (atom %)	
A _o	10.5
B _o	53.6
C _o	33.6
D _o	2.3
C _o /A _o	ca. 3.2

REVIEWERS' COMMENTS:

Reviewer #1 (Remarks to the Author):

The authors have taken care in answering the comments with additional experiments and computational studies. Few minor changes are still needed.

Firstly numbers are provided up to second decimal without standard deviations, for example the surface area values. Error bars and standard deviations are required to validate the reproducibility of the results wherever possible.

Secondly, in Table S9, one of the best performing 2-electrode water splitting OER/HER couple is missing. NiFe-LDH and Ni₄Mo couple provides a cell voltage of 1.51 V at 10mAcm⁻². Does this table only consist of bifunctional catalysts? If not the couples are to be mentioned e.g. + || - or mentioned specifically in the Table caption.

Reviewer #2 (Remarks to the Author):

The authors have addressed those questions well, and this manuscript has been greatly improved. Therefore, I recommend it for publication as is.

Reviewer #3 (Remarks to the Author):

I thank the authors for their very thorough reply to the reviewer's comments and according revision of their manuscript.

I only place question marks at the new procedure used for obtaining the XRD patterns as a reply to comment 9 from reviewer 1. In my opinion, the XRD patterns of the actual material used for the electrochemical measurements should be provided, and not a "polished" XRD pattern in which the impurity phases are removed. Therefore, I suggest to go back to the original XRD pattern.

Reviewer(s)' Comments to Author:

Reviewer: 1

Comments:

The authors have taken care in answering the comments with additional experiments and computational studies. Few minor changes are still needed.

[Response]

→ We thank the reviewer for the thoughtful comments. The comments were greatly helpful to further improve the quality of our work. Below, we provide point-by-point responses and revised work.

1. Firstly numbers are provided up to second decimal without standard deviations, for example the surface area values. Error bars and standard deviations are required to validate the reproducibility of the results wherever possible.

[Response]

→ We are grateful for the reviewer's valuable comments. We went over the manuscript thoroughly and provided statistical analysis if applicable. We provided error bars and standard deviations for the surface area measurements of various samples investigated including LSC&MoSe₂, LSC, MoSe₂, and LSC only. We included the additional discussion in the revised manuscript and the related figure is presented in Supporting Information as follows.

<Revised text and added figure>

Original text:

→ We then performed Brunauer–Emmett–Teller (BET) analysis on LSC&MoSe₂ and LSC to investigate the effect of MoSe₂ on the surface area of the heterostructure. We also measured the BET surface area of MoSe₂ and LSC only as 32.55 and 10.47 m² g⁻¹, respectively (**Supplementary Fig. 6**). However, as shown in **Fig. 1f**, the addition of MoSe₂ to LSC led to a notable increase in the total surface area of the composite structure, where the surface area of LSC&MoSe₂ (142.09 m² g⁻¹) was more than thrice that of LSC (39.95 m² g⁻¹). Increase in the BET surface area for LSC&MoSe₂ can be attributed to the additional MoSe₂ nanoflakes present that are adsorbed onto the LSC surface.

Revised text:

→ We then performed Brunauer–Emmett–Teller (BET) analysis on LSC&MoSe₂ and LSC to investigate the effect of MoSe₂ on the surface area of the heterostructure. We also measured the BET surface area of MoSe₂ and LSC only as 32.55 and 10.47 m² g⁻¹, respectively (**Supplementary Figure 6**). However, as shown in **Fig. 1f**, the addition of MoSe₂ to LSC led to a notable increase in the total surface area of the composite structure, where the surface area of LSC&MoSe₂ (142.09 m² g⁻¹) was more than thrice that of LSC (39.95 m² g⁻¹). **Statistical analysis of the aforementioned samples is provided in Supplementary Figure 7**. Increase in the BET surface area for LSC&MoSe₂ can be attributed to the additional MoSe₂ nanoflakes present that are adsorbed onto the LSC surface.

Added figure:

Supplementary Figure 7 Summary of BET surface area measurements of LSC&MoSe₂, LSC, MoSe₂, and LSC only. Error bars indicate the standard deviation.

2. Secondly, in Table S9, one of the best performing 2-electrode water splitting OER/HER couple is missing. NiFe-LDH and Ni₄Mo couple provides a cell voltage of 1.51 V at 10mAcm⁻². Does this table only consist of bifunctional catalysts? If not the couples are to be mentioned e.g. + || - or mentioned specifically in the Table caption.

[Response]

→ We thank the reviewer's careful comments. The reviewer mentioned that NiFe-LDH and Ni₄Mo couple, which exhibits excellent water electrolysis performance, is missing in Table S9. In fact, Table S9 consists of two representative criteria based on "bifunctional catalyst" and "1 M KOH electrolyte", and we made a mistake on naming one sample information by misusing the notation of "+ || -", which is probably why the reviewer was confused. We are very sorry for this matter, and fixed the typo in Supplementary Table 9 and Figure 6d (NiCo₂O₄ || Ni_{0.33}Co_{0.67}S₂ → NiCo₂O₄) as follows. Nonetheless, we thank the reviewer for recommending a good performance electrocatalyst that we missed when preparing Supplementary Table 1 and 2. The relevant paper^{R1} (NiMo alloy for HER and NiFe-LDH for OER: exact values for OER are not explicitly stated) that we found after carefully searching the literature was added in Supplementary Table 1.

<Modified figure and supplementary table>

Original figure:

Figure 6. Overall water splitting performance. d) Comparison of the overall water electrolysis stability of various catalysts reported in the literature.

Revised figure:

Fig. 6 Overall water splitting performance. **d)** Comparison of the overall water electrolysis stability of various catalysts reported in the literature.

Original table:

Table S1. HER performance survey of representative electrocatalysts in 1 M KOH electrolytes.

Catalyst	Onset potential (mV)	Tafel slope (mV dec ⁻¹)	Electrolyte	Ref.
This work	200	34	1 M KOH	

Transition metal dichalcogenides based electrocatalyst

NiSe ₂ nanosheets	90	184	1 M KOH	1
MoS _{2+x} nanoparticles	200	84	1 M KOH	2
MoS ₂ /MoSe ₂ -0.5	180	96	1 M KOH	3
Pristine MoSe ₂	270	135	1 M KOH	3
Pristine MoS ₂	310	157	1 M KOH	3
Ni(OH) ₂ /MoS ₂	210	105	1 M KOH	4
CoSe ₂ /MoSe ₂	211	76	1 M KOH	5
MoSe ₂ @Ni _{0.85} Se	36	66	1 M KOH	6
MoSe ₂ /GCA	120	119	1 M KOH	7
ex-MoSe ₂ :NiCl ₂	230	114	1 M KOH	8
Perovskite oxide based electrocatalyst				
Ba _{0.5} Sr _{0.5} Co _{0.8} Fe _{0.2} O _{3-δ}	261	75	1M KOH	9
Pr _{0.5} (Ba _{0.5} Sr _{0.5}) _{0.5} Co _{0.8} Fe _{0.2} O _{3-δ}	179	45	1M KOH	9
NdBaMn ₂ O _{5.5}	200	87	1M KOH	10
SrNb _{0.1} Co _{0.7} Fe _{0.2} O _{3-δ} -nanorod	210	103	1M KOH	11
SrNb _{0.1} Co _{0.7} Fe _{0.2} O _{3-δ}	265	128	1M KOH	11
Pr(Ba _{0.8} Ca _{0.2}) _{0.95} (Co _{1.5} Fe _{0.5}) _{0.95} Co _{0.05} O _{5+δ}	200	42	1M KOH	12
La _{0.5} (Ba _{0.4} Sr _{0.4} Ca _{0.2}) _{0.5} Co _{0.8} Fe _{0.2} O _{3-δ}	180	59	1M KOH	13
Pr _{0.5} (Ba _{0.5} Sr _{0.5}) _{0.5} Co _{0.8} Fe _{0.2} O _{3-δ}	180	63	1M KOH	13
Other representative non-precious metal based electrocatalyst				
Mo ₂ C nanoparticles	150	60	1 M KOH	14
CoP nanowires/carbon cloth	38	129	1 M KOH	15
WN nanowires/carbon cloth	100	170	1 M KOH	16
NiNC-800	105	160	1 M KOH	17
FeP nanorod array	86	146	1 M KOH	18

EG/Co _{0.85} Se/NiFe-LDH	240	57	1 M KOH	19
CF-NG-Co	104	75	1 M KOH	20
Ni ₃ S ₂ nanoparticle/CNTs	350	102	1 M KOH	21
Ni ₂ P nanoparticles	150	100	1 M KOH	22
NiCo ₂ S ₄ nanowires/carbon cloth	230	141	1 M KOH	23
NiSn@C	100	145	1 M KOH	24
MoB particles	150	59	1 M KOH	25
Carbon paper/carbon tubes/Co-S	50	131	1 M KOH	26
N, P, Co-doped graphene	350	145	1 M KOH	27

Revised table:

Supplementary Table 1. HER performance survey of representative electrocatalysts in 1 M KOH electrolytes.

Catalyst	Onset potential (mV)	Tafel slope (mV dec ⁻¹)	Electrolyte	Ref.
This work	200	34	1 M KOH	
Transition metal dichalcogenides based electrocatalyst				
NiSe ₂ nanosheets	90	184	1 M KOH	1
MoS _{2+x} nanoparticles	200	84	1 M KOH	2
MoS ₂ /MoSe ₂ -0.5	180	96	1 M KOH	3
Pristine MoSe ₂	270	135	1 M KOH	3
Pristine MoS ₂	310	157	1 M KOH	3
Ni(OH) ₂ /MoS ₂	210	105	1 M KOH	4
CoSe ₂ /MoSe ₂	211	76	1 M KOH	5
MoSe ₂ @Ni _{0.85} Se	36	66	1 M KOH	6
MoSe ₂ /GCA	120	119	1 M KOH	7

ex-MoSe ₂ :NiCl ₂	230	114	1 M KOH	8
Perovskite oxide based electrocatalyst				
Ba _{0.5} Sr _{0.5} Co _{0.8} Fe _{0.2} O _{3-δ}	261	75	1M KOH	9
Pr _{0.5} (Ba _{0.5} Sr _{0.5}) _{0.5} Co _{0.8} Fe _{0.2} O _{3-δ}	179	45	1M KOH	9
NdBaMn ₂ O _{5.5}	200	87	1M KOH	10
SrNb _{0.1} Co _{0.7} Fe _{0.2} O _{3-δ} -nanorod	210	103	1M KOH	11
SrNb _{0.1} Co _{0.7} Fe _{0.2} O _{3-δ}	265	128	1M KOH	11
Pr(Ba _{0.8} Ca _{0.2}) _{0.95} (Co _{1.5} Fe _{0.5}) _{0.95} Co _{0.05} O _{5+δ}	200	42	1M KOH	12
La _{0.5} (Ba _{0.4} Sr _{0.4} Ca _{0.2}) _{0.5} Co _{0.8} Fe _{0.2} O _{3-δ}	180	59	1M KOH	13
Pr _{0.5} (Ba _{0.5} Sr _{0.5}) _{0.5} Co _{0.8} Fe _{0.2} O _{3-δ}	180	63	1M KOH	13
Other representative non-precious metal based electrocatalyst				
Mo ₂ C nanoparticles	150	60	1 M KOH	14
MoB particles	150	59	1 M KOH	14
CoP nanowires/carbon cloth	38	129	1 M KOH	15
WN nanowires/carbon cloth	100	170	1 M KOH	16
NiNC-800	105	160	1 M KOH	17
FeP nanorod array	86	146	1 M KOH	18
EG/Co _{0.85} Se/NiFe-LDH	240	57	1 M KOH	19
CF-NG-Co	104	75	1 M KOH	20
Ni ₃ S ₂ nanoparticle/CNTs	350	102	1 M KOH	21
Ni ₂ P nanoparticles	150	100	1 M KOH	22
NiCo ₂ S ₄ nanowires/carbon cloth	230	141	1 M KOH	23
NiSn@C	100	145	1 M KOH	24
Carbon paper/carbon tubes/Co-S	50	131	1 M KOH	25
N, P, Co-doped graphene	350	145	1 M KOH	26

Original table:**Table S9.** Survey of overall water splitting stability with current density and cell voltage of representative electrocatalysts in 1 M KOH electrolytes.

Catalyst	Current density (mA cm ⁻²)	Cell Voltage (v)	Stability (h)	Electrolyte	Ref.
This work	100	2.3	1000	1 M KOH	
Co-P film	4	η at 0.4	25	1 M KOH	54
Ni ₂ P	10	1.65	10	1 M KOH	55
NiSe/NF	20	1.75	20	1 M KOH	56
NiFeO _x /CF	10	1.51	200	1 M KOH	57
NiCo ₂ O ₄ Ni _{0.33} Co _{0.67} S ₂	4.5	1.65	20	1 M KOH	58
a-CoSe/Ti	10	1.7	27	1 M KOH	59
NiMo/TiM	10	1.64	10	1 M KOH	60
Ni@Cr ₂ O ₃ -NiO	20	1.5	500	1 M KOH	61
a-Co ₂ B	10	1.81	30	1 M KOH	62
a-Co ₂ B	30	2.04	10	1 M KOH	62
Nanoporous carbon/Co	20	1.57	5	1 M KOH	63
VOOH	50	1.75	50	1 M KOH	64
Ni/Mo ₂ C	18	1.74	10	1 M KOH	65
N-, O-, S- doped (NOSD) Co ₉ S ₈	50	1.85	10	1 M KOH	66
Na _{0.08} Ni _{0.9} Fe _{0.1} O ₂	16	1.6	12	1 M KOH	67
Ni@NC-800/NF	17	1.62	50	1 M KOH	17
N-Ni ₃ S ₂ /NF	20	1.55	8	1 M KOH	68
FeB ₂ /NF	50	1.7	4	1 M KOH	46

FeB ₂ /NF	10	1.55	16	1 M KOH	46
Ni ₁₁ (HPO ₃) ₈ -(OH) ₆	10	1.65	100	1 M KOH	69
SrNb _{0.1} Co _{0.7} Fe _{0.2} O _{3-δ}	10	1.7	30	1 M KOH	70
Cu _{0.3} Co _{2.7} P/N doped carbon	10	η at 10 mA cm ⁻²	50	1 M KOH	71
NiFe LDH@NiCoP/NF	10	1.57	100	1 M KOH	72

Revised table:

Supplementary Table 9. Survey of overall water splitting stability with current density and cell voltage of representative bifunctional electrocatalysts in 1 M KOH electrolytes.

Catalyst	Current density (mA cm ⁻²)	Cell Voltage (v)	Stability (h)	Electrolyte	Ref.
This work	100	2.3	1000	1 M KOH	
Co-P film	4	η at 0.4	25	1 M KOH	54
Ni ₂ P	10	1.65	10	1 M KOH	55
NiSe/NF	20	1.75	20	1 M KOH	56
NiFeO _x /CF	10	1.51	200	1 M KOH	57
NiCo ₂ O ₄	4.5	1.65	20	1 M KOH	58
a-CoSe/Ti	10	1.7	27	1 M KOH	59
NiMo/TiM	10	1.64	10	1 M KOH	60
Ni@Cr ₂ O ₃ -NiO	20	1.5	500	1 M KOH	61
a-Co ₂ B	10	1.81	30	1 M KOH	62
a-Co ₂ B	30	2.04	10	1 M KOH	62
Nanoporous carbon/Co	20	1.57	5	1 M KOH	63
VOOH	50	1.75	50	1 M KOH	64
Ni/Mo ₂ C	18	1.74	10	1 M KOH	65
N-, O-, S- doped (NOSD) Co ₉ S ₈	50	1.85	10	1 M KOH	66

$\text{Na}_{0.08}\text{Ni}_{0.9}\text{Fe}_{0.1}\text{O}_2$	16	1.6	12	1 M KOH	67
Ni@NC-800/NF	17	1.62	50	1 M KOH	17
N-Ni ₃ S ₂ /NF	20	1.55	8	1 M KOH	68
FeB ₂ /NF	50	1.7	4	1 M KOH	46
FeB ₂ /NF	10	1.55	16	1 M KOH	46
Ni ₁₁ (HPO ₃) ₈ -(OH) ₆	10	1.65	100	1 M KOH	69
SrNb _{0.1} Co _{0.7} Fe _{0.2} O _{3-δ}	10	1.7	30	1 M KOH	11
Cu _{0.3} Co _{2.7} P/N doped carbon	10	η at 10 mA cm ⁻²	50	1 M KOH	70
NiFe LDH@NiCoP/NF	10	1.57	100	1 M KOH	71

Added reference in Supplementary Information

- ➔ 27. Xu, W., Lu, Z., Wan, P., Kuang, Y. & Sun, X. High performance water electrolysis system with double nanostructured superaerophobic electrodes. *Small* **12**, 2492-2498 (2016).

Supplementary reference

- ➔ R1. Xu, W., Lu, Z., Wan, P., Kuang, Y. & Sun, X. High performance water electrolysis system with double nanostructured superaerophobic electrodes. *Small* **12**, 2492-2498 (2016).

Reviewer: 2

Comments:

The authors have addressed those questions well, and this manuscript has been greatly improved. Therefore, I recommend it for publication as is.

[Response]

→ We sincerely appreciate the reviewer for thoughtful and positive comments once again.

Reviewer: 3

Comments:

I thank the authors for their very thorough reply to the reviewer's comments and according revision of their manuscript.

[Response]

- We thank very much the positive response from the reviewer on our manuscript. The detailed response to the reviewer's comment is provided below.

1. I only place question marks at the new procedure used for obtaining the XRD patterns as a reply to comment 9 from reviewer 1. In my opinion, the XRD patterns of the actual material used for the electrochemical measurements should be provided, and not a "polished" XRD pattern in which the impurity phases are removed. Therefore, I suggest to go back to the original XRD pattern.

[Response]

- We thank the reviewer's prudent comments for this matter. In fact, throughout the entire work in preparing the LSC sample, impurity removed polished samples were utilized for various analyses including morphological, structural, spectroscopic, chemical, electrical, and electrochemical investigations. And as we answered in the first response to decision letter, we made a mistake when preparing the LSC samples for the XRD analysis by omitting the "polishing" process which inevitably led to high impurity LSC samples. Therefore, the newly provided XRD result in the first revised manuscript is indeed consistent with other analyses which involve the LSC sample. Again, we are very sorry for causing such unnecessary confusion to the reviewers. In brief, the followings are the typical experimental procedures adopted in this work for the preparation of LSC sample: (i) Sintering LSC pellet at 950 °C for 4 h. (ii) Surface-polishing of LSC pellet (Surface-impurity phases are removed). (iii) Grinding LSC pellet and obtaining LSC powder. XRD measurements are carried out after step (ii) and before step (iii), and the impurity phase observed in the XRD pattern from the original manuscript is due to the missing step of the surface-polishing of LSC pellet (step (ii)). We thank for the reviewer's comments again.